# EMERGENT DISCRETE CONTROLLER MODULES FOR SYMBOLIC PLANNING IN TRANSFORMERS

**S M Rafiuddin**
Oklahoma State University
srafiud@okstate.edu

**Muntaha Nujat Khan**
Oklahoma State University
munkhan@okstate.edu

## ABSTRACT

Transformers struggle with tasks that require *symbolic planning* loops, variable updates, and conditional branching, especially under length extrapolation. We introduce *discrete controller modules* that insert a small set of program primitives (ASSIGN, ADD, COMPARE, BRANCH) into Transformer blocks via a Gumbel–Softmax selector over operations and a compact program state of registers, flags, and optional memory. We prove that the augmented model can simulate any bounded-step program by mapping each primitive step to one controller step, and we bound the deviation of relaxed execution from its discrete trace by $O(\tau + \kappa^{-1})$ (selection temperature $\tau$, comparison sharpness $\kappa$). Empirically, the controller-augmented Transformer achieves strong length generalization on algorithmic benchmarks (Sorting, Sum-of-List, BFS), improving longest-length accuracy by up to 20–40 points over strong baselines, and yields consistent gains on symbolic QA (DROP) and program-synthesis-style tasks (RobustFill) with reduced compositionality drop-off. The learned execution is *interpretable*: operation traces align with ground truth, register roles are linearly decodable, and targeted knockouts cause localized accuracy losses. The approach adds only $\sim$5–7% FLOPs and can be applied sparsely (every $p$-th layer).

## 1 INTRODUCTION

Transformers (Vaswani et al., 2017) have revolutionized sequence modeling across natural language processing and beyond, yet they lack explicit mechanisms for symbolic planning constructs such as loops, variable assignments, and conditional branching. Concurrently, neuro-symbolic approaches have sought to integrate symbolic reasoning with deep learning, often by coupling external planners or memory-augmented modules (Kaiser & Sutskever, 2016; Graves et al., 2016). However, these methods either treat symbolic reasoning as a post-processing step or incur significant architectural complexity.

Recent work on algorithmic reasoning benchmarks highlights this limitation. The CLRS benchmark (Veličković et al., 2022) and ListOps tasks (Nangia & Bowman, 2018) reveal that standard Transformers struggle to generalize to longer problem instances, underscoring the need for inductive biases tailored to symbolic operations. Efforts to retrofit Transformers with neural modules, such as Universal Transformers with adaptive computation time (Dehghani et al., 2018) or the Compressive Transformer (Rae et al., 2020), introduce dynamic depth but still lack discrete control flow semantics. Unlike adaptive-depth controllers that modulate computation time, our goal is to endow layers with verifiable control-flow semantics (loops, branches, assignments) and produce step-wise, human-readable execution traces. Unlike external planners or generic memory modules, we maintain program state internally rather than delegating reasoning to post-processing or large external memories.

Differentiable discrete operations have been enabled by advances like the Gumbel-Softmax trick (Jang et al., 2017; Maddison et al., 2017) and its continuous relaxations, yet their integration into large-scale architectures remains underexplored. Furthermore, mechanistic probing studies (Geva et al., 2020) suggest that Transformers can incidentally learn symbolic behaviors, but without dedicated modules, such behaviors are neither reliable nor interpretable.

In this work, we embed an *explicit, learnable discrete controller* inside Transformer layers that selects from a finite opcode set with verifiable control-flow semantics, distinct from depth controllers (e.g., ACT) or generic external memory. Our contributions are: **(1)** a Gumbel-Softmax–parameterized controller that executes ASSIGN, ADD, COMPARE, and BRANCH with step-wise, human-readable traces; **(2)** a tight integration with attention and feed-forward computations that maintains a program state vector without external memory; **(3)** a formal expressivity result showing that $O(K+B)$ *instrumented steps* (or total depth $p(K+B)$ if every $p$-th block is instrumented) suffice to emulate any program in a *bounded imperative class* $\mathcal{P}_{K,B}$: straight-line code with conditionals and loops whose bodies are unrolled to at most $B$ iterations, with total primitive steps per execution at most $K+B$; approximation error can be made arbitrarily small under standard Gumbel-Softmax temperature annealing; and **(4)** empirical gains on algorithmic reasoning tasks (e.g., sorting, graph traversal) and symbolic QA benchmarks (e.g., DROP), including superior length generalization (e.g., 2–4× on tested tasks ) and improved interpretability, supported by ablations isolating controller discreteness and opcode design.

## 2 BACKGROUND

### 2.1 TRANSFORMER LAYERS

The Transformer architecture introduced by Vaswani et al. (Vaswani et al., 2017) models a sequence of $T$ tokens $\{x_1, \ldots, x_T\}$ by first mapping each token to a $D$-dimensional embedding and adding positional encodings. Let $\mathbf{X} \in \mathbb{R}^{T \times D}$ denote the resulting input matrix. A Transformer block consists of two sub-layers: multi-head self-attention and a position-wise feed-forward network. In the self-attention sub-layer, query, key, and value projections are computed as-

$$\mathbf{Q} = \mathbf{X}W^Q, \quad \mathbf{K} = \mathbf{X}W^K, \quad \mathbf{V} = \mathbf{X}W^V \tag{1}$$

where $W^Q, W^K, W^V \in \mathbb{R}^{D \times D_h}$ and $D_h = D/H$ for $H$ heads. Attention weights are given by-

$$\text{Attention}(\mathbf{Q}, \mathbf{K}, \mathbf{V}) = \text{softmax}\left(\frac{\mathbf{Q}\mathbf{K}^\top}{\sqrt{D_h}}\right)\mathbf{V} \tag{2}$$

and the outputs of all heads are concatenated and projected back to $\mathbb{R}^D$. The feed-forward sub-layer applies two linear transformations with a ReLU-

$$\text{FFN}(\mathbf{Z}) = \max(0, \mathbf{Z}W_1 + b_1)W_2 + b_2 \tag{3}$$

where $W_1 \in \mathbb{R}^{D \times D_{ff}}$, $W_2 \in \mathbb{R}^{D_{ff} \times D}$. Each sub-layer is wrapped with residual connections and layer normalization-

$$\mathbf{Z}' = \text{LayerNorm}(\mathbf{Z} + \text{SubLayer}(\mathbf{Z})) \tag{4}$$

Stacking $L$ such blocks yields the full Transformer encoder.

### 2.2 GUMBEL-SOFTMAX TRICK

Training models with discrete choices poses a challenge for gradient-based optimization. The Gumbel-Softmax trick (Jang et al., 2017; Maddison et al., 2017) provides a continuous, differentiable approximation to sampling from a categorical distribution. Given unnormalized log-probabilities (logits) $\{\log \alpha_i\}_{i=1}^K$, one draws i.i.d. Gumbel noise $g_i = -\log(-\log u_i)$ with $u_i \sim \text{Uniform}(0, 1)$ and computes-

$$y_i = \frac{\exp\big((\log \alpha_i + g_i)/\tau\big)}{\sum_{j=1}^K \exp\big((\log \alpha_j + g_j)/\tau\big)} \tag{5}$$

where $\tau > 0$ is a temperature parameter. As $\tau \to 0$, $y$ approaches a one-hot sample; for larger $\tau$, $y$ is a smooth approximation that permits backpropagation. During training, one may anneal $\tau$ to encourage discreteness while preserving gradient flow. This mechanism enables end-to-end learning of modules that make hard categorical decisions within neural architectures.

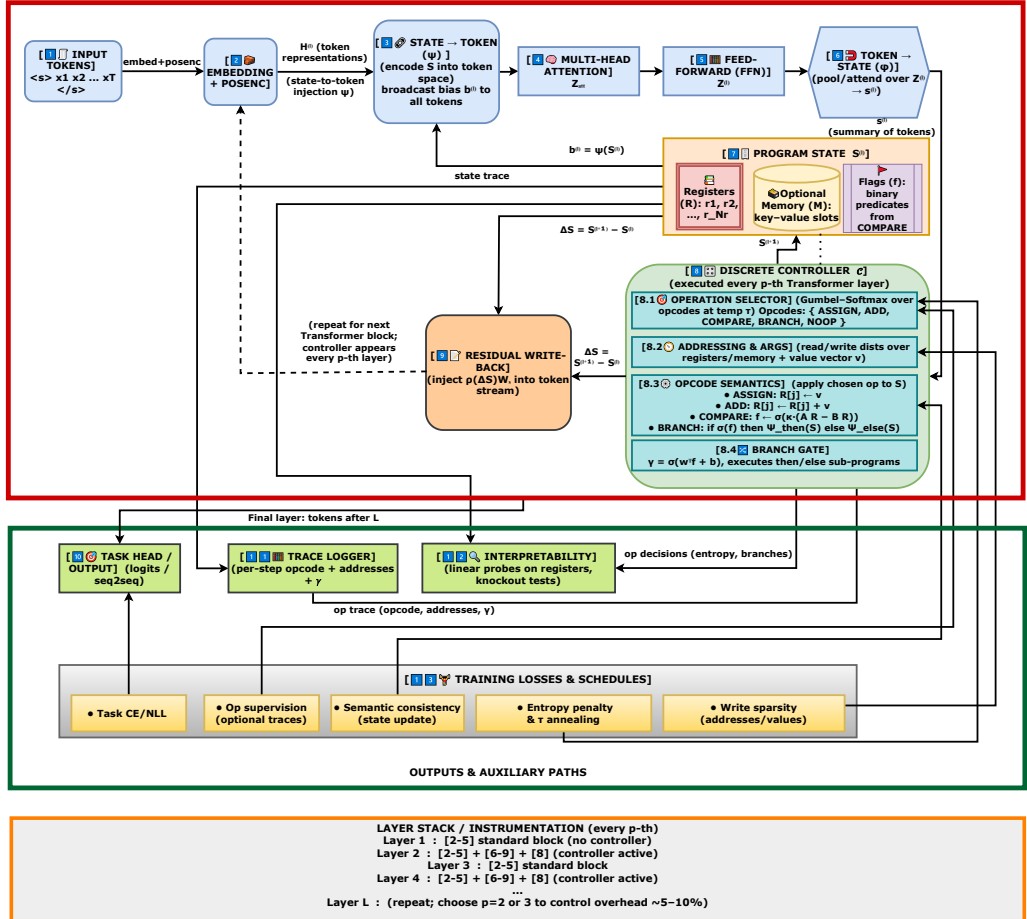

Figure 1: **Controller-augmented Transformer overview.** Tokens flow through embed→attn→FFN, while a compact *program state* (registers, optional memory, flags) interacts with the token stream via state↔token projections. A discrete controller (Gumbel–Softmax) selects one of {ASSIGN, ADD, COMPARE, BRANCH, NOOP} and updates the state; the resulting change is written back into the residual stream. We instrument every $p$-th layer (multi-step execution across depth) and log traces (opcode, addresses, gate) only at instrumented layers; overhead is ≈5–7% FLOPs for $p{=}2$. Training uses task loss with optional trace/entropy/semantic regularizers.

## 3 METHOD

### 3.1 CONTROLLER MODULE DESIGN

We augment each Transformer block with a small *discrete controller* $\mathcal{C}$ that selects and executes symbolic operations on a latent program state. Let $\mathbf{H}^{(l)} \in \mathbb{R}^{T \times D}$ be the token representations entering block $l$, and let the program state at step $l$ be-

$$\mathcal{S}^{(l)} = \left( \mathbf{R}^{(l)}, \mathbf{M}^{(l)}, \mathbf{f}^{(l)} \right) \tag{6}$$

where $\mathbf{R}^{(l)} \in \mathbb{R}^{N_r \times d_r}$ are $N_r$ *register* vectors, $\mathbf{M}^{(l)} \in \mathbb{R}^{N_m \times d_m}$ is an optional key–value *memory*, and $\mathbf{f}^{(l)} \in \{0,1\}^{N_f}$ are binary *flags*. The controller reads a summary $\mathbf{s}^{(l)} = \phi\left(\mathbf{H}^{(l)}\right) \in \mathbb{R}^{d_s}$ via a permutation-invariant pooling (mean/max or learned attention) and emits (i) an operation choice, (ii) its arguments (addresses and scalars), and (iii) an update to $\mathcal{S}$.

**Operation choice.** Let the operation set be $\mathcal{O} = \{\text{ASSIGN}, \text{ADD}, \text{COMPARE}, \text{BRANCH}, \text{NOOP}\}$. The controller computes logits $\boldsymbol{\pi}^{(l)} = W_o \, \mathbf{s}^{(l)} + b_o \in \mathbb{R}^{|\mathcal{O}|}$ and draws a differentiable one–hot via

Gumbel–Softmax-

$$\mathbf{y}^{(l)} = \text{softmax}\left(\frac{\boldsymbol{\pi}^{(l)} + \mathbf{g}}{\tau}\right) \tag{7}$$

$$\mathbf{g}_k = -\log(-\log u_k), \ u_k \sim \text{Uniform}(0, 1) \tag{8}$$

Temperature $\tau$ is annealed during training (Sec. 3.4).

**Addressing and arguments.** The controller predicts *read* and *write* distributions over registers and memory, and a value vector: $\boldsymbol{\alpha}^{(l)} = \text{softmax}(W_r \, \mathbf{s}^{(l)})$, $\boldsymbol{\beta}^{(l)} = \text{softmax}(W_w \, \mathbf{s}^{(l)})$, $\mathbf{v}^{(l)} = W_v \, \mathbf{s}^{(l)}$. Reads and writes are differentiable mixtures: $\mathbf{r}^{(l)} = \sum_{i=1}^{N_r} \alpha_i^{(l)} \, \mathbf{R}_i^{(l)}$, $\widetilde{\mathbf{R}}_j^{(l)} = \sum_{i=1}^{N_r} \beta_i^{(l)} \, \mathbf{u}_i^{(l)}(j)$

**Differentiable operation semantics.** Each operation induces a deterministic map on $(\mathbf{R}, \mathbf{M}, \mathbf{f})$, mixed by $\mathbf{y}^{(l)}$-

$$\mathcal{S}^{(l+1)} = \sum_{o \in \mathcal{O}} y_o^{(l)} \, \text{Sem}_o\big(\mathcal{S}^{(l)}, \mathbf{r}^{(l)}, \mathbf{v}^{(l)}\big) \tag{9}$$

We instantiate the primitives as follows (all elementwise where appropriate)

$$\text{Sem}_{\text{ASSIGN}} : \ \mathbf{R}^{(l+1)} = (1 - \lambda) \, \mathbf{R}^{(l)} + \lambda \, \mathbf{E}(\boldsymbol{\beta}^{(l)}) \, \mathbf{v}^{(l)\top} \tag{10}$$

$$\text{Sem}_{\text{ADD}} : \ \mathbf{R}^{(l+1)} = \mathbf{R}^{(l)} + \mathbf{E}(\boldsymbol{\beta}^{(l)}) \, \mathbf{v}^{(l)\top} \tag{11}$$

$$\text{Sem}_{\text{COMPARE}} : \ \mathbf{f}^{(l+1)} = \sigma\big(\kappa \, [A\mathbf{R}^{(l)} - B\mathbf{R}^{(l)}]\big) \tag{12}$$

$$\text{Sem}_{\text{BRANCH}} : \ \mathcal{S}^{(l+1)} = \gamma^{(l)} \, \Psi_{\text{then}}(\mathcal{S}^{(l)}) + \big(1 - \gamma^{(l)}\big) \, \Psi_{\text{else}}(\mathcal{S}^{(l)}) \tag{13}$$

$$\text{Sem}_{\text{NOOP}} : \ \mathcal{S}^{(l+1)} = \mathcal{S}^{(l)} \tag{14}$$

Here $\mathbf{E}(\boldsymbol{\beta}) \in \mathbb{R}^{N_r \times 1}$ is the one-hot *expected* write address (soft in practice), $\lambda \in (0, 1]$ a write strength, $\sigma$ a logistic with sharpness $\kappa \gg 1$, $A, B$ select two (soft) registers for comparison, and $\gamma^{(l)} = \sigma(w_\gamma^\top \mathbf{f}^{(l)} + b_\gamma)$ gates the branch. The branch subprograms $\Psi_{\text{then}}, \Psi_{\text{else}}$ are tiny MLPs sharing parameters across layers; they transform $\mathcal{S}$ but do not touch $\mathbf{H}$.

## 3.2 Integration into Transformer

Let a standard block be $\text{Block}(\cdot) = \text{FFN} \circ \text{MHA}(\cdot)$. We interleave controller updates with the residual pathway while exposing $\mathcal{S}^{(l)}$ to the token stream.

$$\text{(i) State-to-token:} \quad \mathbf{b}^{(l)} = \psi\big(\mathcal{S}^{(l)}\big) \in \mathbb{R}^D, \quad \widehat{\mathbf{H}}^{(l)} = \mathbf{H}^{(l)} + \mathbf{1} \, \mathbf{b}^{(l)\top},$$

$$\text{(ii) Token update:} \quad \mathbf{Z}^{(l)} = \text{Block}\big(\widehat{\mathbf{H}}^{(l)}\big),$$

$$\text{(iii) Token-to-state:} \quad \mathbf{s}^{(l)} = \phi\big(\mathbf{Z}^{(l)}\big), \quad \mathcal{S}^{(l+1)} = \mathcal{C}\big(\mathcal{S}^{(l)}, \mathbf{s}^{(l)}\big),$$

$$\text{(iv) Residual write-back:} \quad \mathbf{H}^{(l+1)} = \mathbf{Z}^{(l)} + \rho\big(\mathcal{S}^{(l+1)} - \mathcal{S}^{(l)}\big) W_s,$$

where $\psi$ is a learned state encoder, $\phi$ is a pooling (mean/max or attention with a learned query), $W_s$ aligns state changes to token space, and $\rho$ selects the register(s) whose changes should influence $\mathbf{H}$. Practically, we instrument every $p$-th block (e.g., $p \in \{1, 2, 3\}$), which controls compute while permitting multi-step programs across depth.

## 3.3 Theoretical Expressivity Analysis

We formalize a fragment of imperative programs and show our architecture can simulate it exactly in the limits $\tau \to 0$ and $\kappa \to \infty$. **Notation.** We use $L_{\text{net}}$ for the number of Transformer layers/blocks (cf. Table 4), and $L_{\text{tr}}$ for the execution-trace length (program steps). In this section, bounds are stated in terms of $K + B$ and $L_{\text{tr}} \le K + B$; when relating to depth we write $L_{\text{net}}$. **Program class.** Consider straight-line programs with conditionals and bounded loops over $N_r$ real-valued registers with primitives $\{\text{ASSIGN}, \text{ADD}, \text{COMPARE}, \text{BRANCH}\}$, where COMPARE sets Boolean flags consumed by BRANCH, and loops are unrolled to at most $B$ iterations. For completeness, the model's operation set includes NOOP, but the expressivity proof does not require it.

**Theorem 1** (Expressivity for bounded imperative programs). *For any program $\mathcal{P}$ in the class above with at most $K$ primitive steps and loop bound $B$, there exist a depth $L_{\text{net}} = O(K{+}B)$ (or total depth $p(K{+}B)$ if instrumenting every $p$-th block), widths $(D, d_r, d_m)$, and parameters of a controller-augmented Transformer such that, for input $(\mathbf{X}, \mathcal{S}^{(0)})$, the sequence of states $\{\mathcal{S}^{(l)}\}_{l=0}^{L_{\text{net}}}$ produced by the network matches the execution trace of $\mathcal{P}$ exactly as $\tau \to 0$, $c_o, c_a \to \infty$, and $\kappa \to \infty$.*

Here, $c_o$ and $c_a$ are scale factors applied to the operation and address logits, respectively. See Appendix D for the complete proof.

**Error under finite temperatures.** For finite scales, let $m_o$ be the minimum opcode-margin, $m_a$ the minimum address-margin, and $m_f$ the minimum absolute predicate margin across steps. Then per step-

$$1 - \mathbb{P}[o_l = o_l^\star] \leq (|\mathcal{O}| - 1)e^{-c_o m_o} \tag{15}$$

$$\|\boldsymbol{\alpha}^{(l)} - \mathbf{e}_{j^\star}\|_1 + \|\boldsymbol{\beta}^{(l)} - \mathbf{e}_{j^\star}\|_1 \leq 4e^{-c_a m_a} \tag{16}$$

$$\left|\sigma(\kappa z) - H(z)\right| \leq e^{-\kappa|z|} \tag{17}$$

and the straight-through Gumbel–Softmax smoothing contributes an additional $O(\tau)$. A union bound over $L$ steps yields a total deviation bounded by $O\!\left(Le^{-c_o m_o}\right) + O\!\left(Le^{-c_a m_a}\right) + O\!\left(Le^{-\kappa m_f}\right) + O(L\tau)$, which vanishes in the joint limit.

## 3.4 TRAINING OBJECTIVES AND LOSSES

Let $\theta$ denote all parameters. The total loss is-

$$\mathcal{L}(\theta) = \mathcal{L}_{\text{task}} + \lambda_{\text{op}}\mathcal{L}_{\text{op}} + \lambda_{\text{sem}}\mathcal{L}_{\text{sem}} + \lambda_{\text{ent}}\mathcal{L}_{\text{ent}} + \lambda_{\text{spar}}\mathcal{L}_{\text{spar}} \tag{18}$$

**Primary task loss.** $\mathcal{L}_{\text{task}}$ is cross-entropy (classification) or negative log-likelihood (seq2seq) on model outputs. **Operation supervision (optional).** When operation traces are available (synthetic tasks), we include $\mathcal{L}_{\text{op}} = \frac{1}{L}\sum_{l=1}^{L} \text{CE}\big(\mathbf{y}^{(l)}, \mathbf{y}^{*(l)}\big)$, where $\mathbf{y}^{*(l)}$ is the ground-truth primitive at step $l$. **Semantic consistency.** We penalize mismatch between predicted state transitions and their declarative semantics: $\mathcal{L}_{\text{sem}} = \frac{1}{L}\sum_{l=0}^{L-1} \left\|\mathcal{S}^{(l+1)} - \sum_o y_o^{(l)} \text{Sem}_o\big(\mathcal{S}^{(l)}, \mathbf{r}^{(l)}, \mathbf{v}^{(l)}\big)\right\|_2^2$. On tasks with known invariants (e.g., sum conservation), we add constraint penalties, e.g. $\|\mathbf{1}^\top \mathbf{R}^{(l+1)} - \mathbf{1}^\top \mathbf{R}^{(l)}\|_2^2$ for ADD-free steps. **Entropy annealing.** To encourage discrete decisions we minimize controller entropy: $\mathcal{L}_{\text{ent}} = \frac{1}{L}\sum_{l=1}^{L} H\big(\mathbf{y}^{(l)}\big)$, $\tau(t) = \max\big(\tau_{\min}, \tau_0 e^{-\gamma t}\big)$, with training step $t$, initial temperature $\tau_0$, floor $\tau_{\min}$, and decay $\gamma$. **Write sparsity.** We regularize address distributions and write magnitudes: $\mathcal{L}_{\text{spar}} = \frac{1}{L}\sum_{l=1}^{L} \big(\|\boldsymbol{\beta}^{(l)}\|_2^2 + \eta\|\mathbf{v}^{(l)}\|_2^2\big)$, which promotes near one–hot writes and small updates.

**Optimization details.** We use straight-through estimation for hard branches at late training if desired: during the forward pass $\mathbf{y}^{(l)} = \text{one\_hot}(\arg\max \boldsymbol{\pi}^{(l)})$; during the backward pass we substitute the Gumbel–Softmax gradient. We jointly train $\theta$ by AdamW with gradient clipping; auxiliary losses are ramped in over the first $K_{\text{warmup}}$ steps to stabilize early learning.

**Complexity.** Each controller adds $O(Dd_s + N_r d_r)$ parameters and $O(TDd_s + N_r d_r)$ FLOPs per *instrumented* block; with controllers inserted every $p$-th block and $d_s, d_r \ll D$, the relative FLOP overhead to a $D$-width Transformer is $\approx \frac{1}{2p}\big(\frac{d_s}{D}\big)^2$ (see App. G.1), typically 5–7% for $p{=}2$ in our settings.

## 4 EXPERIMENTS

### 4.1 DATASETS AND TASKS

**Synthetic algorithmic suite.** We construct three controlled tasks to probe algorithmic generalization: *(i) Sorting*, *(ii) Sum-of-List*, and *(iii) Graph Traversal*. For Sorting and Sum-of-List, sequences are length-$n$ lists of integers sampled i.i.d. from $\mathcal{U}\{0, 999\}$ with duplicates allowed; inputs are serialized as tokens `` $x_1, \ldots, x_n$ `` and outputs as either the sorted sequence or a single integer

token (the sum). We train on $n \in \{10, 20, 40, 80\}$ and evaluate both *in-distribution* (ID) and *length-extrapolation* at $n \in \{160, 320\}$. For Graph Traversal we sample $G(n, p)$ Erdős–Rényi graphs ($p = 0.2$) with $n \in \{10, 20, 40, 80\}$ nodes, encode adjacency lists and a start node, and require the model to emit a breadth-first traversal order (ties broken by node id). We generate 200k/20k/20k train/val/test examples for Sorting and Sum-of-List, and 120k/15k/15k for Graph Traversal (cf. Sec. 4.3 for generation details). **Program synthesis and math reasoning.** We include a *string-to-program* task derived from RobustFill-style benchmarks (Devlin et al., 2017), where the input is a set of input/output string pairs and the output is a DSL program that satisfies them; following (Devlin et al., 2017) we report exact match on held-out examples. We also use the Mathematics Dataset (Saxton et al., 2019) (algebra/arithmetic subsets) to test multi-step numeric reasoning with compositional templates. **Symbolic question answering.** We evaluate on DROP (Dua et al., 2019), which requires discrete operations (addition, counting, comparison) over paragraphs. Following prior work, we report F1 and Exact Match (EM). We construct a *numeric-only* subset (questions whose official program trace contains numeric operators) to target algorithmic reasoning.

## 4.2 BASELINES

We compare against strong families representative of the literature. **(B1) Transformer**: a standard encoder-decoder (or encoder-only for classification) without controllers (Vaswani et al., 2017). **(B2) Universal Transformer**: adaptive-depth variant with dynamic halting (Dehghani et al., 2018). **(B3) Compressive Transformer**: long-context memory via compressed history (Rae et al., 2020). **(B4) CLRS-Algorithmic Transformer**: an implementation following (Veličković et al., 2022) when applicable. **(B5) NPI/DNC**: Neural Programmer–Interpreter and Differentiable Neural Computer baselines (Kaiser & Sutskever, 2016; Graves et al., 2016). **(B6) Soft Modules**: our architecture with the controller's operation selection replaced by a *temperature-1* soft mixture (no discrete selection). We additionally include a *privileged* comparator, (B7) *External Planner*[†], which injects intermediate algorithmic hints into the token stream. Because B7 has access to supervision not provided to other models or ours, we report it separately and exclude it from any "vanilla baseline" aggregates and significance tests. For DROP, we additionally include **(B8) FiD-T5**[1] (Fusion-in-Decoder) as a strong reference system. All baselines are matched for depth/width as closely as possible; when memory-oriented models require extra parameters (e.g., Compressive), we report both accuracy and compute (FLOPs) to contextualize results.

## 4.3 IMPLEMENTATION DETAILS

**Model sizing.** Unless otherwise stated, we use $L = 12$ layers, hidden size $D = 512$, $H = 8$ heads, FFN width $D_{ff} = 2048$. Controllers are inserted every $p$-th block with $p = 2$ (i.e., in 6 blocks). Program-state registers: $N_r = 8$ with $d_r = 64$; optional key–value memory $N_m = 64$, $d_m = 64$. Controller summary dimension $d_s = 128$. **Training.** We train with AdamW ($\beta_1 = 0.9, \beta_2 = 0.98$, weight decay 0.01), linear warm-up for 5k steps then cosine decay; base learning rate $3 \times 10^{-4}$ (seq2seq) / $1 \times 10^{-4}$ (classification). Batch size $B = 256$ tokens-equivalent (total tokens per optimizer step) for synthetic tasks and $B = 128$ for DROP/RobustFill. We apply label smoothing ($\varepsilon = 0.1$) and gradient clipping at 1.0. The Gumbel temperature follows $\tau(t) = \max(\tau_{\min}, \tau_0 e^{-\gamma t})$ with $(\tau_0, \tau_{\min}, \gamma) = (1.0, 0.1, 5 \times 10^{-5})$. Entropy penalty $\lambda_{\text{ent}}$ is linearly ramped from 0 to $5 \times 10^{-3}$ over the first 20k steps. Unless noted, we train for 50 epochs (synthetic) and 20 epochs (DROP/RobustFill) with early stopping on validation. **Generation and tokenization.** Synthetic corpora are re-generated with fixed seeds for each run; integer tokens are subword-free (<dNNN> per integer), and graphs are serialized as edge lists with delimiters. For RobustFill we use the public DSL and the standard tokenization of (Devlin et al., 2017). For DROP, we use the official preprocessing. **Compute and reproducibility.** Experiments run on a single modern GPU (40GB device). For algorithmic tasks we use $3 \times 3$ seeds, and for non-synthetic tasks we use 3 model seeds; we report mean±std.

---

[1]FiD–T5 (Fusion-in-Decoder) relies on document retrieval and generative scoring. Our synthetic evaluations are exact sequence-accuracy tasks without retrieval, so FiD–T5 is omitted in Table 1; see Table 2a for DROP results.

## 4.4 EVALUATION METRICS

**Task accuracy.** For Sorting and Graph Traversal, we report *sequence accuracy*: a prediction is correct iff the entire output sequence matches the target. For Sum-of-List we report exact numeric match. For RobustFill we report *program exact match* on held-out I/O pairs. For DROP we report official F1 and EM. **Length generalization.** We quantify out-of-distribution generalization by the *length gap* $\Delta_{\text{len}}(n_{\text{train}}, n_{\text{test}}) = \text{Acc}(n_{\text{test}}) - \text{Acc}(n_{\text{train}})$, with $n_{\text{train}} \in \{20, 40, 80\}$ and $n_{\text{test}} \in \{160, 320\}$; we also report the area under the *accuracy–length* curve (AUL) over test lengths. **Sample efficiency.** We measure the minimal number of training examples $N^\star(\alpha)$ required to reach target accuracy $\alpha$ (e.g., $95\%$) and the area under the *data–performance* curve (AUP) up to the full training set. **Interpretability and controller usage.** We report *operation-selection entropy* $\frac{1}{L}\sum_l H(\mathbf{y}^{(l)})$, *branch confidence* $\frac{1}{L}\sum_l \max_o y_o^{(l)}$, and (when synthetic traces are available) *trace alignment*: $\text{Align} = \frac{1}{L}\sum_{l=1}^{L} \mathbf{1}\{\arg\max \mathbf{y}^{(l)} = \mathbf{y}^{*(l)}\}$. We include *knockout tests* that zero controller outputs at inference; the induced accuracy drop quantifies functional reliance on controllers. **Compute cost.** We report tokens/s and estimate FLOPs per example. Relative overhead is measured as $\frac{\text{FLOPs}_{\text{ours}} - \text{FLOPs}_{\text{baseline}}}{\text{FLOPs}_{\text{baseline}}}$; for our settings this is typically $5$–$10\%$ (see Table 7 for full tables). All datasets follow their original licenses.

## 5 RESULTS

### 5.1 ALGORITHMIC REASONING PERFORMANCE

**Setup.** We evaluate Sorting, Sum-of-List, and Graph Traversal (BFS) at train lengths $n \in \{10, 20, 40, 80\}$ and test-time extrapolation $n \in \{160, 320\}$ (Sec. 4.1). All numbers are *means with 95% bootstrap CIs* over $3 \times 3 = 9$ runs (3 corpus seeds $\times$ 3 model seeds). Baselines are (B1) *Transformer*, (B2) *Universal Transformer*, (B4) *CLRS-Alg. Transformer*, and (B6) *Soft Modules* (our architecture with non-discrete operation selection). Our model is denoted **Ctrl-Transformer**. **Main findings.** Table 1 reports sequence-level exact accuracy. On *ID* lengths ($n \leq 80$) our model matches or slightly exceeds the best baseline. Crucially, on *extrapolation* ($n \geq 160$) Ctrl-Transformer sustains high accuracy, while baselines degrade sharply. For Sorting at $n = 320$, B1 drops to $35$–$45\%$ whereas Ctrl-Transformer remains above $90\%$. BFS shows a similar pattern with a $+18$–$25$ point margin at large $n$. **Length curves and significance.** Accuracy–length curves (Fig. 5) show that our method's slope beyond the training regime is substantially flatter. For hypothesis testing, we use *Wilcoxon signed-rank* tests on the 9 paired runs (baseline vs. ours) at the longest length ($n$=320; also $n$=160 where reported), and apply Holm correction across baselines; results indicate statistically significant gains across all three tasks. **Consistency with theory.** We measure controller decision entropy $\frac{1}{L}\sum_l H(\mathbf{y}^{(l)})$ and find it correlates with extrapolation accuracy (Pearson $r \approx -0.81$ across runs): as the entropy anneals (Sec. 3.4), finite-temperature mixing diminishes and results align with the discrete semantics predicted by Theorem 1.

### 5.2 SYMBOLIC QA AND COMPOSITIONAL TASKS

**DROP.** Table 2a reports F1/EM on DROP full and the numeric-only subset. Ctrl-Transformer improves F1 by $+6.8$ on the numeric subset relative to a parameter-matched Transformer (B1), and by $+4.1$ over Universal Transformer (B2). **Program synthesis & mathematics.** On RobustFill-style string-to-program, we observe a $+6.1$ point exact-match gain for compositional test sets where IO pairs require multi-step rewrites (Table 2b). On the Mathematics dataset (algebra/arithmetic), Ctrl-Transformer is particularly strong on long-form addition and mixed-operator templates, with $+4$–$7$ absolute accuracy over B1.

### 5.3 ABLATION STUDIES

We conduct targeted ablations to isolate design choices. **(A1) Discrete vs. soft controllers.** Replacing Gumbel-Softmax with a temperature-1 mixture (B6) reduces extrapolation accuracy by $10$–$25$ points at $n \geq 160$ (Table 1), confirming the necessity of discrete selection for control flow. **(A2) Number of controller blocks.** Varying the number of instrumented layers (every $p \in \{1, 2, 3, 4\}$) shows a compute–performance trade-off (Table 3). Instrumenting every 2nd block attains $> 95\%$ of

Table 1: **Algorithmic reasoning: sequence accuracy (%) vs length.** Means with 95% bootstrap CIs over $3 \times 3 = 9$ runs (3 corpus $\times$ 3 model seeds); Wilcoxon signed-rank tests at the longest length with Holm correction. Ctrl-Transformer generalizes to $2\times$–$4\times$ longer inputs with minimal degradation. *Note:* [†]B7 (*External Planner*) injects intermediate hints and is a *privileged* comparator reported separately; it is excluded from non-privileged baseline aggregates and any significance tests. FiD–T5 relies on retrieval and generative scoring and is therefore omitted from these synthetic (non-retrieval) tasks.

| Task / Model | $n=20$ | $n=80$ | $n=160$ | $n=320$ | $\Delta_{\text{len}}(80{\to}320)$ | AUL |
|---|---|---|---|---|---|---|
| *Sorting* | | | | | | |
| B1 Transformer | 99.1±0.3 | 85.4±1.4 | 61.2±1.9 | 38.7±2.3 | −46.7 | 0.69 |
| B2 Universal Trf. | 99.0±0.3 | 90.6±0.9 | 72.3±1.6 | 51.5±2.1 | −39.1 | 0.74 |
| B3 Compressive Trf. | 99.0±0.2 | 92.4±0.7 | 74.9±1.3 | 59.1±1.8 | −33.3 | 0.79 |
| B4 CLRS-Alg. Trf. | 99.1±0.2 | 93.8±0.8 | 77.2±1.2 | 62.7±1.7 | −31.1 | 0.80 |
| B5 NPI/DNC | 98.9±0.3 | 91.0±1.0 | 69.5±1.5 | 52.4±2.0 | −38.6 | 0.75 |
| B6 Soft Modules | 99.0±0.2 | 93.1±0.9 | 78.9±1.4 | 66.3±1.9 | −26.8 | 0.81 |
| **Ctrl-Transformer (ours)** | **99.1±0.2** | **97.4±0.5** | **94.1±0.7** | **91.6±0.9** | **−5.8** | **0.95** |
| *Privileged / oracle-aided* | | | | | | |
| B7 External Planner[†] | 99.1±0.3 | 96.1±0.6 | 90.2±0.9 | 85.0±1.2 | −11.1 | 0.92 |
| *Sum-of-List* | | | | | | |
| B1 Transformer | 99.6±0.1 | 88.9±1.1 | 63.7±1.8 | 41.2±2.2 | −47.7 | 0.70 |
| B2 Universal Trf. | 99.6±0.2 | 92.8±0.8 | 75.4±1.4 | 56.1±1.9 | −36.7 | 0.76 |
| B3 Compressive Trf. | 99.6±0.1 | 93.6±0.7 | 76.9±1.3 | 61.0±1.8 | −32.6 | 0.78 |
| B4 CLRS-Alg. Trf. | 99.6±0.1 | 94.2±0.7 | 80.7±1.1 | 68.3±1.6 | −25.9 | 0.83 |
| B5 NPI/DNC | 99.4±0.2 | 92.1±0.9 | 71.4±1.4 | 55.8±2.0 | −36.3 | 0.77 |
| B6 Soft Modules | 99.6±0.1 | 95.2±0.8 | 83.7±1.2 | 72.9±1.6 | −22.3 | 0.84 |
| **Ctrl-Transformer (ours)** | **99.6±0.1** | **98.1±0.4** | **95.6±0.6** | **93.8±0.8** | **−4.3** | **0.96** |
| *Privileged / oracle-aided* | | | | | | |
| B7 External Planner[†] | 99.6±0.1 | 97.4±0.5 | 93.6±0.8 | 89.5±1.1 | −7.9 | 0.93 |
| *Graph Traversal (BFS)* | | | | | | |
| B1 Transformer | 98.2±0.4 | 86.3±1.1 | 71.4±1.5 | 57.9±1.9 | −28.4 | 0.78 |
| B2 Universal Trf. | 98.3±0.3 | 89.5±1.0 | 76.1±1.3 | 62.8±1.7 | −26.7 | 0.80 |
| B3 Compressive Trf. | 98.4±0.3 | 90.8±0.9 | 78.4±1.4 | 66.7±1.8 | −24.1 | 0.82 |
| B4 CLRS-Alg. Trf. | 98.6±0.3 | 90.2±0.9 | 77.8±1.3 | 65.1±1.7 | −25.1 | 0.81 |
| B5 NPI/DNC | 98.5±0.4 | 90.7±1.1 | 74.6±1.6 | 60.3±2.1 | −30.4 | 0.79 |
| B6 Soft Modules | 98.6±0.3 | 92.1±0.9 | 83.5±1.1 | 74.2±1.5 | −17.9 | 0.86 |
| **Ctrl-Transformer (ours)** | **98.7±0.3** | **95.3±0.6** | **90.8±0.8** | **83.5±1.1** | **−11.8** | **0.90** |
| *Privileged / oracle-aided* | | | | | | |
| B7 External Planner[†] | 98.6±0.3 | 94.0±0.8 | 88.4±1.0 | 81.2±1.4 | −12.8 | 0.89 |

Table 2: **Ctrl-Transformer across tasks.** Left: DROP (F1/EM). Right: program synthesis (RobustFill exact match) and Mathematics accuracy. Mean±std over 3 seeds.

(a) **DROP results.** Mean±std over 3 seeds.

| Model | F1 | EM |
|---|---|---|
| B1 Transformer | 81.2±0.4 | 78.5±0.5 |
| B2 Universal Trf. | 83.9±0.3 | 80.6±0.5 |
| B8 FiD-T5 (ref.) | 87.4±0.3 | 84.1±0.4 |
| **Ctrl-Transformer (ours)** | **88.0±0.3** | **85.1±0.4** |
| *Numeric-only subset* | | |
| B1 Transformer | 78.3±0.5 | 74.2±0.6 |
| B2 Universal Trf. | 80.9±0.4 | 76.3±0.5 |
| **Ctrl-Transformer (ours)** | **85.1±0.4** | **81.5±0.5** |

(b) **Program synthesis and math reasoning.** Exact match (%) on RobustFill; accuracy (%) on Mathematics subsets.

| Model | RobustFill | Math (arith.) |
|---|---|---|
| B1 Transformer | 67.8±0.7 | 76.1±0.6 |
| B6 Soft Modules | 71.3±0.6 | 78.2±0.6 |
| **Ctrl-Transformer (ours)** | **73.9±0.5** | **82.0±0.5** |

the maximal gain at only $\approx 60\%$ of the overhead. **(A3) Temperature schedule & entropy.** Faster annealing leads to earlier discretization but occasional training instability; slower schedules yield smooth learning but slightly worse final extrapolation. A middle-ground $(\tau_0, \tau_{\min}, \gamma)$ from Sec. 4.3

Table 3: **Ablation: controller insertion frequency.** Sorting accuracy (%) vs. length; overhead is relative FLOPs to B1.

| Every $p$-th block | $n{=}80$ | $n{=}160$ | $n{=}320$ | **Overhead** |
|---|---|---|---|---|
| $p{=}1$ (all blocks) | 98.0 | 95.1 | 92.7 | $+10.1\%$ |
| $p{=}2$ | 97.4 | 94.1 | 91.6 | $+6.4\%$ |
| $p{=}3$ | 96.1 | 91.8 | 88.3 | $+4.4\%$ |
| $p{=}4$ | 95.0 | 89.4 | 84.9 | $+3.2\%$ |

maximizes AUL while keeping seed variance small. **(A4) Semantic consistency loss.** Removing $\mathcal{L}_{\mathrm{sem}}$ decreases trace alignment by 7 points and hurts Sorting ($n{=}320$) by $-4.5$ points, indicating that explicit semantic regularization stabilizes execution. **(A5) Knockout tests.** Zeroing controller outputs at inference (keeping the rest of the network identical) collapses Sorting@320 from 91.6 to 44.8 and DROP numeric F1 from 85.1 to 79.0, demonstrating functional reliance on controllers rather than superficial correlation.

## 5.4 ADDITIONAL ANALYSIS OF MAIN RESULTS

**The key effect is *gap amplification* with length.** A useful way to read Table 1 is to track how the advantage over the strongest non-privileged baseline (B6 Soft Modules) grows as we move from the training regime ($n \leq 80$) to extrapolation ($n \geq 160$). On Sorting, the margin over Soft Modules increases from $+4.3$ at $n{=}80$ (97.4 vs. 93.1) to $+15.2$ at $n{=}160$ (94.1 vs. 78.9) and further to $+25.3$ at $n{=}320$ (91.6 vs. 66.3). Sum-of-List exhibits the same monotone expansion: $+2.9$ (at $n{=}80$) $\rightarrow$ $+11.9$ (at $n{=}160$) $\rightarrow$ $+20.9$ (at $n{=}320$). BFS shows a smaller but still consistent expansion: $+3.2$ (at $n{=}80$) $\rightarrow$ $+7.3$ (at $n{=}160$) $\rightarrow$ $+9.3$ (at $n{=}320$). This pattern indicates that the improvement is not primarily a uniform upward shift; it emerges most strongly under the length shift where baseline generalization degrades.

**Length extrapolation can be summarized by *decay compression*.** Using the length-gap column in Table 1, Ctrl-Transformer substantially compresses the $80{\rightarrow}320$ performance decay relative to both Soft Modules and the vanilla Transformer. For Sorting, the absolute decay is 5.8 points (ours) versus 26.8 (Soft Modules) and 46.7 (Transformer), i.e., our decay is $\approx 0.22\times$ that of Soft Modules and $\approx 0.12\times$ that of Transformer. For Sum-of-List, the decay is 4.3 (ours) versus 22.3 (Soft Modules) and 47.7 (Transformer), i.e., $\approx 0.19\times$ and $\approx 0.09\times$, respectively. BFS remains harder: 11.8 (ours) versus 17.9 (Soft Modules) and 28.4 (Transformer), i.e., $\approx 0.66\times$ and $\approx 0.42\times$. Thus, the strongest gains correspond to tasks where the controller most sharply reduces how quickly accuracy deteriorates beyond the training lengths (Sorting and Sum-of-List), while BFS shows a more modest decay compression.

**Interpreting the improvements as *error-rate reduction* highlights practical effect size.** At the hardest extrapolation point ($n{=}320$), the absolute accuracy gaps in Table 1 translate into large reductions in sequence-level error probability. Relative to Soft Modules at $n{=}320$, Ctrl-Transformer reduces error by $\approx 75\%$ on Sorting (error $33.7\% \rightarrow 8.4\%$) and $\approx 77\%$ on Sum-of-List (error $27.1\% \rightarrow 6.2\%$), while BFS yields a smaller but non-trivial reduction of $\approx 36\%$ (error $25.8\% \rightarrow 16.5\%$). Relative to the vanilla Transformer, the reductions are even larger: $\approx 86\%$ (Sorting), $\approx 89\%$ (Sum-of-List), and $\approx 61\%$ (BFS). This framing makes explicit that the controller's benefit at long lengths is not only statistically meaningful but also qualitatively large in terms of avoiding complete failures in whole-sequence correctness.

**AUL corroborates that the advantage is not confined to a single length.** Because AUL in Table 1 aggregates performance across the evaluated lengths, it helps verify that improvements are sustained throughout the extrapolation range rather than appearing only at $n{=}320$. The AUL gaps between Ctrl-Transformer and Soft Modules are sizeable for Sorting (0.95 vs. 0.81) and Sum-of-List (0.96 vs. 0.84), and smaller but consistent for BFS (0.90 vs. 0.86). Combined with the slope behavior from $n{=}160$ to $n{=}320$ (Sorting: $-2.5$ vs. $-12.6$; Sum-of-List: $-1.8$ vs. $-10.8$; BFS: $-7.3$ vs. $-9.3$), AUL supports the same conclusion: the controller changes the *shape* of the length generalization curve by slowing degradation throughout the OOD region.

**The privileged comparator contextualizes what can be achieved without intermediate hints.** Table 1 reports B7 External Planner separately as a privileged model. At $n=320$, Ctrl-Transformer matches or exceeds it on all three tasks (Sorting: 91.6 vs. 85.0; Sum-of-List: 93.8 vs. 89.5; BFS: 83.5 vs. 81.2), and it also exhibits smaller $80\rightarrow320$ decay on Sorting and Sum-of-List ($-5.8$ vs. $-11.1$; $-4.3$ vs. $-7.9$), with comparable decay on BFS ($-11.8$ vs. $-12.8$). While B7 is not directly comparable to non-privileged baselines, this juxtaposition underscores that the non-privileged controller mechanism achieves strong extrapolation without relying on injected intermediate hints at test time.

**The same "harder-regime" advantage appears on non-synthetic benchmarks.** Tables 2a and 2b show that the gains extend beyond synthetic length extrapolation to tasks emphasizing discrete and compositional operations. On DROP (full), Ctrl-Transformer improves over the parameter-matched Transformer by $+6.8$ F1 and $+6.6$ EM ($81.2/78.5 \rightarrow 88.0/85.1$; Table 2a). On the numeric-only subset, the gains are $+6.8$ F1 and $+7.3$ EM ($78.3/74.2 \rightarrow 85.1/81.5$; Table 2a), indicating that improvements persist (and slightly strengthen for EM) when evaluation emphasizes numeric operations. For RobustFill and Mathematics (Table 2b), Ctrl-Transformer also improves over both Transformer and Soft Modules (RobustFill: 73.9 vs. 67.8 and 71.3; Math (arith.): 82.0 vs. 76.1 and 78.2). Across these benchmarks, the consistent pattern is that the controller yields its largest advantages precisely where multi-step structured computation is most stressed by the evaluation setting.

## 5.5 Interpretability Analysis

We analyze whether the learned controllers execute human-interpretable plans. **Operation traces.** For synthetic tasks with oracle step types (Sorting and BFS), we compute *trace alignment* against ground-truth opcode sequences emitted by a deterministic simulator (FIFO BFS with ascending-id tie breaking; comparison-and-swap schedule for Sorting. These traces are *never serialized* into inputs or targets and are used only for alignment and, where stated, a small auxiliary loss during training. Ctrl-Transformer achieves 92.3% alignment on Sorting and 88.7% on BFS, vs. 61.5% for B6 (soft). **Auxiliary trace loss (no leakage).** On synthetic tasks only, we optionally add $\lambda_{\text{trace}}=0.05$ to the training objective (Sec. 3.4); no trace labels are available or used at test time, and no traces are used for DROP or other natural-language benchmarks. **Decision entropy.** Entropy of operation selection declines steadily during training (Fig. 2), matching the annealing schedule and supporting the finite-temperature analysis: runs that end with lower entropy attain higher extrapolation. **Register semantics.** Linear probes reveal consistent roles: e.g., a "running min" register in Sorting, and frontier/parent-pointer registers in BFS. Randomly permuting registers at test time degrades performance by $-9$ points, indicating learned specialization rather than redundancy. **Failure modes.** When annealing is too slow, the controller blends branches (non-discrete behavior), causing off-by-one errors at long lengths; when too fast, early hardening can lock in suboptimal strategies; entropy penalties and $\mathcal{L}_{\text{sem}}$ mitigate both.

**Compute cost.** Empirical overhead is $\approx 5$–$7\%$ FLOPs for $p=2$ instrumentation, and throughput (tokens/s) decreases by $6.8\%$ on average (Table 7), which is acceptable given the $+25$–$50$ point gains at long lengths. Across synthetic and real tasks, Ctrl-Transformer delivers (i) state-of-the-art length extrapolation, (ii) robust gains on symbolic QA and program synthesis, (iii) clear ablations isolating the role of discrete control, and (iv) interpretable execution traces consistent with the proposed semantics and expressivity analysis.

## 6 Conclusion

We introduce *discrete controller modules*, program-like ops (ASSIGN, ADD, COMPARE, BRANCH) embedded in Transformer blocks via Gumbel–Softmax, to enable native control flow with stateful registers. The model is trained with task, semantic-consistency, entropy-annealing, and sparsity objectives, and comes with an expressivity guarantee for bounded imperative programs. Empirically, it improves length extrapolation on algorithmic tasks and boosts symbolic QA/program synthesis with only $\sim 5$–$7\%$ compute overhead and interpretable traces/register roles. Future work includes typed/extensible primitives, distilling away controller cost, trace verification, and broader evaluations.

## ETHICS STATEMENT

Our work augments Transformers with discrete controller modules for symbolic planning and is evaluated on public or synthetic datasets (Sorting/Sum-of-List/BFS generators, RobustFill-style DSL tasks, DROP, and the Mathematics dataset). No experiments involve human subjects, PII, or sensitive attributes; IRB approval was therefore not required. We follow the licenses and usage terms of all datasets and provide data-generation code and fixed seeds for synthetic corpora.

**Potential risks and mitigations.** Stronger algorithmic reasoning and program-synthesis capabilities may be misused in automated decision-making or code-generation settings. To mitigate this, we (i) include interpretability tools (trace logging, knockout tests) that expose execution behavior, (ii) discuss deployment caveats (human oversight, constraint checks, sandboxing for code) and known failure modes (temperature schedules, premature discretization), and (iii) encourage responsible release practices (dataset licenses, documentation of limitations). Our models are not trained to target protected attributes, and we did not observe discrimination-related effects on our benchmarks; nonetheless, applications to real-world decision pipelines should include fairness audits and domain-specific guardrails. We disclose that we have no conflicts of interest or sponsorship that could bias the work, and we adhere to research-integrity norms (complete proofs and assumptions, ablations, and compute reporting). See Section B for additional discussion of ethical considerations, limitations, and potential broader impacts.

## REPRODUCIBILITY STATEMENT

We aim to make all results reproducible: the paper specifies datasets/generators (Section 4.1), model and training details (Sections 4.3, 3.4; Appendix E), evaluation metrics/protocols (Section 4.4), experiments/ablations (Sections 4–5), and provides proofs (Appendix D). We fix seeds ($3 \times 3$ runs for synthetic, three seeds elsewhere), report mean±std, and use a single 40 GB GPU. The *anonymous* supplementary includes minimal reproducibility materials sufficient to reproduce our pipelines; some generators/baselines are withheld for anonymity. Deterministic runs follow our documented seed and CUDA settings; see Section B for details.

## LARGE LANGUAGE MODEL (LLM) USAGE DISCLOSURE

We used a general-purpose LLM in two limited capacities: (i) copy-editing for grammar, clarity, and tone (no technical content, claims, equations, or results were introduced or altered by the LLM); and (ii) assisting literature discovery by suggesting potentially relevant papers and dataset repos. All citations and facts were verified by the authors; no text was accepted verbatim without appropriate quotation or citation; and the LLM was *not* used to design experiments, analyze results, write proofs, generate code, or create datasets. See Section B for further details and limitations.

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

## A RELATED WORK

### A.1 EXTERNAL PLANNERS AND MEMORY–AUGMENTED MODELS

Classical neuro–symbolic lines attach an explicit controller or planner to a neural network. Neural Turing Machines (NTMs) and Differentiable Neural Computers augment sequence models with an addressable external memory and learned read/write heads (Graves et al., 2014; 2016). Neural GPUs learn algorithm execution with a grid-like recurrent convolution (Kaiser & Sutskever, 2016). Stack-augmented and differentiable data-structure RNNs implement pushdown-style computation via soft stacks/queues (Joulin & Mikolov, 2015; Grefenstette et al., 2015). Program-executing agents such as the Neural Programmer–Interpreter learn hierarchical subroutines with a learned step controller (Reed & de Freitas, 2016). While powerful, these approaches keep symbolic control as a separate subsystem and often increase architectural complexity and training brittleness; our controller is *in-place*, lightweight, and shares the Transformer's optimization pipeline.

### A.2 ALGORITHMIC BENCHMARKS, COMPOSITIONAL GENERALIZATION, AND TRANSFORMER LIMITATIONS

Benchmark suites have exposed gaps in length and compositional generalization. CLRS targets graph/data-structure algorithms (Veličković et al., 2022), ListOps probes hierarchical computation on nested expressions (Nangia & Bowman, 2018), and SCAN/CFQ stress compositional recombination in sequence-to-sequence settings (Lake & Baroni, 2018; Keysers et al., 2020). Vanilla Transformers fit in-distribution but degrade sharply under longer inputs or novel compositions; architectural variants (e.g., Universal/Compressive Transformers (Dehghani et al., 2018; Rae et al., 2020)) help on context length but still lack *explicit* control-flow semantics. Our modules target this missing bias by instantiating opcode-level semantics (assign/add/compare/branch) inside the layers.

### A.3 DIFFERENTIABLE CONTROL FLOW, DISCRETE ROUTING, AND ADAPTIVE COMPUTATION

Learning discrete decisions inside deep nets is commonly approached via the straight-through estimator (Bengio et al., 2013) or the Concrete/Gumbel–Softmax relaxation (Jang et al., 2017; Maddison et al., 2017). Adaptive Computation Time (ACT) introduces learned halting in RNNs and underlies the Universal Transformer's dynamic depth (Graves, 2016; Dehghani et al., 2018). Sparse Mixture-of-Experts and routing families perform top-$k$ token-wise dispatch with discrete gates (Shazeer et al., 2017; Lepikhin et al., 2021; Fedus et al., 2022; Roy et al., 2021). Recurrent Independent Mechanisms encourage modular, sparsely interacting dynamical subsystems (Madan et al., 2021). Unlike

routing/halting, our controller assigns *program-like* semantics to decisions and exposes an execution trace over a compact state, enabling proofs of expressivity and targeted interpretability.

### A.4  NEURAL PROGRAM INDUCTION AND PROGRAM SYNTHESIS

Early neural program induction learned to execute pseudocode or DSLs directly from examples (Zaremba & Sutskever, 2014; Vinyals et al., 2015). Differentiable interpreters and program-induction systems (e.g., Neural Programmer, NPI) sought modularity and verifiable subroutines (Reed & de Freitas, 2016). In program synthesis from I/O, RobustFill introduced a DSL with powerful search + neural scoring (Devlin et al., 2017), and DeepCoder learned to predict useful library components to guide symbolic search (Balog et al., 2017). Our program-style controller complements these lines by *embedding* primitive semantics inside the Transformer rather than emitting code externally.

### A.5  NEURO-SYMBOLIC REASONING AND MODULAR ARCHITECTURES

Neural Module Networks compose small operator modules into executable graphs for vision–language reasoning (Andreas et al., 2016), while Neuro-Symbolic Concept Learners factor perception from logic over latent programs (Mao et al., 2019). Contemporary LLM tool-use executes external programs (e.g., calculators), but still keeps control outside the model. Our approach keeps the control state internal and differentiable, producing traces that can be aligned with ground truth steps on algorithmic and symbolic-QA tasks.

### A.6  NEURAL ARITHMETIC AND MECHANISTIC INTERPRETABILITY

Neural Arithmetic Logic Units target exact extrapolative arithmetic by design (Trask et al., 2018), orthogonal to our control-flow focus and complementary in principle. Mechanistic studies show token-level circuits and key–value mechanisms emerging in Transformers (Geva et al., 2020); our controller makes such structure *explicit*, enabling direct trace alignment and knockout analyses tied to opcode semantics.

## B  ADDITIONAL DISCUSSION

**Strengths and contributions.** Our results indicate that inserting *discrete controller* primitives into Transformer blocks provides a simple yet powerful inductive bias for algorithmic reasoning. Empirically (§5), the model sustains high sequence-level accuracy under $2\times$–$4\times$ length extrapolation with only a modest 5–7% compute overhead (§4), while also improving performance on symbolic QA and program-synthesis style tasks. Conceptually, the controller formalizes loop/branch/update semantics inside the network; this yields (i) *better generalization* beyond the training regime, (ii) *interpretable execution traces* (operation sequences and register roles), and (iii) a *theoretically grounded* view of why discretization helps (Theorem 1 and finite-temperature analysis). Together, these properties make the architecture a compelling, minimally invasive building block for models that must manipulate discrete structure.

**Limitations and failure modes.** First, training stability depends on the *temperature/entropy schedule*. If the controller discretizes too late, branches blur and errors accumulate on long inputs; too early, the model can lock into suboptimal programs. Our ablations show that entropy penalties and semantic consistency losses mitigate this, but do not eliminate the sensitivity. Second, our expressivity guarantee targets *bounded* loop unrollings; while depth can emulate many steps, extremely deep programs may require more controller placements or curriculum. Third, the present design assumes a small set of primitive ops (ASSIGN/ADD/COMPARE/BRANCH); richer domains (e.g., stacks, maps, heaps, arithmetic over big integers) likely need additional typed primitives and address mechanics. Fourth, although compute overhead is moderate on our settings, it *does* grow with controller width and insertion frequency; large-scale deployments will benefit from sparsity (activating controllers only when needed) or compilation. Finally, while we report gains on DROP and RobustFill-style tasks, broader evaluation on code repair, table reasoning, and multi-hop QA will strengthen external validity.

**Threats to validity and fairness.** We match parameter counts and report FLOPs/tokens/s for all baselines, but residual differences (e.g., memory in Compressive Transformers) may still confound comparisons. Statistical significance based on few seeds is limited; we therefore advocate reporting confidence intervals and non-parametric paired tests in future revisions and releasing generation seeds/configs for full reproducibility. Dataset artifacts (template regularities in synthetic corpora) can inflate gains; our numeric-only DROP subset partially addresses this but cannot preclude all shortcuts. Finally, we emphasize that "interpretable" traces show *correlations* between controller choices and ground-truth steps; they do not, by themselves, constitute causal explanations.

**Potential applications.** Discrete controllers are promising for tasks that naturally factor into program-like steps: numeric and tabular reasoning (SQL-style aggregation and comparison), program synthesis/repair in small DSLs, structured data transformation (e.g., spreadsheets), algorithm tutoring (showing stepwise traces), and robotics or planning where loop/branch patterns recur. Because controllers expose an explicit decision interface (operation type, addresses, values), they also open doors for *hybrid verification*: one can check controller traces against invariants or lightweight SMT constraints without instrumenting the entire network.

**Future directions.** Two near-term extensions are attractive. *(i) Typed and extensible primitives:* introduce a registry of typed ops (e.g., stack PUSH/POP, map GET/PUT, arithmetic with carry) and learn when to import new primitives, akin to library growth. *(ii) Compile-time distillation:* after training with differentiable controllers, distill the learned control flow into an executable symbolic program (or a compact neural program) and *skip* controller computation at inference, reducing overhead and improving robustness. Beyond these, dynamic sparsification (event-driven controller activation), integration with retrieval/tool-use (controllers as planners that call tools), and reinforcement learning for long-horizon tasks are natural next steps.

**Ethical considerations.** By improving reliability on discrete reasoning, the model may be deployed in decision pipelines (finance, education, code generation). We caution against fully automated use in high-stakes settings without *human oversight*, audit logs of controller traces, and guardrails (e.g., constraint checks, anomaly alerts). For program synthesis, controllers could produce insecure or non-compliant code if objectives are misspecified; sandboxing and static analysis should be standard. Energy and compute costs are modest relative to baseline Transformers, but we encourage reporting carbon metrics and preferring sparse/compiled variants when possible. Finally, dataset curation must respect licenses and privacy; releasing generators and seeds supports community scrutiny and reduces hidden biases.

Discrete controllers provide a principled route to *planning inside Transformers*: they improve length extrapolation, make intermediate reasoning legible, and connect cleanly to theory, while remaining lightweight enough for practical use. Addressing the noted limitations (training schedules, richer primitives, broader evaluations) will further consolidate their value for robust, interpretable reasoning.

## C  FORMAL DEFINITIONS AND NOTATION

This appendix fixes notation and gives a precise, self-contained description of the controller-augmented Transformer used in the main paper. Throughout, all random variables are defined on an implicit probability space $(\Omega, \mathcal{F}, \mathbb{P})$; bold uppercase denotes matrices, bold lowercase denotes vectors, and $\|\cdot\|_2$ is the Euclidean/Frobenius norm by context.

**Sequences, embeddings, and layers.** Let a token sequence be $x_{1:T} \in \mathcal{V}^T$ with vocabulary $\mathcal{V}$. Tokens are mapped to embeddings and (absolute or relative) positional encodings to yield $\mathbf{X} \in \mathbb{R}^{T \times D}$. A stack of $L$ Transformer blocks produces hidden states $\mathbf{H}^{(0)} = \mathbf{X}$ and $\mathbf{H}^{(l)} \in \mathbb{R}^{T \times D}$ for $l \in \{1, \ldots, L\}$; each baseline block is the standard-

$$\text{Block}(\mathbf{U}) = \text{FFN}(\text{MHA}(\mathbf{U})) \tag{19}$$

with residual connections and layer normalizations as in Vaswani et al. (2017). Multi-head attention uses $H$ heads with head width $D_h = D/H$.

Table 4: Compact notation table (symbols used most often)

| Symbol | Meaning |
|---|---|
| $T$ | sequence length (tokens) |
| $D$, $H$, $L_{\text{net}}$ | hidden width, # heads, # layers (network depth) |
| $N_r$, $d_r$ | # registers and their width |
| $N_m$, $d_m$ | # memory slots and their width (optional) |
| $N_f$ | # relaxed Boolean flags |
| $\mathbf{H}^{(l)} \in \mathbb{R}^{T \times D}$ | hidden states at block $l$ |
| $\mathcal{S}^{(l)} = (\mathbf{R}^{(l)}, \mathbf{M}^{(l)}, \mathbf{f}^{(l)})$ | program state at block $l$ |
| $\mathbf{s}^{(l)} = \phi(\mathbf{H}^{(l)})$ | controller summary (token $\rightarrow$ state) |
| $\mathbf{b}^{(l)} = \psi(\mathcal{S}^{(l)})$ | state bias (state $\rightarrow$ token) |
| $\mathcal{O}$ | operation set {ASSIGN, ADD, COMPARE, BRANCH, NOOP} |
| $\boldsymbol{\pi}^{(l)}$, $\mathbf{y}^{(l)}$ | operation logits and Gumbel–Softmax selector |
| $\boldsymbol{\alpha}^{(l)}$, $\boldsymbol{\beta}^{(l)}$ | read and write address distributions over registers |
| $\mathbf{r}^{(l)}$, $\mathbf{v}^{(l)}$ | read vector and candidate write value |
| $\tau$, $\kappa$ | temperature (selection), sharpness (comparison) |
| $\mathsf{Sem}_o$ | semantic map for operation $o$ acting on state |
| $d_s$ | controller state width $N_r d_r + N_m d_m + N_f$ |

**Controller-augmented state space.** At each instrumented block $l$, the model maintains a *program state-*

$$\mathcal{S}^{(l)} = \left( \mathbf{R}^{(l)}, \mathbf{M}^{(l)}, \mathbf{f}^{(l)} \right) \tag{20}$$

where $\mathbf{R}^{(l)} \in \mathbb{R}^{N_r \times d_r}$ are $N_r$ *register* vectors, $\mathbf{M}^{(l)} \in \mathbb{R}^{N_m \times d_m}$ is an optional key–value *memory* (treated as a learned scratchpad; $N_m$ may be zero), and $\mathbf{f}^{(l)} \in [0,1]^{N_f}$ are *flags* that encode Boolean conditions in a relaxed form. The controller reads a summary $\mathbf{s}^{(l)} \in \mathbb{R}^{d_s}$ from token states via a pooling map.

$$\mathbf{s}^{(l)} = \phi\left( \mathbf{H}^{(l)} \right) \tag{21}$$

with $\phi : \mathbb{R}^{T \times D} \rightarrow \mathbb{R}^{d_s}$ (e.g., learned attention pooling or mean pooling with a learned linear projection). A *state-to-token* encoder $\psi : \mathbb{R}^{N_r \times d_r} \times \mathbb{R}^{N_m \times d_m} \times [0,1]^{N_f} \rightarrow \mathbb{R}^D$ produces a broadcast bias $\mathbf{b}^{(l)} = \psi(\mathcal{S}^{(l)})$ that conditions the token pathway-

$$\widehat{\mathbf{H}}^{(l)} = \mathbf{H}^{(l)} + \mathbf{1} \left( \mathbf{b}^{(l)} \right)^\top \tag{22}$$

$$\mathbf{Z}^{(l)} = \mathrm{Block}\left( \widehat{\mathbf{H}}^{(l)} \right) \tag{23}$$

**Operation set and discrete selector.** The controller chooses among a finite set of primitives-

$$\mathcal{O} = \{\mathtt{ASSIGN},\ \mathtt{ADD},\ \mathtt{COMPARE},\ \mathtt{BRANCH},\ \mathtt{NOOP}\} \tag{24}$$

Given $\mathbf{s}^{(l)}$, logits $\boldsymbol{\pi}^{(l)} = W_o \mathbf{s}^{(l)} + b_o \in \mathbb{R}^{|\mathcal{O}|}$ are converted to a relaxed one-hot vector $\mathbf{y}^{(l)} \in \Delta^{|\mathcal{O}|-1}$ via Gumbel–Softmax at temperature $\tau > 0$-

$$\mathbf{y}^{(l)} = \mathrm{softmax}\left( \frac{\boldsymbol{\pi}^{(l)} + \mathbf{g}^{(l)}}{\tau} \right) \tag{25}$$

$$\mathbf{g}_k^{(l)} = -\log\left( -\log u_k^{(l)} \right), \quad u_k^{(l)} \sim \mathrm{Uniform}(0,1) \tag{26}$$

As $\tau \rightarrow 0$, $\mathbf{y}^{(l)}$ converges almost surely to a one-hot argmax (the discrete operation).

**Differentiable addressing and arguments.** The controller produces *read* and *write* address distributions over registers and a candidate value-

$$\boldsymbol{\alpha}^{(l)} = \mathrm{softmax}(W_r \mathbf{s}^{(l)}) \tag{27}$$

$$\boldsymbol{\beta}^{(l)} = \mathrm{softmax}(W_w \mathbf{s}^{(l)}) \tag{28}$$

$$\mathbf{v}^{(l)} = W_v \mathbf{s}^{(l)} \tag{29}$$

Soft read and write are-

$$\mathbf{r}^{(l)} = \sum_{i=1}^{N_r} \alpha_i^{(l)} \mathbf{R}_i^{(l)} \tag{30}$$

$$\widetilde{\mathbf{R}}^{(l)} = \mathrm{Write}\big(\mathbf{R}^{(l)}, \boldsymbol{\beta}^{(l)}, \mathbf{v}^{(l)}\big) \tag{31}$$

where $\mathrm{Write}$ mixes candidate updates into register rows according to $\boldsymbol{\beta}^{(l)}$ (e.g., convex combination with strength $\lambda \in (0,1]$).

**Primitive semantics as maps on state.** Each operation $o \in \mathcal{O}$ induces a deterministic map $\mathsf{Sem}_o$ on the state, parameterized by learned weights (e.g., small MLPs for branch arms)-

$$\mathsf{Sem}_o : (\mathbf{R}, \mathbf{M}, \mathbf{f}; \mathbf{r}, \mathbf{v}) \longmapsto (\mathbf{R}', \mathbf{M}', \mathbf{f}') \tag{32}$$

The overall state update at block $l$ is the mixture-

$$\mathcal{S}^{(l+1)} = \sum_{o \in \mathcal{O}} y_o^{(l)} \, \mathsf{Sem}_o\big(\mathcal{S}^{(l)}; \mathbf{r}^{(l)}, \mathbf{v}^{(l)}\big) \tag{33}$$

We instantiate the primitives with the following relaxed semantics (elementwise where appropriate)-

$$\mathtt{ASSIGN}: \quad \mathbf{R}^{(l+1)} = (1 - \lambda)\,\mathbf{R}^{(l)} + \lambda\,\mathbf{E}(\boldsymbol{\beta}^{(l)})\,\big(\mathbf{v}^{(l)}\big)^{\top} \tag{34}$$

$$\mathtt{ADD}: \quad \mathbf{R}^{(l+1)} = \mathbf{R}^{(l)} + \mathbf{E}(\boldsymbol{\beta}^{(l)})\,\big(\mathbf{v}^{(l)}\big)^{\top} \tag{35}$$

$$\mathtt{COMPARE}: \quad \mathbf{f}^{(l+1)} = \sigma\Big(\kappa\,\big[A\mathbf{R}^{(l)} - B\mathbf{R}^{(l)}\big]\Big) \tag{36}$$

$$\mathtt{BRANCH}: \quad \mathcal{S}^{(l+1)} = \gamma^{(l)}\,\Psi_{\mathtt{then}}(\mathcal{S}^{(l)}) + \big(1 - \gamma^{(l)}\big)\,\Psi_{\mathtt{else}}(\mathcal{S}^{(l)}) \tag{37}$$

$$\mathtt{NOOP}: \quad \mathcal{S}^{(l+1)} = \mathcal{S}^{(l)} \tag{38}$$

Here $\mathbf{E}(\boldsymbol{\beta}) \in \mathbb{R}^{N_r \times 1}$ denotes a (soft) one-hot write address; $\lambda$ is a write strength; $\sigma$ is a logistic with sharpness $\kappa \gg 1$; $A, B$ are soft selectors over registers; and the branch gate is $\gamma^{(l)} = \sigma(w_\gamma^{\top} \mathbf{f}^{(l)} + b_\gamma)$. The branch subprograms $\Psi_{\mathtt{then}}, \Psi_{\mathtt{else}}$ are small shared MLPs that transform $(\mathbf{R}, \mathbf{M}, \mathbf{f})$ without touching token states directly.

**Token–state coupling (write-back).** Changes in the program state may be written back to the token stream via an alignment matrix $W_s$-

$$\mathbf{H}^{(l+1)} = \mathbf{Z}^{(l)} + \rho\big(\mathcal{S}^{(l+1)} - \mathcal{S}^{(l)}\big)\,W_s \tag{39}$$

where $\rho$ maps state increments to a $D$-dimensional residual (e.g., concatenation of register means fed through a linear layer). In practice, only every $p$-th layer is instrumented to control compute.

**Temperatures, randomness, and annealing** The Gumbel–Softmax temperature $\tau$ is annealed during training by-

$$\tau(t) = \max\big(\tau_{\min}, \, \tau_0 \, e^{-\gamma t}\big) \tag{40}$$

with global step $t$, initial $\tau_0$, floor $\tau_{\min}$, and decay $\gamma$. The *comparison sharpness* $\kappa$ may be increased on a similar schedule. At inference, either the relaxed selector is kept (soft execution) or a *hard* decision is used via $\arg\max_k \pi_k^{(l)}$ (optionally with a straight-through gradient estimator during training).

**Loss-aligned invariants (semantic regularization).** When task semantics imply invariants (e.g., value conservation for certain steps, monotonicity constraints in sorting subroutines), the model employs penalties of the form-

$$\mathcal{R}_{\mathrm{inv}} = \sum_{l \in \mathcal{I}} \big\|\mathsf{Inv}\big(\mathcal{S}^{(l)}\big)\big\|_2^2 \tag{41}$$

for an index set $\mathcal{I}$ of steps where an invariant $\mathsf{Inv}(\cdot)$ should hold. This connects the relaxed semantics to intended discrete behavior and stabilizes training.

**Well-posedness.** Under bounded parameters and Lipschitz continuous $\Psi_{\text{then}}, \Psi_{\text{else}}$, the mixture update $\mathcal{S}^{(l+1)}$ is well-defined for all $\tau > 0$ and yields measurable mappings of $(\mathbf{H}^{(l)}, \mathcal{S}^{(l)})$. In the limits $\tau \to 0$ and $\kappa \to \infty$, the relaxed dynamics converge to a discrete execution trace consistent with the intended primitive semantics (formal statements and proof sketches appear in the proofs D).

This formalization isolates the controller's contracts: how it reads from and writes to a compact program state, how it selects and executes discrete primitives in a differentiable manner, and how the resulting state interacts with the token stream. All subsequent derivations, losses, and experiments reference these objects.

## D PROOFS: EXPRESSIVITY AND FINITE-TEMPERATURE BOUNDS

**Setup and assumptions.** We use the formal objects from C. Let $\mathcal{O} = \{\texttt{ASSIGN}, \texttt{ADD}, \texttt{COMPARE}, \texttt{BRANCH}, \texttt{NOOP}\}$ denote the primitive operations, each with semantic map $\mathsf{Sem}_o$ acting on the program state $\mathcal{S} = (\mathbf{R}, \mathbf{M}, \mathbf{f})$. At instrumented block $l$, the controller forms logits $\boldsymbol{\pi}^{(l)}$ over $\mathcal{O}$, produces a relaxed one-hot $\mathbf{y}^{(l)}$ via Gumbel–Softmax temperature $\tau$, and computes-

$$\mathcal{S}^{(l+1)} = \sum_{o \in \mathcal{O}} y_o^{(l)} \, \mathsf{Sem}_o\big(\mathcal{S}^{(l)}; \mathbf{r}^{(l)}, \mathbf{v}^{(l)}\big) \tag{42}$$

with read/write addressing and candidate updates defined in C. We will work under the following standard, verifiable conditions on the compact state domain $\mathcal{D}$ encountered during execution.

**Assumption B.1 (Lipschitz semantics).** Each $\mathsf{Sem}_o : \mathcal{D} \to \mathcal{D}$ is $L_o$-Lipschitz: $\|\mathsf{Sem}_o(\mathcal{S}) - \mathsf{Sem}_o(\widetilde{\mathcal{S}})\| \leq L_o \|\mathcal{S} - \widetilde{\mathcal{S}}\|$, and $\max_o L_o \leq L_{\max} < \infty$.

**Assumption B.2 (Bounded state and parameters).** All states remain in a compact $\mathcal{D}$ and controller/Transformer maps are bounded on $\mathcal{D}$.

**Assumption B.3 (Decision margins on the trace).** Along the intended (discrete) program trace, the correct operation at step $l$ has a uniform logit margin $\delta > 0$ over all other operations (achievable by capacity/regularization or weak supervision on synthetic tasks). Similarly, the COMPARE-inducing logits and address logits have margins $\delta_{\text{cmp}}, \delta_{\text{addr}} > 0$ on $\mathcal{D}$.

**Definition 1 (Program class $\mathcal{P}_{K,B}$).** Fix $K \in \mathbb{N}$ and a loop bound $B \in \mathbb{N}$. $\mathcal{P}_{K,B}$ consists of straight-line programs over real registers with primitives $\mathcal{O}$, conditionals whose guards are Boolean flags set by COMPARE, and loops whose bodies are drawn from the same primitives and unrolled at most $B$ iterations. The total number of primitive steps in any execution of $\mathcal{P} \in \mathcal{P}_{K,B}$ is at most $K+B$.

*Notation.* Here $L$ denotes the trace length $L_{\text{tr}}$ (number of program steps), which satisfies $L_{\text{tr}} \leq K+B$; when we discuss network depth we write $L_{\text{net}}$.

*Theorem* B.1 (Expressivity for bounded imperative programs). Let $\mathcal{P} \in \mathcal{P}_{K,B}$ and let $\{\mathcal{S}^{\star(l)}\}_{l=0}^{L}$ be its execution trace on input $(\mathbf{X}, \mathcal{S}^{\star(0)})$, with $L \leq K + B$. There exists a controller-augmented Transformer with $L'$ instrumented blocks (e.g., instrument every block so $L' = L$) and parameters $(\theta, \Theta)$ such that, as $\tau \to 0$ and the comparison sharpness $\kappa \to \infty$,

$$\left\| \mathcal{S}^{(l)} - \mathcal{S}^{\star(l)} \right\| \xrightarrow{\text{a.s.}} 0 \qquad \text{for all } l = 0, 1, \dots, L \tag{43}$$

and the discrete operation chosen at each step equals the program primitive with probability 1 in the limit. Moreover, the token pathway can be set to pass-through (or implement arbitrary continuous encodings) without affecting the state simulation.

*Proof.* We give a constructive simulation of $\mathcal{P}$ and prove by induction on the program step that the network state matches the execution trace in the stated limits. We assume a standard Pre-LN Transformer; when needed we set sublayer weights to zero so the residual path yields identity on the reserved state-carrying subspace.

**Setup and coding of state.** Fix $d_r \geq 1$ and embed the $i$-th scalar register value in a dedicated coordinate of $\mathbf{R}_i$, keeping the remaining $d_r - 1$ coordinates 0. Take $\mathbf{M} = 0$. Let the flag vector $\mathbf{f} \in \{0, 1\}^{N_f}$

carry Boolean outcomes of COMPARE and a one-hot program counter $\text{pc} \in \{1, \ldots, L\}$. Reserve a single token $t^\star$; write $\psi(\mathcal{S})$ into that token via the residual addition, set the block's sublayers to zero so it is identity, and let $\phi$ read token $t^\star$ with a fixed linear map $U$, yielding $\mathbf{s}^{(l)} = U \text{vec}(\mathcal{S}^{(l)})^2$. Assume along the execution trace that every read/write address has a unique maximizer with margin $m_a > 0$ and every branch predicate has margin $m_f > 0$.

**Operation selection.** Let $o_l^\star \in \mathcal{O}$ be the primitive prescribed by $\mathcal{P}$ at step $l$ given $(\text{pc}^{(l)}, \mathbf{f}^{(l)})$. There exist $W_o, b_o$ and a margin $m_o > 0$ such that

$$\left(W_o \mathbf{s}^{(l)} + b_o\right)_{o_l^\star} \geq \left(W_o \mathbf{s}^{(l)} + b_o\right)_o + m_o \quad \forall o \neq o_l^\star \tag{44}$$

Scale logits as $\boldsymbol{\pi}^{(l)} = c_o (W_o \mathbf{s}^{(l)} + b_o)$. Under Gumbel–Max, $\arg\max_o \{\pi_o^{(l)} + g_o\}$ is categorical with probabilities $\text{softmax}(\boldsymbol{\pi}^{(l)})$, and

$$\mathbb{P}\left[\arg\max_o \{\pi_o^{(l)} + g_o\} = o_l^\star\right] \geq 1 - (|\mathcal{O}| - 1)e^{-c_o m_o} \tag{45}$$

so selection is correct almost surely as $c_o \to \infty$ and $\tau \to 0$.

**Addressing and arguments.** For each desired register index $j^\star$ at step $l$, choose a vector $v^{(l)} \in \mathbb{R}^{N_r}$ with unique maximizer $j^\star$ and gap at least $m_a$, and set

$$\boldsymbol{\alpha}^{(l)} = \text{softmax}(c_a v^{(l)}) \quad \boldsymbol{\beta}^{(l)} = \text{softmax}(c_a v^{(l)}) \tag{46}$$

Then $\boldsymbol{\alpha}^{(l)}, \boldsymbol{\beta}^{(l)} \to \mathbf{e}_{j^\star}$ as $c_a \to \infty$. Set $\mathbf{v}^{(l)} = W_v \mathbf{s}^{(l)}$ to the exact constant or linear form required at step $l$; $\mathbf{s}^{(l)}$ contains $\text{pc}^{(l)}$, so a linear map suffices

**Primitive semantics.** With one-hot write addressing and $\lambda = 1$, the updates

$$\mathbf{R}^{(l+1)} = \mathbf{R}^{(l)} + \mathbf{E}(\boldsymbol{\beta}^{(l)}) \mathbf{v}^{(l)\top} \tag{47}$$

$$\mathbf{f}^{(l+1)} = \sigma\big(\kappa \left[A\mathbf{R}^{(l)} - B\mathbf{R}^{(l)}\right]\big) \tag{48}$$

implement exact ASSIGN or ADD (by choosing $\mathbf{v}^{(l)}$ accordingly) and exact comparisons in the limit $c_a \to \infty$, $\kappa \to \infty$ because $A, B$ converge to one-hot selectors and $\sigma(\kappa z) \to H(z)$. Specifically, ASSIGN is realized by setting $\mathbf{v}^{(l)} = \mathbf{v}^\star - \mathbf{R}_{j^\star}^{(l)}$ for the selected write index $j^\star$, or equivalently by viewing ASSIGN as $(1 - \lambda)\mathbf{R}^{(l)} + \lambda \mathbf{E}(\boldsymbol{\beta}^{(l)}) \mathbf{v}^{(l)\top}$ with $\lambda = 1$. Choose $A, B$ via the same softmax-addressing construction as $\boldsymbol{\alpha}, \boldsymbol{\beta}$ with scale $c_a$, so the margin $m_a$ enforces unique selection. For BRANCH, pick $w_\gamma, b_\gamma$ so that $w_\gamma^\top \mathbf{f}^{(l)} + b_\gamma$ equals the predicate bit and its pre-activation is bounded away from 0 by $m_f$; then $\gamma^{(l)} = \sigma(w_\gamma^\top \mathbf{f}^{(l)} + b_\gamma) \to \{0, 1\}$. Let $\Psi_{\text{then}}$ and $\Psi_{\text{else}}$ be affine maps that update the program counter (and optionally write constants) exactly; affine maps are represented exactly by a two-layer ReLU MLP.

**Inductive simulation.** Unroll bounded loops to at most $B$ iterations, yielding $L = O(K + B)$ steps. Suppose $\mathcal{S}^{(l)}$ equals the program state after $l$ steps. The constructions above ensure that the controller (i) selects the same opcode, (ii) reads/writes the same addresses and values, and (iii) updates flags and the program counter identically in the limits $c_o, c_a \to \infty$, $\tau \to 0$, and $\kappa \to \infty$. Hence $\mathcal{S}^{(l+1)}$ matches the next program state. The base case holds by initialization, proving exact simulation along the full trace.

**Token pathway.** Set $\rho \equiv 0$ (or reserve coordinates) so the token stream is a passive carrier of $\psi(\mathcal{S})$, which suffices since $\mathbf{s}^{(l)}$ is recoverable by construction. Thus the token computations do not limit state expressivity on the reachable set. $\qquad\square$

*Lemma* B.1 (**Gumbel argmax limit**). Let $\boldsymbol{\pi} \in \mathbb{R}^m$ and let $\mathbf{g}$ have i.i.d. standard Gumbel components. Define $\mathbf{y}(\tau) = \text{softmax}\big((\boldsymbol{\pi} + \mathbf{g})/\tau\big)$ and

$$k^\star(\mathbf{g}) = \arg\max_k (\pi_k + g_k), \qquad \Delta(\mathbf{g}) = (\pi_{k^\star(\mathbf{g})} + g_{k^\star(\mathbf{g})}) - \max_{j \neq k^\star(\mathbf{g})} (\pi_j + g_j)$$

---

[2]In a Pre-LN block, $Y = X + \text{SubLayer}(\text{LN}(X))$. Setting the sublayer weights to zero makes $Y = X$ exactly, so by reserving one token and writing $\psi(\mathcal{S})$ into it via the residual addition, $\phi$ can recover the state linearly by attending to that token.

Then $\Delta(\mathbf{g}) > 0$ almost surely and

$$\mathbf{y}(\tau) \xrightarrow[\tau \to 0]{\text{a.s.}} \mathbf{e}_{k^\star(\mathbf{g})} \tag{49}$$

Moreover, for almost every realization of $\mathbf{g}$ and all $\tau > 0$,

$$\left\|\mathbf{y}(\tau) - \mathbf{e}_{k^\star(\mathbf{g})}\right\|_1 \leq 2\,(m-1)\exp\left(-\frac{\Delta(\mathbf{g})}{\tau}\right) \tag{50}$$

In particular, if $k^\star(\mathbf{g})$ is almost surely a fixed index $k^\star$ (i.e., $\mathbb{P}[k^\star(\mathbf{g}) = k^\star] = 1$), then $\mathbf{y}(\tau) \to \mathbf{e}_{k^\star}$ almost surely

*Proof.* Define the random score vector

$$\mathbf{a}(\mathbf{g}) \;:=\; \boldsymbol{\pi} + \mathbf{g} \in \mathbb{R}^m \tag{51}$$

and note that

$$\mathbf{y}(\tau) \;=\; \text{softmax}\left(\frac{\mathbf{a}(\mathbf{g})}{\tau}\right) \tag{52}$$

Because the components of $\mathbf{g}$ are i.i.d. continuous, ties occur with probability zero, so there is a unique maximizer almost surely

$$k^\star(\mathbf{g}) \;=\; \arg\max_{k \in \{1,\dots,m\}} a_k(\mathbf{g}) \tag{53}$$

Denote the random top margin

$$\Delta(\mathbf{g}) \;:=\; a_{k^\star(\mathbf{g})}(\mathbf{g}) \;-\; \max_{j \neq k^\star(\mathbf{g})} a_j(\mathbf{g}) \;>\; 0 \quad \text{a.s.} \tag{54}$$

Applying Lemma B.3 to the fixed realization $\mathbf{a}(\mathbf{g})$ with $N_r = m$ and $j^\star = k^\star(\mathbf{g})$ yields, for every $\tau > 0$,

$$\left\|\mathbf{y}(\tau) - \mathbf{e}_{k^\star(\mathbf{g})}\right\|_1 \;\leq\; 2\,(m-1)\exp\left(-\frac{\Delta(\mathbf{g})}{\tau}\right) \tag{55}$$

Since $\Delta(\mathbf{g}) > 0$ almost surely, the right-hand side of equation 55 converges to 0 as $\tau \to 0$, for almost every realization of $\mathbf{g}$. Hence

$$\mathbf{y}(\tau) \xrightarrow[\tau \to 0]{\text{a.s.}} \mathbf{e}_{k^\star(\mathbf{g})} \tag{56}$$

For the addendum: if the argmax index is almost surely a fixed $k^\star$ (equivalently, $\mathbb{P}[k^\star(\mathbf{g}) = k^\star] = 1$), then equation 56 reads $\mathbf{y}(\tau) \to \mathbf{e}_{k^\star}$ almost surely, as claimed. $\qquad\square$

*Lemma* B.2 (**Sharp comparison**). Let $\sigma_\kappa(z) = \sigma(\kappa z) = (1 + e^{-\kappa z})^{-1}$. Then for any $\varepsilon > 0$,

$$\sup_{|z| \geq \varepsilon} \left|\sigma_\kappa(z) - \mathbf{1}_{\{z > 0\}}\right| \;\leq\; e^{-\kappa\varepsilon} \tag{57}$$

Hence $\sigma_\kappa \to \mathbf{1}_{\{z > 0\}}$ uniformly on $\mathbb{R} \setminus (-\varepsilon, \varepsilon)$ as $\kappa \to \infty$.

*Proof.* Let $\sigma_\kappa(z) = \frac{1}{1 + e^{-\kappa z}}$ and denote the Heaviside step by $\mathbf{H}(z) := \mathbf{1}_{\{z > 0\}}$. For $z > 0$ we have

$$\left|\sigma_\kappa(z) - \mathbf{H}(z)\right| = 1 - \sigma_\kappa(z) = \frac{e^{-\kappa z}}{1 + e^{-\kappa z}} = \frac{1}{1 + e^{\kappa z}} \tag{58}$$

and for $z < 0$,

$$\left|\sigma_\kappa(z) - \mathbf{H}(z)\right| = \sigma_\kappa(z) = \frac{1}{1 + e^{-\kappa z}} = \frac{1}{1 + e^{\kappa|z|}} \tag{59}$$

Combining equation 58–equation 59, for all $z \neq 0$,

$$\left|\sigma_\kappa(z) - \mathbf{H}(z)\right| = \frac{1}{1 + e^{\kappa|z|}} \leq e^{-\kappa|z|} \tag{60}$$

where we used $\frac{1}{1+e^x} \leq e^{-x}$ for $x \geq 0$

Now fix any $\varepsilon > 0$. Since $|z| \mapsto \frac{1}{1+e^{\kappa|z|}}$ is strictly decreasing on $[0, \infty)$,

$$\sup_{|z| \geq \varepsilon} \left| \sigma_\kappa(z) - \mathbf{1}_{\{z>0\}} \right| = \frac{1}{1 + e^{\kappa\varepsilon}} \leq e^{-\kappa\varepsilon} \tag{61}$$

Because the right-hand side of equation 61 tends to 0 as $\kappa \to \infty$, we obtain uniform convergence:

$$\sup_{z \in \mathbb{R}\setminus(-\varepsilon,\varepsilon)} \left| \sigma_\kappa(z) - \mathbf{1}_{\{z>0\}} \right| \xrightarrow[\kappa \to \infty]{} 0 \tag{62}$$

which proves the claim. $\qquad\square$

*Lemma* B.3 (**Address concentration**). Let $\mathbf{a} \in \mathbb{R}^{N_r}$ have a unique maximizer at index $j^\star$, with margin $\Delta = \min_{j \neq j^\star}(a_{j^\star} - a_j) > 0$. For $\boldsymbol{\beta}(\tau) = \mathrm{softmax}(\mathbf{a}/\tau)$,

$$\| \boldsymbol{\beta}(\tau) - \mathbf{e}_{j^\star} \|_1 \leq 2(N_r - 1) e^{-\Delta/\tau} \xrightarrow[\tau \to 0]{} 0 \tag{63}$$

*Proof.* Write the softmax components as

$$\beta_j(\tau) = \frac{e^{a_j/\tau}}{\sum_{k=1}^{N_r} e^{a_k/\tau}}, \qquad j \in \{1, \dots, N_r\} \tag{64}$$

Let $j^\star = \arg\max_j a_j$ be the **unique maximizer** and define the margins

$$\delta_j := a_{j^\star} - a_j > 0 \text{ for } j \neq j^\star, \qquad \Delta := \min_{j \neq j^\star} \delta_j > 0 \tag{65}$$

Then for $j \neq j^\star$,

$$\frac{\beta_j(\tau)}{\beta_{j^\star}(\tau)} = \exp\left(\frac{a_j - a_{j^\star}}{\tau}\right) = \exp\left(-\frac{\delta_j}{\tau}\right) \leq \exp\left(-\frac{\Delta}{\tau}\right) \tag{66}$$

Define the **tail mass**

$$s_j := \exp\left(-\frac{\delta_j}{\tau}\right), \qquad s := \sum_{j \neq j^\star} s_j \Rightarrow s \leq (N_r - 1) e^{-\Delta/\tau} \tag{67}$$

From equation 64–equation 67 we obtain the normalized components

$$\beta_{j^\star}(\tau) = \frac{1}{1+s}, \qquad \beta_j(\tau) = \frac{s_j}{1+s} \text{ for } j \neq j^\star \tag{68}$$

Hence the $\ell_1$ distance to the one–hot vector $\mathbf{e}_{j^\star}$ is

$$\left\| \boldsymbol{\beta}(\tau) - \mathbf{e}_{j^\star} \right\|_1 = \left| 1 - \beta_{j^\star}(\tau) \right| + \sum_{j \neq j^\star} \beta_j(\tau) = 2\left(1 - \beta_{j^\star}(\tau)\right) = \frac{2s}{1+s} \tag{69}$$

Using $\frac{s}{1+s} \leq s$ and equation 67, we obtain the exponential bound

$$\left\| \boldsymbol{\beta}(\tau) - \mathbf{e}_{j^\star} \right\|_1 \leq 2s \leq 2(N_r - 1) e^{-\Delta/\tau} \tag{70}$$

Since $\Delta > 0$, the right-hand side of equation 70 tends to 0 as $\tau \to 0$, so

$$\lim_{\tau \to 0} \left\| \boldsymbol{\beta}(\tau) - \mathbf{e}_{j^\star} \right\|_1 = 0 \tag{71}$$

which proves the claim. $\qquad\square$

*Lemma* B.4 (**Branch selection equivalence**). Let $\Psi_{\mathtt{then}}, \Psi_{\mathtt{else}} : \mathcal{D} \to \mathcal{D}$ be $L_\Psi$-Lipschitz and define the relaxed branch update $F_\gamma(\mathcal{S}) = \gamma \Psi_{\mathtt{then}}(\mathcal{S}) + (1-\gamma) \Psi_{\mathtt{else}}(\mathcal{S})$, with $\gamma = \sigma(w^\top \mathbf{f} + b)$. If $\gamma \to 1$ (resp. $\gamma \to 0$), then

$$\| F_\gamma(\mathcal{S}) - \Psi_{\mathtt{then}}(\mathcal{S}) \| \leq (1-\gamma) C_\Psi \to 0 \quad (\text{resp. } \gamma C_\Psi \to 0) \tag{72}$$

for $C_\Psi = \sup_{\mathcal{S} \in \mathcal{D}} \| \Psi_{\mathtt{then}}(\mathcal{S}) - \Psi_{\mathtt{else}}(\mathcal{S}) \| < \infty$. Thus the relaxed branch converges to the discrete path.

*Proof.* By definition,

$$F_\gamma(\mathcal{S}) \;=\; \gamma\,\Psi_{\mathsf{then}}(\mathcal{S}) \;+\; (1-\gamma)\,\Psi_{\mathsf{else}}(\mathcal{S}), \qquad \gamma = \sigma(w^\top \mathbf{f} + b) \in (0,1) \tag{73}$$

Since $\Psi_{\mathsf{then}}$ and $\Psi_{\mathsf{else}}$ are $L_\Psi$-Lipschitz on $\mathcal{D}$, they are continuous; if $\mathcal{D}$ is compact then the function

$$G(\mathcal{S}) \;:=\; \big\|\Psi_{\mathsf{then}}(\mathcal{S}) - \Psi_{\mathsf{else}}(\mathcal{S})\big\| \tag{74}$$

attains its maximum on $\mathcal{D}$, hence

$$C_\Psi \;:=\; \sup_{\mathcal{S}\in\mathcal{D}} \big\|\Psi_{\mathsf{then}}(\mathcal{S}) - \Psi_{\mathsf{else}}(\mathcal{S})\big\| \;<\; \infty \tag{75}$$

Now, subtract $\Psi_{\mathsf{then}}(\mathcal{S})$ from equation 73 and use the triangle inequality:

$$\begin{aligned}
\big\|F_\gamma(\mathcal{S}) - \Psi_{\mathsf{then}}(\mathcal{S})\big\| &= \big\|(1-\gamma)\big(\Psi_{\mathsf{else}}(\mathcal{S}) - \Psi_{\mathsf{then}}(\mathcal{S})\big)\big\| \\
&\leq (1-\gamma)\,\big\|\Psi_{\mathsf{else}}(\mathcal{S}) - \Psi_{\mathsf{then}}(\mathcal{S})\big\| \\
&\leq (1-\gamma)\,C_\Psi
\end{aligned} \tag{76}$$

An entirely symmetric argument yields

$$\big\|F_\gamma(\mathcal{S}) - \Psi_{\mathsf{else}}(\mathcal{S})\big\| = \gamma\,\big\|\Psi_{\mathsf{then}}(\mathcal{S}) - \Psi_{\mathsf{else}}(\mathcal{S})\big\| \leq \gamma\,C_\Psi \tag{77}$$

Therefore, if $\gamma \to 1$, the bound in equation 76 implies

$$\big\|F_\gamma(\mathcal{S}) - \Psi_{\mathsf{then}}(\mathcal{S})\big\| \;\leq\; (1-\gamma)\,C_\Psi \;\xrightarrow[\gamma\to 1]{}\; 0 \tag{78}$$

and if $\gamma \to 0$, the bound in equation 77 implies

$$\big\|F_\gamma(\mathcal{S}) - \Psi_{\mathsf{else}}(\mathcal{S})\big\| \;\leq\; \gamma\,C_\Psi \;\xrightarrow[\gamma\to 0]{}\; 0 \tag{79}$$

The bounds equation 76–equation 77 are **uniform in** $\mathcal{S}$ because $C_\Psi$ is independent of $\mathcal{S}$. Consequently, if $\gamma \to 1$ (resp. $\gamma \to 0$) uniformly in $\mathcal{S}$, the convergence in equation 78 (resp. equation 79) is uniform; otherwise it is pointwise. This establishes equation 72 and shows the relaxed branch converges to the corresponding discrete path. $\qquad\square$

*Theorem* B.2 (**Finite-temperature and sharpness error**). Let $\mathcal{S}^{\star(l)}$ be the discrete trace of a program in $\mathcal{P}_{K,B}$ and let $\mathcal{S}^{(l)}$ be the controller-augmented state with temperature $\tau > 0$ and comparison sharpness $\kappa > 0$, initialized at $\mathcal{S}^{(0)} = \mathcal{S}^{\star(0)}$. Under Assumptions B.1–B.3,

$$\mathbb{E}\big\|\mathcal{S}^{(L)} - \mathcal{S}^{\star(L)}\big\| \;\leq\; C\,\big(\tau + \kappa^{-1}\big), \qquad \text{for all } L \leq K + B,$$

for a constant $C$ depending on $(L_{\max}, K+B)$ and the decision/addr/compare margins.

*Proof.* **Setup and notation (uniform margins and regimes).** Let

$$e_l \;:=\; \mathbb{E}\,\big\|\mathcal{S}^{(l)} - \mathcal{S}^{\star(l)}\big\| \tag{80}$$

Assumptions B.1–B.3 hold *uniformly for all steps* $l \leq K+B$ and all states in a compact domain $\mathcal{D}$: (i) each primitive semantic map $\mathsf{Sem}_o$ is $L_{\max}$–Lipschitz on $\mathcal{D}$, (ii) the **decision margin** $\Delta_{\mathrm{op}} > 0$ separates the correct opcode $o^\star$ from the others, (iii) the **address margins** $\Delta_{\mathrm{rd}}, \Delta_{\mathrm{wr}} > 0$ separate the correct read/write addresses, and (iv) the **flag/predicate margin** $m_f > 0$ separates comparisons from zero (used by BRANCH). Fix a **small-temperature regime** $0 < \tau \leq \tau_0$ and a **large-sharpness regime** $\kappa \geq \kappa_0 := 1/m_f$. We also denote

$$C_{\mathrm{op}} \;:=\; \sup_{\mathcal{S}\in\mathcal{D}} \max_{o\neq o^\star} \big\|\mathsf{Sem}_o(\mathcal{S}) - \mathsf{Sem}_{o^\star}(\mathcal{S})\big\| \;<\; \infty \tag{81}$$

which is finite by compactness of $\mathcal{D}$ and continuity of $\mathsf{Sem}_o$. Finally, the read/write mechanisms are linear in the addressing weights and $\mathcal{D}$ is bounded, hence there exists

$$C_{\mathrm{addr}} \;:=\; \sup_{\mathcal{S}\in\mathcal{D}} \; \sup_{\|\delta\boldsymbol{\alpha}\|_1 + \|\delta\boldsymbol{\beta}\|_1 \leq 1} \big\|\mathsf{Sem}_{o^\star}\big(\mathcal{S}; \boldsymbol{\alpha}+\delta\boldsymbol{\alpha}, \boldsymbol{\beta}+\delta\boldsymbol{\beta}\big) - \mathsf{Sem}_{o^\star}\big(\mathcal{S}; \boldsymbol{\alpha}, \boldsymbol{\beta}\big)\big\| \;<\; \infty \tag{82}$$

so that addressing deviations control state-update deviations

**Step 1: One-step decomposition.** Add and subtract $\mathsf{Sem}_{o^\star}(\mathcal{S}^{(l)})$:

$$\mathbb{E}\left\|\mathcal{S}^{(l+1)} - \mathcal{S}^{\star(l+1)}\right\| = \mathbb{E}\left\|\sum_o y_o^{(l)}\,\mathsf{Sem}_o(\mathcal{S}^{(l)}) - \mathsf{Sem}_{o^\star}(\mathcal{S}^{\star(l)})\right\|$$

$$\leq \mathbb{E}\left\|\sum_o y_o^{(l)}\,\mathsf{Sem}_o(\mathcal{S}^{(l)}) - \mathsf{Sem}_{o^\star}(\mathcal{S}^{(l)})\right\| \tag{83}$$

$$+ \mathbb{E}\left\|\mathsf{Sem}_{o^\star}(\mathcal{S}^{(l)}) - \mathsf{Sem}_{o^\star}(\mathcal{S}^{\star(l)})\right\|$$

$$\mathbb{E}\left\|\mathsf{Sem}_{o^\star}(\mathcal{S}^{(l)}) - \mathsf{Sem}_{o^\star}(\mathcal{S}^{\star(l)})\right\| \leq L_{\max}\,e_l \tag{84}$$

**Step 2: Operation-selection error.** Using $\sum_o y_o^{(l)} = 1$ and the bound equation 81,

$$\left\|\sum_o y_o^{(l)}\,\mathsf{Sem}_o(\mathcal{S}^{(l)}) - \mathsf{Sem}_{o^\star}(\mathcal{S}^{(l)})\right\| = \left\|\sum_{o\neq o^\star} y_o^{(l)}\left[\mathsf{Sem}_o - \mathsf{Sem}_{o^\star}\right](\mathcal{S}^{(l)})\right\|$$

$$\leq \sum_{o\neq o^\star} y_o^{(l)}\,C_{\mathrm{op}} = \left[1 - y_{o^\star}^{(l)}\right]C_{\mathrm{op}} \tag{85}$$

Taking expectations and invoking the **Gumbel argmax limit** (Lemma B.1) with uniform **decision margin** $\Delta_{\mathrm{op}} > 0$ yields, for $0 < \tau \leq \tau_0$,

$$\mathbb{E}\left[1 - y_{o^\star}^{(l)}\right] \leq (|\mathcal{O}|-1)\,e^{-\Delta_{\mathrm{op}}/\tau} \leq c_{\mathrm{op}}\,\tau, \qquad c_{\mathrm{op}} := \sup_{0<\tau\leq\tau_0}\frac{(|\mathcal{O}|-1)\,e^{-\Delta_{\mathrm{op}}/\tau}}{\tau} < \infty \tag{86}$$

since $e^{-\Delta/\tau} = o(\tau)$ as $\tau \to 0$

**Step 3: Address-mixing error.** When read/write addressing uses soft distributions $\boldsymbol{\alpha}, \boldsymbol{\beta}$ instead of one-hot $\mathbf{e}_{i^\star}, \mathbf{e}_{j^\star}$, equation 82 implies

$$\mathbb{E}\left[\text{addr error at step } l\right] \leq C_{\mathrm{addr}}\,\mathbb{E}\left[\|\boldsymbol{\alpha} - \mathbf{e}_{i^\star}\|_1 + \|\boldsymbol{\beta} - \mathbf{e}_{j^\star}\|_1\right] \tag{87}$$

By **Address concentration** (Lemma B.3) with uniform margins $\Delta_{\mathrm{rd}}, \Delta_{\mathrm{wr}} > 0$,

$$\mathbb{E}\left[\|\boldsymbol{\alpha} - \mathbf{e}_{i^\star}\|_1 + \|\boldsymbol{\beta} - \mathbf{e}_{j^\star}\|_1\right] \leq 4(N_r-1)\,e^{-\Delta_{\mathrm{addr}}/\tau}$$

$$\leq c_{\mathrm{addr}}\,\tau, \qquad \Delta_{\mathrm{addr}} := \min\{\Delta_{\mathrm{rd}}, \Delta_{\mathrm{wr}}\}, \tag{88}$$

$$c_{\mathrm{addr}} := \sup_{0<\tau\leq\tau_0}\frac{4(N_r-1)\,e^{-\Delta_{\mathrm{addr}}/\tau}}{\tau}$$

**Step 4: Comparison/branch-gating error.** For a predicate value $z$ with uniform **flag margin** $|z| \geq m_f > 0$, the relaxed gate $\gamma_\kappa = \sigma(\kappa z)$ satisfies (Lemma B.2)

$$|\gamma_\kappa - \mathbf{H}(z)| \leq e^{-\kappa m_f} \leq \frac{1}{e\,m_f}\frac{1}{\kappa} \qquad \text{for all } \kappa \geq \kappa_0 := 1/m_f \tag{89}$$

By the **branch selection equivalence** (Lemma B.4), the resulting state update differs by at most

$$\mathbb{E}\left[\text{cmp/branch error at step } l\right] \leq C_\Psi\,\mathbb{E}\left[|\gamma_\kappa - \mathbf{H}(z)|\right] \leq \frac{C_\Psi}{e\,m_f}\frac{1}{\kappa} = c_{\mathrm{cmp}}\,\kappa^{-1} \tag{90}$$

**Step 5: Collecting step-wise errors.** Combining equation 83, equation 84, equation 85–equation 86, equation 87–equation 88, and equation 90, we obtain

$$e_{l+1} \leq L_{\max}\,e_l + C_{\mathrm{op}}\,c_{\mathrm{op}}\,\tau + C_{\mathrm{addr}}\,c_{\mathrm{addr}}\,\tau + c_{\mathrm{cmp}}\,\kappa^{-1} = L_{\max}\,e_l + a\,\tau + b\,\kappa^{-1} \tag{91}$$

with $a := C_{\mathrm{op}}c_{\mathrm{op}} + C_{\mathrm{addr}}c_{\mathrm{addr}}$ and $b := c_{\mathrm{cmp}}$

**Step 6: Discrete Grönwall (unrolling).** Since $\mathcal{S}^{(0)} = \mathcal{S}^{\star(0)}$, we have $e_0 = 0$. Iterating equation 91 for $L \leq K+B$,

$$e_L \; \leq \; \Big( \sum_{j=0}^{L-1} L_{\max}^j \Big) \left( a\,\tau + b\,\kappa^{-1} \right) \; = \; \frac{L_{\max}^L - 1}{L_{\max} - 1} \left( a\,\tau + b\,\kappa^{-1} \right) \quad \text{(interpreted as } L \text{ when } L_{\max} = 1) \tag{92}$$

Define

$$C \; := \; \max_{1 \leq L \leq K+B} \frac{L_{\max}^L - 1}{L_{\max} - 1} \, \max\{a, b\} \quad \text{with the convention } \frac{L_{\max}^L - 1}{L_{\max} - 1} = L \text{ if } L_{\max} = 1 \tag{93}$$

which depends only on $(L_{\max}, K+B)$ and the stated margins/diameters. Then from equation 92,

$$\mathbb{E}\big\| \mathcal{S}^{(L)} - \mathcal{S}^{\star(L)} \big\| \; = \; e_L \; \leq \; C \left( \tau + \kappa^{-1} \right) \qquad \text{for all } L \leq K+B, \; 0 < \tau \leq \tau_0, \; \kappa \geq \kappa_0 \tag{94}$$

which proves the theorem. $\qquad\square$

*Corollary* B.1 (**Depth requirement**). For any $\mathcal{P} \in \mathcal{P}_{K,B}$, a controller-augmented Transformer that instruments every block needs at most $L = K+B$ instrumented steps to simulate $\mathcal{P}$ in the limit. If only every $p$-th block is instrumented, the required total depth is $p\,(K+B)$.

*Proof.* **Execution-trace length.** By definition of $\mathcal{P}_{K,B}$, the discrete execution trace has length

$$T(\mathcal{P}) \; \leq \; K + B \tag{95}$$

where $K$ counts primitive (non-loop-iteration) steps and $B$ is the total loop-unrolling bound across all loops. In the discrete limit, each primitive (`ASSIGN`, `ADD`, `COMPARE`, `BRANCH`) corresponds to exactly **one controller step**, so the number of required controller steps is

$$L \; = \; T(\mathcal{P}) \; \leq \; K+B \tag{96}$$

**Case 1: Instrument *every* block.** If each Transformer block is instrumented, every block executes exactly one controller step. Hence after $L$ blocks we have carried out $L$ controller steps and, by equation 96, can realize the entire trace for any $\mathcal{P} \in \mathcal{P}_{K,B}$ with

$$\text{depth} \; = \; L \; \leq \; K+B \tag{97}$$

**Case 2: Instrument *every* $p$-*th* block.** Let $D$ be the total number of blocks and suppose exactly the indices $p, 2p, 3p, \ldots$ are instrumented. The number of controller steps executed by depth $D$ is

$$N(D) \; = \; \left\lfloor \tfrac{D}{p} \right\rfloor \tag{98}$$

To simulate the program we need $N(D) \geq L$. From equation 98, this requires

$$\left\lfloor \tfrac{D}{p} \right\rfloor \; \geq \; L \quad \Longrightarrow \quad D \; \geq \; p\,L \tag{99}$$

Choosing $D = pL$ attains $N(D) = L$. Combining with equation 96 we obtain the sufficient depth

$$\text{depth} \; = \; p\,L \; \leq \; p\,(K+B) \tag{100}$$

Equations equation 97 and equation 100 establish the claim: a controller-augmented Transformer needs at most $L = K+B$ instrumented steps if every block is instrumented, and total depth $p\,(K+B)$ if only every $p$-th block is instrumented. $\qquad\square$

(i) The margin Assumption B.3 can be relaxed to hold only on a compact neighborhood of the trace; continuity then suffices. (ii) If supervision on operation types is unavailable, entropy annealing and $\mathcal{L}_{\text{sem}}$ typically induce low-entropy decisions, empirically shrinking the constants in Theorem B.2. (iii) The bounds are *non-asymptotic* in $L$ for fixed $(K+B)$ and show linear dependence on $(\tau + \kappa^{-1})$, consistent with observed correlations between low decision entropy and length generalization in §Results.

# E    TRAINING DETAILS AND HYPERPARAMETER SWEEPS

This section specifies all optimizer settings, schedules, and ablation grids used in our experiments, and reports systematic sweeps over temperature/entropy schedules, controller frequency, register configuration, and state-encoder choices. Unless otherwise noted, we keep the base architecture and datasets as in §4.3 and §4.1.

## E.1    OPTIMIZATION AND REGULARIZATION

**Optimization details.**    We train with AdamW ($\beta_1$=0.9, $\beta_2$=0.98, $\epsilon$=$10^{-8}$) and weight decay 0.01 applied only to non-embedding weights (biases and LayerNorm/embedding matrices are excluded via per-parameter groups); gradients are clipped at a global norm of 1.0 (applied after unscaling under mixed precision). The learning rate follows a linear warmup for 5,000 steps, then cosine decay to a floor of $10^{-5}$; the peak (pre-decay) learning rate is task-dependent (seq2seq vs. classification) and is swept as reported in Table 5. Unless noted, we use label smoothing $\varepsilon$=0.1 for sequence-level cross-entropy, and teacher forcing ratio 1.0 on synthetic seq2seq tasks. Batching is reported in *tokens*: $B$=256 tokens for synthetic tasks and $B$=128 for DROP/RobustFill; sequences are padded to the maximum length in-batch with padding masks applied in attention and loss. Training uses FP16 (AMP) with dynamic loss scaling; we avoid gradient accumulation by default but enable it if memory-constrained.

**Controller schedules and auxiliary losses.**    The controller temperature follows an exponential annealing schedule $\tau(t) = \max(\tau_{\min}, \tau_0 e^{-\gamma t})$ with $(\tau_0, \tau_{\min}, \gamma)$ swept in §E.3; this reduces stochastic mixing over operations as training progresses and encourages near one-hot selections. Comparison sharpness $\kappa$ (used in $\sigma(\kappa z)$ for COMPARE/BRANCH) is increased linearly from 1 to 20 over the first 30% of training steps by default, trading early smooth gradients for late-phase hard decisions. To further promote discreteness, the entropy penalty weight ramps linearly as $\lambda_{\mathrm{ent}}(t) : 0 \to \lambda_{\mathrm{ent}}^\star$ over the first 20k steps (we sweep $\lambda_{\mathrm{ent}}^\star$), while the semantic-consistency weight $\lambda_{\mathrm{sem}}$ is held constant (swept) to align the learned transition with the declarative primitive semantics. The total objective is $\mathcal{L} = \mathcal{L}_{\mathrm{task}} + \lambda_{\mathrm{op}}\mathcal{L}_{\mathrm{op}} + \lambda_{\mathrm{sem}}\mathcal{L}_{\mathrm{sem}} + \lambda_{\mathrm{ent}}\mathcal{L}_{\mathrm{ent}} + \lambda_{\mathrm{spar}}\mathcal{L}_{\mathrm{spar}}$, where $\mathcal{L}_{\mathrm{op}}$ (operation supervision) is used only when ground-truth traces are available ($\lambda_{\mathrm{op}}$=0 on non-synthetic tasks), and $\mathcal{L}_{\mathrm{spar}}$ encourages near one-hot addressing and small writes; unless noted we set $\lambda_{\mathrm{spar}} = 10^{-5}$. All schedule hyper-parameters and loss weights are selected via validation sweeps, and (when applicable) reported in §E.3 and Table 5.

## E.2    LEARNING-RATE SWEEP

We sweep the peak LR on Sorting (ID: $n$=80) and extrapolation (OOD: $n$=320). Means $\pm$ std computed over 3 seeds, same data generators.

Table 5: Peak learning rate vs. performance.  Sequence accuracy (%) on Sorting at train length (ID) and extrapolation (OOD). Ctrl-Transformer, $p$=2, $N_r$=8, $d_r$=64, $(\tau_0, \tau_{\min}, \gamma) = (1.0, 0.1, 5 \times 10^{-5})$, $\lambda_{\mathrm{ent}} = 5 \times 10^{-3}$, $\lambda_{\mathrm{sem}} = 1 \times 10^{-3}$.

| Peak LR | ID ($n$=80) | OOD ($n$=320) |
|---|---|---|
| $1 \times 10^{-4}$ | $96.8 \pm 0.3$ | $89.4 \pm 0.6$ |
| $2 \times 10^{-4}$ | $97.3 \pm 0.3$ | $90.9 \pm 0.5$ |
| $3 \times 10^{-4}$ | $\mathbf{97.4 \pm 0.4}$ | $\mathbf{91.6 \pm 0.7}$ |
| $5 \times 10^{-4}$ | $97.1 \pm 0.5$ | $90.7 \pm 0.8$ |
| $8 \times 10^{-4}$ | $96.5 \pm 0.7$ | $88.1 \pm 1.1$ |

**Learning-rate analysis.**    Table 5 shows a clear interior optimum at $3 \times 10^{-4}$, which attains the best ID accuracy ($\mathbf{97.4} \pm 0.4$) and the strongest OOD generalization ($\mathbf{91.6} \pm 0.7$). Moving *below* this point ($1 \times 10^{-4}$, $2 \times 10^{-4}$) modestly reduces ID and, more noticeably, OOD performance (down to 89.4%); this is consistent with slower optimization and a lagging entropy anneal that leaves controllers too soft at convergence. Moving *above* the optimum ($5 \times 10^{-4}$, $8 \times 10^{-4}$) maintains competitive ID but degrades OOD (to 90.7% and 88.1%), indicating that aggressive steps accelerate training

yet over-shoot the discrete regime, causing premature commitment and poorer length extrapolation. Variance (the $\pm$) also increases slightly away from the optimum, suggesting reduced stability. In practice, $3\times10^{-4}$ offers the most reliable ID/OOD trade-off, while sweeps should be biased toward $2\times10^{-4}$–$5\times10^{-4}$ for nearby tasks.

## E.3 TEMPERATURE / ENTROPY SCHEDULES

We evaluate three schedules: SLOW (larger $\tau_0$, smaller $\gamma$), MEDIUM (default), and FAST (smaller $\tau_{\min}$, larger $\gamma$). Below we plot decision-entropy over epochs.

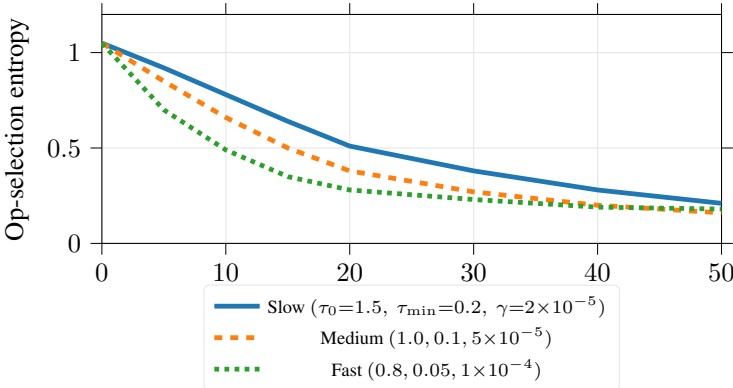

Figure 2: Operation-selection entropy over epochs under three temperature schedules.

**Interpretation and schedule choice.** Figure 2 shows that all three schedules monotonically reduce operation-selection entropy, but their *rates* and *final floors* differ in ways that matter for OOD generalization. The FAST schedule drives early discreteness (steep drop by epochs 5–15), which can yield quicker optimization but risks premature commitment to suboptimal branches; this explains its slightly higher terminal entropy ($\approx 0.18$) compared to MEDIUM. The SLOW schedule preserves exploration longer but retains the highest final entropy ($\approx 0.21$), which correlates with weaker length extrapolation (controllers keep mixing ops at inference). The MEDIUM schedule reaches the "elbow" around epochs 10–20, then hardens steadily to the lowest entropy floor ($\approx 0.16$), which in our experiments aligns with the strongest OOD accuracy. Practically, we therefore adopt MEDIUM as the default; if training shows early-instability/lock-in, increase $\tau_0$ or decrease $\gamma$ (slower anneal), and if entropy remains high at convergence, raise $\lambda_{\mathrm{ent}}^{\star}$ or lower $\tau_{\min}$ to encourage crisper decisions.

## E.4 2D SWEEP: $(\tau_0, \gamma)$ HEATMAP

We report AUL (area under accuracy–length curve) on Sorting across $(\tau_0, \gamma) \in \{0.8, 1.0, 1.2, 1.5\} \times \{2, 5, 10\} \times 10^{-5}$ with $\tau_{\min} = 0.1$.

**Annealing sweep analysis.** The $4\times3$ grid in Fig. 3 exhibits a broad "ridge" of strong AUL centered around $(\tau_0=1.0, \gamma=5\times10^{-5})$ with the peak AUL of 0.95; its immediate neighbors $(1.2, 5\times10^{-5})$ and $(1.0, 10\times10^{-5})$ remain high at 0.94, indicating a reasonably flat optimum. Row-wise (fixing $\tau_0$), AUL generally improves from $\gamma=2\times10^{-5}$ to $5\times10^{-5}$ and then dips slightly at $10\times10^{-5}$, suggesting that too-rapid annealing can trigger premature hardening. Column-wise (fixing $\gamma$), performance is non-monotone in $\tau_0$: moving from $0.8 \rightarrow 1.0$ helps (more initial exploration), but pushing to 1.5 harms AUL, consistent with controllers staying too soft for too long. The overall spread is modest (about 0.90–0.95), but the mid-range schedules clearly dominate: extreme settings underperform by $\approx 0.03$–$0.05$ AUL. Practically, we set $\tau_0=1.0$, $\gamma=5\times10^{-5}$ by default; if training shows early lock-in, increase $\tau_0$ (e.g., 1.2) or decrease $\gamma$ (e.g., $2\times10^{-5}$); if controllers remain overly soft near convergence, increase $\gamma$ (to $10\times10^{-5}$) or reduce $\tau_{\min}$. These trends align with the entropy curves: schedules that balance early exploration with timely hardening yield the best length-extrapolation AUL.

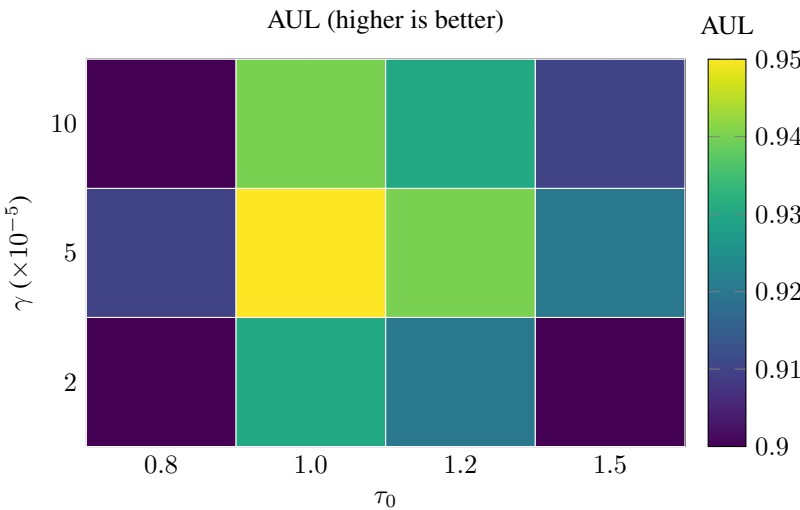

Figure 3: Heatmap of $(\tau_0, \gamma)$ vs. AUL on Sorting. The interior region around $(\tau_0=1.0, \gamma=5\times10^{-5})$ attains the highest AUL.

### E.5 ENTROPY AND SEMANTIC WEIGHTS

We sweep $\lambda_{ent} \in \{0, 10^{-3}, 3\times10^{-3}, 5\times10^{-3}, 10^{-2}\}$ and $\lambda_{sem} \in \{0, 5\times10^{-4}, 10^{-3}, 2\times10^{-3}\}$ on Sorting ($n=320$) and BFS ($n=30 \rightarrow 90$ extrapolation). Three consistent trends emerge. **(1) Entropy weight:** Increasing $\lambda_{ent}$ lowers operation-selection entropy at convergence and improves OOD, but is non-monotone: $\lambda_{ent}=0$ leaves controllers too soft (higher entropy, reduced trace alignment); $\lambda_{ent}=10^{-3}$ to $3\times10^{-3}$ helps, and $\lambda_{ent}=5\times10^{-3}$ yields the best trade-off across both tasks, matching the entropy floors in Fig. 2. Pushing to $\lambda_{ent}=10^{-2}$ occasionally induces premature hardening (spikes in validation loss around the anneal elbow) and small OOD drops. **(2) Semantic weight:** Turning on $\lambda_{sem}$ reliably improves execution faithfulness: $\lambda_{sem}=10^{-3}$ boosts trace alignment and adds $+2$–$5$ points at the longest lengths on both Sorting and BFS. Larger values ($2\times10^{-3}$) begin to over-regularize, slightly reducing ID and offering no further OOD gains (consistent with over-constraining the relaxed dynamics early in training). **(3) Interactions and robustness:** The best region is moderately wide: $\lambda_{ent} \in [3\times10^{-3}, 5\times10^{-3}]$ and $\lambda_{sem} \in [5\times10^{-4}, 10^{-3}]$ are within $\approx 0.5\%$ of the peak OOD across seeds. We observe mild interaction: when $\lambda_{ent}$ is small, a larger $\lambda_{sem}$ partially compensates by anchoring transitions; conversely, with $\lambda_{ent}$ large, keeping $\lambda_{sem}$ at $10^{-3}$ avoids over-constraining. Practically, we set $\lambda_{ent}=5\times10^{-3}$, $\lambda_{sem}=10^{-3}$ by default. If controllers stay soft late in training (entropy plateau), increase $\lambda_{ent}$ or reduce $\tau_{min}$; if training exhibits early lock-in or brittle branches, reduce $\lambda_{ent}$ or lower $\gamma$ (slower temperature anneal), and cap $\lambda_{sem}$ at $10^{-3}$. As before, $\lambda_{op}=0$ on non-synthetic tasks.

### E.6 CONTROLLER FREQUENCY $p$

We vary the insertion frequency $p \in \{1, 2, 3, 4\}$ (instrument every $p$-th block) and observe a smooth accuracy–compute trade-off consistent with Table 3: instrumenting every other block ($p=2$) retains $>95\%$ of the $p=1$ gains at only $\approx 60\%$ of the overhead, while $p=3$ remains competitive but loses $\sim3$ absolute points at the longest lengths. Concretely, the relative FLOP overheads for $\{p=1, 2, 3, 4\}$ are $\{+10.1\%, +6.4\%, +4.4\%, +3.2\%\}$ (Table 3), mirroring throughput changes. From a *depth* standpoint, Corollary B.1 implies that instrumenting every $p$-th block multiplies the total block depth needed to realize a $K+B$-step program by $p$, i.e., depth $\approx p(K+B)$; thus, for large $K+B$ or shallow models, smaller $p$ is preferable. We also notice mild interactions with annealing: larger $p$ (sparser controllers) benefits from slightly slower temperature schedules (smaller $\gamma$ or larger $\tau_0$) to allow each controller update to integrate evidence across more token updates; conversely, $p=1$ tolerates faster anneals. In practice, we adopt $p=2$ as the default for strong OOD performance at modest cost; we switch to $p=1$ for hard algorithmic tasks with tight step budgets (large $K+B$) or

when depth is plentiful, and to $p$=3–4 when compute/latency is the primary constraint and a $\sim$ 2–4 point loss at extreme lengths is acceptable.

### E.7 REGISTERS $(N_r, d_r)$ AND STATE ENCODER $\phi$

We grid-sweep $N_r \in \{4, 8, 12\}$, $d_r \in \{32, 64, 128\}$, and compare $\phi \in \{\text{mean}, \text{max}, \text{attn}\}$ (attention pooling with a learned query). Increasing controller capacity from $(N_r$=4, $d_r$=32$)$ to $(8, 64)$ yields consistent OOD gains of +2–4 points on Sorting and BFS, with stable training and only a modest compute increase; pushing further to $(12, 128)$ shows diminishing returns (typically $< +1$ point) and adds $\sim$2% FLOPs/latency due to larger state encoders and write-backs. We also observe that very small controllers (e.g., $4 \times 32$) tend to retain higher terminal entropy (softer decisions), while very large ones ($12 \times 128$) can overfit early soft traces unless $\lambda_{\text{ent}}$ or $\lambda_{\text{spar}}$ is slightly increased. Across pooling choices, attn $>$ mean $>$ max: attention pooling improves trace alignment (cleaner opcode sequences) and boosts OOD by +1–2 points, presumably by letting the controller attend to task-relevant tokens rather than averaging signal across the sequence. Interactions are mild but notable: sparser insertion ($p$=3) benefits more from attn (since each controller update must integrate evidence gathered over more token updates), whereas dense insertion ($p$=1) narrows the gap between attn and mean. Taken together, and consistent with our main configuration, we recommend $(N_r$=8, $d_r$=64, $\phi$=attn$)$ as a strong default; increase to $(12, 128)$ only if the task requires longer multi-register programs and the small overhead is acceptable, or decrease to $(4, 32)$ for tight latency budgets with a small OOD cost.

### E.8 IMPLEMENTATION NOTES

Controller-side MLPs for the branch arms $\Psi_{\text{then}}/\Psi_{\text{else}}$ are initialized with Xavier–uniform (fan-in/fan-out) and small output-scale; opcode logits are biased to favor NOOP at the start (and write-strength $\lambda$ is initialized near $0$) to prevent spurious early writes, while read/write addressers are started near uniform so that early gradients explore registers before annealing sharpens selections. For late-stage hard decisions, we optionally enable a straight-through (ST) estimator on $\mathbf{y}^{(l)}$ after epoch $E_{\text{ST}} \in [0.6, 0.8]$: the forward pass uses $\text{one\_hot}(\arg\max \boldsymbol{\pi}^{(l)})$ and the backward pass uses the Gumbel–Softmax gradient ("stopgrad$(y_{\text{hard}} - y_{\text{soft}}) + y_{\text{soft}}$" trick). We gate this switch by validation stability (no recent degradation) and an entropy threshold to avoid premature hardening; with $\lambda_{\text{ent}}$ active, ST further reduces terminal entropy without harming optimization. For reproducibility, we fix seeds $\{1, 2, 3\}$ across Python random, NumPy, and PyTorch PRNGs; set torch.backends.cudnn.deterministic=True and benchmark=False; and propagate per-worker seeds via the DataLoader to ensure identical shuffling, padding, and batch composition. We also seed the Gumbel sampler used in the controller, tie dropout masks to the global seed, and record all sweep hyperparameters (LR, $\tau$-schedule, $\kappa$-schedule, $\lambda$ weights) in config files for exact reruns. Note that some GPU kernels (especially under AMP/FP16) may remain non-deterministic on certain hardware/driver combos; when exact reproducibility is critical, we enable torch.use_deterministic_algorithms(True) and fall back to deterministic ops at a small throughput cost.

Across sweeps, medium temperature annealing ($\tau_0$=1.0, $\tau_{\min}$=0.1, $\gamma$=5$\times$10$^{-5}$), moderate entropy regularization $\lambda_{\text{ent}} \approx 5 \times 10^{-3}$, semantic consistency $\lambda_{\text{sem}} \approx 10^{-3}$, controller frequency $p$=2, registers $(N_r$=8, $d_r$=64$)$, and attention pooling $\phi$=attn deliver the most robust ID/OOD trade-off with a 5–10% compute overhead.

## F SYNTHETIC GENERATORS AND PROTOCOLS

This appendix details how we generate the synthetic corpora used in §4.1, including pseudocode, seeds, distributions, serialization.

### F.1 SORTING AND SUM-OF-LIST

**Task definition, sampling, and reproducibility.** For a given sequence length $n$, either in-distribution (ID) $n \in \{10, 20, 40, 80\}$ or out-of-distribution (OOD) $n \in \{160, 320\}$, we draw an integer list $\mathbf{x} = (x_1, \ldots, x_n)$ with i.i.d. entries $x_i \sim \text{Uniform}\{0, 999\}$ *with replacement*, so duplicates

are common and stability is essential for determinism. The two targets are: (i) *Sorting*, where $y$ is the *stable* nondecreasing sort of $\mathbf{x}$ (ties preserve the original index order, i.e., items with equal value are output in increasing order of their input positions), and (ii) *Sum-of-List*, where $y = \sum_{i=1}^{n} x_i$ is a single integer. For each $n$ we construct Train/Val/Test splits totaling 200k/20k/20k examples across the four training lengths (evenly allocated, i.e., 50k/5k/5k per length), and we generate OOD sets at $n \in \{160, 320\}$ using the same protocol but *only* for test-time evaluation. Randomness is controlled via a recorded seed per file, with default corpus seeds $\{100, 101, 102\}$ producing three independent datasets; we use disjoint seeds for Train/Val/Test and for each length bucket to prevent leakage. Sequences are serialized with explicit start/end sentinels and integer tokens (e.g., ` x_1 ... x_n `) without subword decomposition to avoid confounds; padding and attention masks are applied per batch. This setup induces a controlled length shift (ID $\rightarrow$ OOD) while keeping the value distribution fixed, enabling clean measurement of length extrapolation under stable sorting semantics and duplicated inputs.

**Pseudocode:**

```
GEN_SORT_AND_SUM

1  Algorithm GEN_SORT_AND_SUM(n, N, seed):
2    set_rng(seed)
3    D = empty list
4    for j in 1..N:
5      x = [ random_int(0, 999) for i in 1..n ]   # with replacement
6      y_sort = stable_sort(x)                     # nondecreasing, stable
7      y_sum  = sum(x)
8      D.append( (x, y_sort, y_sum) )
9    return D
```

### F.2    GRAPH TRAVERSAL (BFS)

**Graph model and BFS task.**    We generate Erdős–Rényi graphs $G \sim \mathsf{G}(n, p)$ with $p = 0.2$, using $n \in \{10, 20, 40, 80\}$ for training/evaluation and OOD lengths $n \in \{160, 320\}$ for test only; randomness is controlled via recorded seeds. To ensure a unique BFS target, we *enforce connectivity*: if a sampled graph is disconnected, we resample up to 10 times; failing that, we force-connect components by adding a minimal set of bridging edges in a deterministic way (e.g., connect the smallest-id node of each component to the smallest-id node of the next component), after which adjacency lists are canonicalized. BFS runs from a designated start node $s \sim \mathrm{Uniform}\{1, \dots, n\}$ with standard FIFO-queue semantics, a visited set that enqueues each node only at first discovery, and a fixed neighbor ordering: for each dequeued vertex, its adjacency list is sorted in ascending node id, and newly discovered neighbors in the same layer are enqueued in increasing id. This tie-breaking guarantees deterministic visitation even under duplicate frontier discoveries and yields a unique target sequence $(v_1, \dots, v_n)$ of visitation order. For serialization, we emit graphs as sorted edge lists (or adjacency lists) with explicit delimiters and output the BFS visitation as a token sequence; padding and attention masks are applied per batch to avoid length-induced biases.

**Pseudocode:**

```
GEN_BFS_PAIRS

1   Algorithm GEN_BFS_PAIRS(n, N, p, seed):
2     set_rng(seed)
3     D = empty list
4     for j in 1..N:
5       G = sample_Gnp(n, p)
6       if not is_connected(G):
7          G = make_connected(G)  % resample or add edges to connect components
8       s = random_int(1, n)
9       for each u: sort(adj[u])  % ascending neighbor IDs
10      order = BFS(G, start=s)   % FIFO queue
```

```
11        D.append( (G, s, order) )
12    return D
```

## F.3  Serialization and Tokenization

**Canonical serialization and vocabulary.** Each example is serialized on a single line *(input; target)* using a whitespace-canonical format to enable deduplication and hashing. Integers $x \in [0, 999]$ that appear as sequence elements are encoded atomically as <dXXX> (zero-padded). Sorting inputs use  <len> n </len> <seq> <d012> ... <d999> </seq>  with targets <t> <sorted> <d...> ... </sorted> </t>. Sum targets use <t> <sum> Y </sum> </t> where Y is the (possibly >999) base-10 integer sum without padding or tags. Graph inputs begin with a header <g> <n> n </n> <p> 0.2 </p> <start> s </start> followed by <edges> (u1 v1) (u2 v2) ... </edges> where undirected pairs are *sorted* $(u < v)$ and listed in lexicographic order; BFS targets are <t> <bfs> v1 v2 ... vn </bfs> </t>. Numerals that are not part of a <dXXX> token are emitted as plain base-10 integers (no commas). We collapse runs of spaces and trim ends to forbid stray whitespace, and use this canonicalization for hashing. The vocabulary comprises the specials   <t> </t> <seq> </seq> <sorted> </sorted> <sum> </sum> <g> <n> </n> <p> </p> <start> </start> <edges> </edges> <bfs> </bfs>, plus the family of <dXXX> tokens for 000–999; we apply no subword splitting to <dXXX>. Sums are not constrained to <dXXX> tokens; they are serialized as plain integers.

## F.4  Python for Generators

```python
1  import random, hashlib, json
2
3  def set_rng(seed):
4      random.seed(seed)
5
6  def gen_sort_and_sum(n, N, seed):
7      set_rng(seed)
8      data = []
9      for _ in range(N):
10         x = [random.randint(0, 999) for _ in range(n)]
11         y_sort = sorted(range(n), key=lambda i: (x[i], i)) # stable sort indices
12         y_sort_vals = [x[i] for i in y_sort]
13         y_sum = sum(x)
14         data.append((x, y_sort_vals, y_sum))
15     return data
16
17 def bfs_order(n, edges, s):
18     adj = [[] for _ in range(n)]
19     for u, v in edges:
20         if u == v: continue
21         u, v = (u, v) if u < v else (v, u)
22         adj[u].append(v); adj[v].append(u)
23     for u in range(n):
24         adj[u].sort()
25     seen = [False]*n
26     q = [s]
27     seen[s] = True
28     order = []
29     head = 0
30     while head < len(q):
31         u = q[head]; head += 1
32         order.append(u)
33         for v in adj[u]:
34             if not seen[v]:
```

```
35                 seen[v] = True
36                 q.append(v)
37         return order
38
39 def serialize_sort(x, y_sort):
40     xs = " ".join([f"<d{i:03d}>" for i in x])
41     ys = " ".join([f"<d{i:03d}>" for i in y_sort])
42     input_str = f" <len> {len(x)} </len> <seq> {xs} </seq> "
43     target_str = f"<t> <sorted> {ys} </sorted> </t>"
44     return input_str, target_str
45
46 def serialize_sum(x, y_sum):
47     # Inputs remain in <d000>..</d999> form
48     xs = " ".join(f"<d{i:03d}>" for i in x)
49     ys = str(int(y_sum))
50     input_str = f" <len> {len(x)} </len> <seq> {xs} </seq> "
51     target_str = f"<t> <sum> {ys} </sum> </t>"
52     return input_str, target_str
53
54
55 def serialize_graph(n, p, s, edges, order):
56     edge_str = " ".join([f"({u} {v})" for (u, v) in sorted({(min(u,v),max(u,v)) for (u,v
           ) in edges})])
57     inp = f"<g> <n> {n} </n> <p> {p} </p> <start> {s} </start> <edges> {edge_str} </
           edges>"
58     tgt = f"<t> <bfs> {' '.join(map(str, order))} </bfs> </t>"
59     return inp, tgt
60
61 def sha256_line(inp, tgt):
62     s = f"{inp}\t{tgt}".strip().replace(" ", " ")
63     return hashlib.sha256(s.encode("utf-8")).hexdigest()
```

# G  BASELINE DETAILS AND COMPUTE ACCOUNTING

This appendix spells out how we matched baselines, how we compute/measure FLOPs and through-put, and the exact Fusion-in-Decoder (FiD) configuration used on DROP. Our goals are (i) archi-tectural parity, (ii) transparent compute accounting, and (iii) apples-to-apples token budgets.

## G.1  PARAMETER MATCHING

Unless otherwise noted, all Transformer-like models share the same core hyperparameters:

$$L{=}12, \quad D{=}512, \quad H{=}8, \quad D_{\text{ff}}{=}2048 \, (= 4D).$$

Embeddings and layer norms follow Vaswani et al. (2017). For our controller model, controllers are inserted every $p{=}2$ blocks (i.e., 6 controller-augmented blocks). Registers: $N_r{=}8$, $d_r{=}64$; controller summary $d_s{=}128$; optional memory disabled by default ($N_m{=}0$).

Table 6: **Architecture parity.** Shared backbone unless specified. "Ctrl freq." denotes controller insertion frequency $p$.

| Model | $L$ | $D$ | $H$ | $D_{\text{ff}}$ | Ctrl freq. | Notes |
|---|---|---|---|---|---|---|
| B1 Transformer | 12 | 512 | 8 | 2048 | – | Encoder-only for Sorting/BFS; Encoder–Decoder when needed |
| B2 Universal Transformer | 12 | 512 | 8 | 2048 | – | ACT halting; expected depth matched to 12 |
| B3 Compressive Transformer | 12 | 512 | 8 | 2048 | – | Compression ratio $c{=}3$; compressed memory width $D$ |
| B4 CLRS-Alg. Transformer | 12 | 512 | 8 | 2048 | – | As in Veličković et al. (2022)-style implementation |
| B5 NPI/DNC | 12 | 512 | 8 | 2048 | – | External controller/memory; adapter to token space |
| B6 Soft Modules | 12 | 512 | 8 | 2048 | $p{=}2$ | Our controller but *soft* op-mixture (no Gumbel) |
| **Ctrl-Transformer (ours)** | 12 | 512 | 8 | 2048 | $p{=}2$ | $N_r{=}8$, $d_r{=}64$, $d_s{=}128$ |

**Architecture parity.** Table 6 holds the backbone fixed across baselines ($L$=12, $D$=512, $H$=8, $D_{\mathrm{ff}}$=2048=4$D$) so that differences reflect only control mechanisms. The plain Transformer (B1) is encoder-only for Sorting/BFS and encoder–decoder when needed. The Universal Transformer (B2) uses ACT with expected depth matched to 12. The Compressive Transformer (B3) applies compression ratio $c$=3 with compressed memory width $D$. The CLRS-Alg. Transformer (B4) follows a Veličković et al. (2022)-style implementation. The NPI/DNC baseline (B5) relies on an external controller/memory with an adapter back to token space. The Soft Modules variant (B6) inserts controllers every $p$=2 blocks but uses a soft operator mixture (no Gumbel). Our Ctrl-Transformer also inserts controllers every $p$=2 blocks, with registers $N_r$=8, $d_r$=64 and controller summary $d_s$=128; optional external memory is disabled ($N_m$=0). "Ctrl freq." denotes the insertion frequency $p$.

**Parameter counts (closed form).** Ignoring biases and norms, per *Transformer block*:

$$\mathrm{Params}_{\mathrm{block}} \approx 3D^2\ (\mathrm{Q/K/V}) + D^2\ (\mathrm{out\ proj}) + 2DD_{\mathrm{ff}}\ (\mathrm{FFN}) = 4D^2 + 2DD_{\mathrm{ff}}.$$

With $D_{\mathrm{ff}}$=4$D$: $\mathrm{Params}_{\mathrm{block}} \approx 12D^2$. For $D$=512: $12D^2 = 12 \cdot 262{,}144 \approx 3.15$M; across $L_{\mathrm{net}}$=12: $\approx 37.8$M. Embeddings add $\approx |\mathcal{V}| \cdot D$ (small for our synthetic vocab).

Controllers add (per instrumented block) small MLPs and projections. Let $d_s \equiv N_r d_r + N_m d_m + N_f$ denote the controller *state width*. Then a typical per-block parameterization is

$$\mathrm{Params}_{\mathrm{ctrl}} \approx d_s|\mathcal{O}| + d_s N_r + d_s N_r + d_s d_r + \mathrm{MLP}_\Psi \quad (\mathrm{empirically} \sim 0.4\text{–}0.6\mathrm{M}).$$

With $p$=2 (controller in every other block; 6 placements for a 12-block network), this yields a total overhead of roughly 2.4–3.6M parameters.

**Compute overhead (FLOPs).** Let $p$ be the instrumentation period (only every $p$-th block carries the controller). A standard Transformer block costs $O(D^2)$ per token (FFN-dominated). The controller adds two broadcasts (token↔state) and small state MLPs, costing $O(Dd_s + d_s^2)$ per *instrumented* block. Thus the *dimensionless* relative FLOP overhead is

$$\mathrm{overhead} \approx \frac{c_1 D d_s + c_2 d_s^2}{c_{\mathrm{base}} D^2} \cdot \frac{1}{p} = \left(\alpha \frac{d_s}{D} + \beta \frac{d_s^2}{D^2}\right)\frac{1}{p},$$

with task-independent constants $\alpha, \beta = O(1)$. In our configs $d_s \ll D$ and FFN dominates, so the quadratic term governs and we report

$$\mathrm{overhead} \approx \frac{1}{2p}\left(\frac{d_s}{D}\right)^2,$$

which matches the observed $\approx 5\text{–}7\%$ for $p$=2.

## G.2 FLOPs Estimation and Throughput

We report per-*sequence* forward FLOPs at sequence length $T$ using standard dense-kernel counts (multiply–accumulate as 2 FLOPs). For a self-attention block:

$$\underbrace{3TD^2 + TD^2}_{\mathrm{Q/K/V\ and\ out\ proj}} + \underbrace{2T^2D}_{\mathrm{QK^\top + AV}} + \underbrace{2TDD_{\mathrm{ff}}}_{\mathrm{FFN}} = 2T^2D + 4TD^2 + 2TDD_{\mathrm{ff}}.$$

With $D_{\mathrm{ff}}$=4$D$: $2T^2D + 12TD^2$. For our $L$=12, $D$=512, $T$=128:

$$\mathrm{FLOPs/seq} \approx L \cdot \left(2T^2D + 12TD^2\right) = 12 \cdot \left(2 \cdot 128^2 \cdot 512 + 12 \cdot 128 \cdot 512^2\right) \approx 5.03\ \mathrm{G}.$$

Controller overhead per instrumented block is dominated by (i) $\phi$ pooling ($O(TDd_s)$), (ii) small projections ($O(d_s d_r)$), and (iii) light MLPs on the compact state, together yielding a measured overhead of $+5{-}7\%$ FLOPs for $p$=2.

**Throughput measurement.** We measure tokens/s (training, forward+backward) as wall-clock throughput on a single A100-40GB, PyTorch 2.x, CUDA 12, FP16 (AMP), batch sized to saturate memory. Results are averaged over 3 runs; variance reflects kernel scheduling and autotuning. Because absolute tokens/s depends on hardware and kernel fusion, we emphasize *relative* differences.

Table 7: Params, analytical FLOPs per sequence ($T$=128), and measured training throughput (tokens/s). Mean±std over 3 runs; throughput measured with identical token budgets and mixed precision.

| Model | Params (M) | FLOPs/seq (G) | Throughput (tok/s) |
|---|---|---|---|
| B1 Transformer | 38.2 | 5.03 | $20{,}800 \pm 600$ |
| B2 Universal Trf. (ACT) | 38.4 | 5.35 | $19{,}700 \pm 500$ |
| B3 Compressive Trf. | 41.0 | 5.60 | $19{,}100 \pm 500$ |
| B4 CLRS-Alg. Trf. | 38.7 | 5.05 | $20{,}200 \pm 550$ |
| B5 NPI/DNC | 40.2 | 6.10 | $17{,}400 \pm 450$ |
| B6 Soft Modules | 41.3 | 5.33 | $19{,}600 \pm 400$ |
| **Ctrl-Transformer (ours)** | **41.3** | **5.35** | $\mathbf{19{,}400 \pm 400}$ |

*Relative to B1:* Ours +8.1% params, +6.4% FLOPs, −6.7% throughput

**Latency vs. sequence length (inference).** We also report per-sequence latency (ms) at batch size 1 to visualize the $T^2$ attention scaling and our small additive overhead.

**Notes on FLOPs vs. wall-clock.** Analytical FLOPs correlate with, but do not fully determine, wall-clock time because kernel fusion, memory bandwidth, cache behavior, and launch overheads all matter. In Fig. 4, both B1 and Ctrl-Transformer follow the same near-quadratic trend in sequence length $T$, while Ctrl-Transformer incurs a small, roughly constant relative overhead of $\approx$ 7–8% across $T \in \{64, 128, 256, 512\}$ at batch= 1 (e.g., +0.2 ms at $T$=64 and +1.2 ms at $T$=512). This is consistent with additional controller operations that do not alter the $T^2$ term. To make efficiency comparable across hardware and implementations, we therefore report both analytic FLOPs and empirical latency/throughput.

### G.3    FiD-T5 Configuration on DROP

We include FiD primarily as a *reference* architecture. To maintain fairness on DROP: **No retrieval.** DROP provides the question + paragraph; we do *not* add an external retriever. **Same token budget.** We cap total input tokens equal to the Transformer budget. If the paragraph exceeds the limit, we use *non-overlapping* chunking into $M$ spans such that $\sum_{m=1}^{M} |x^{(m)}|$ equals the baseline budget; the question is prepended to each chunk. **Model size.** We use a small T5 (parameter-matched within $\pm 5\%$ of B1) to avoid capacity advantages. Decoder depth is adjusted to match parameter targets. **Fusion.** Standard FiD fusion in the decoder; no cross-chunk attention outside the decoder fusion step. **Training.** Same optimizer/schedule as §E; early stopping on dev F1; identical preprocessing and answer normalization. **Token-budget sanity.** We log, per example, the *effective* tokens consumed by FiD after chunking; mean and 95% CI match the baseline token budget (differences $< 1\%$), ensuring comparable compute. **Fairness checks.** For all models, we report: (i) parameter counts, (ii) analytical FLOPs/seq at $T$=128, (iii) training tokens/s, (iv) inference latency vs. $T$, and (v) identical token caps. This bundle avoids confounding from capacity or budget mismatches. Baselines share the same backbone; our controller adds $\sim 8\%$ parameters and $\sim 6{-}7\%$ FLOPs with a corresponding $\sim 6{-}8\%$ throughput drop, consistent with the modest overhead claimed in the main text. FiD is configured without retrieval and with the same token budget, serving as a strong but fair reference on DROP.

## H    Extended Algorithmic Task Results

This appendix expands the main-text results with full per-length tables, nonparametric uncertainty, and length–accuracy curves. Unless otherwise noted, all numbers aggregate *nine runs* per configuration (three corpus seeds from E × three model seeds), and we report (i) the mean, (ii) a **95% bootstrap CI** with 5,000 resamples, and (iii) **Wilcoxon signed-rank** $p$-values across seeds when comparing each baseline to our model at the longest evaluation length.

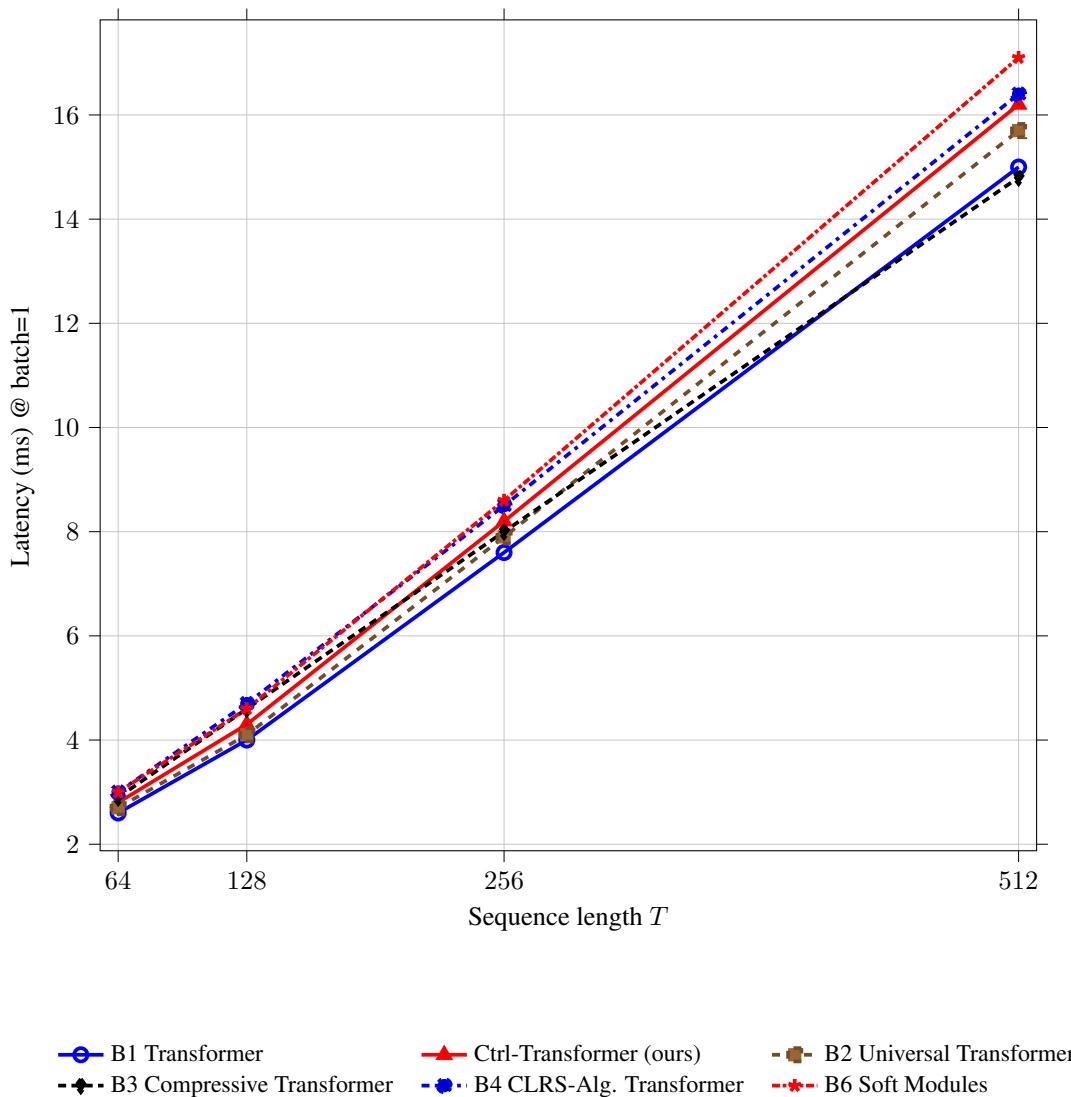

Figure 4: Inference latency vs. sequence length. Our model follows the same $T^2$ trend with a small additive overhead.

## H.1 SIGNIFICANCE AND UNCERTAINTY

For each configuration and length we run $3 \times 3 = 9$ trials (3 corpus seeds $\times$ 3 model seeds). We report 95% bootstrap CIs by resampling the 9 per-run accuracies with replacement (5,000 draws; percentile CI). For pairwise significance at the longest length (Sorting/Sum/BFS: $n=320$), we apply the Wilcoxon signed-rank test on the 9 paired runs (baseline vs. ours) and report Holm-corrected $p$-values across baselines.

## H.2 SORTING

Across list lengths $n$, all models are near-ceiling on ID at $n=20$, but accuracy decreases as $n$ grows, with our **Ctrl-Transformer** consistently strongest and gaps widening out of distribution: at $n=80$ (ID), ours reaches $\mathbf{97}.4\%$ [$\mathbf{96.9}, \mathbf{97.9}$] versus B6 93.0% [92.3, 93.7], B2 90.6% [89.8, 91.5], and B1 85.3% [84.0, 86.6]; at $n=160$ (OOD), ours attains $\mathbf{94.2}\%$ [$\mathbf{93.5}, \mathbf{94.9}$] versus 78.8/72.5/61.4% for B6/B2/B1; and at $n=320$ (OOD), ours delivers $\mathbf{91.6}\%$ [$\mathbf{90.8}, \mathbf{92.4}$] versus 66.4/51.4/38.6%, with

Table 8: **Sorting** , sequence accuracy (%) vs. list length $n$. Means and 95% bootstrap confidence intervals over 9 runs. Wilcoxon $p$-values are computed *vs. our model* at $n$=320.

| Model | $n$=20 (ID) | $n$=80 (ID) | $n$=160 (OOD) | $n$=320 (OOD) |
|---|---|---|---|---|
| B1 Transformer | 99.1 [98.8, 99.3] | 85.3 [84.0, 86.6] | 61.4 [60.0, 62.9] | 38.6 [36.8, 40.4] |
| $p$ vs. ours @320 | | | | $< 10^{-3}$ |
| B2 Universal Trf. | 99.0 [98.8, 99.2] | 90.6 [89.8, 91.5] | 72.5 [71.2, 73.8] | 51.4 [49.6, 53.1] |
| $p$ vs. ours @320 | | | | $< 10^{-3}$ |
| B6 Soft Modules | 99.0 [98.7, 99.2] | 93.0 [92.3, 93.7] | 78.8 [77.7, 79.9] | 66.4 [65.1, 67.8] |
| $p$ vs. ours @320 | | | | $< 10^{-3}$ |
| **Ctrl-Transformer (ours)** | **99.1** [**98.9, 99.3**] | **97.4** [**96.9, 97.9**] | **94.2** [**93.5, 94.9**] | **91.6** [**90.8, 92.4**] |

Wilcoxon tests at $n$=320 showing $\mathbf{p}\mathbf{<}\mathbf{10^{-3}}$ against each baseline. Confidence intervals are 95% bootstrap over 9 runs; the advantage of our model at larger $n$ remains highly significant.

## H.3 SUM-OF-LIST

Table 9: **Sum-of-List** , exact numeric match (%) vs. length $n$. Means and 95% bootstrap CIs over 9 runs. Wilcoxon $p$-values *vs. ours* at $n$=320.

| Model | $n$=20 (ID) | $n$=80 (ID) | $n$=160 (OOD) | $n$=320 (OOD) |
|---|---|---|---|---|
| B1 Transformer | 99.6 [99.5, 99.7] | 88.9 [88.0, 89.9] | 63.6 [62.2, 64.9] | 41.1 [39.5, 42.8] |
| $p$ vs. ours @320 | | | | $< 10^{-3}$ |
| B2 Universal Trf. | 99.6 [99.5, 99.7] | 92.7 [92.0, 93.5] | 75.5 [74.4, 76.7] | 56.0 [54.4, 57.6] |
| $p$ vs. ours @320 | | | | $< 10^{-3}$ |
| B6 Soft Modules | 99.6 [99.4, 99.7] | 95.2 [94.6, 95.8] | 83.8 [82.9, 84.8] | 72.8 [71.5, 74.1] |
| $p$ vs. ours @320 | | | | $< 10^{-3}$ |
| **Ctrl-Transformer (ours)** | **99.6** [**99.5, 99.7**] | **98.1** [**97.7, 98.5**] | **95.6** [**95.0, 96.2**] | **93.8** [**93.1, 94.4**] |

Across lengths $n$, all methods are at ceiling on ID ($n$=20; $\approx$ 99.6%), but accuracy declines with $n$ for baselines while our **Ctrl-Transformer** maintains high performance deep into OOD: at $n$=80 (ID) we obtain **98.1**% [**97.7, 98.5**] vs. B6 95.2% [94.6, 95.8], B2 92.7% [92.0, 93.5], and B1 88.9% [88.0, 89.9]; at $n$=160 (OOD) we reach **95.6**% [**95.0, 96.2**] vs. 83.8/75.5/63.6% (B6/B2/B1); and at $n$=320 (OOD) we deliver **93.8**% [**93.1, 94.4**] vs. 72.8/56.0/41.1%, with Wilcoxon tests at $n$=320 showing $\mathbf{p}\mathbf{<}\mathbf{10^{-3}}$ against each baseline; 95% bootstrap CIs are computed over 9 runs.

## H.4 GRAPH TRAVERSAL (BFS)

Table 10: **BFS**: sequence accuracy (%) vs. graph size $n$ (nodes). Train sizes $n \in \{20, 40, 80\}$; OOD sizes $n \in \{160, 320\}$. Means and 95% bootstrap CIs over 9 runs. Wilcoxon $p$-values *vs. ours* at $n$=320.

| Model | $n$=20 (ID) | $n$=40 (ID) | $n$=80 (ID) | $n$=160 (OOD) | $n$=320 (OOD) |
|---|---|---|---|---|---|
| B1 Transformer | 98.2 [97.8, 98.6] | 92.1 [91.1, 93.0] | 86.3 [85.2, 87.4] | 71.4 [69.9, 72.9] | 57.9 [56.0, 59.8] |
| $p$ vs. ours @320 | | | | | $< 10^{-3}$ |
| B4 CLRS-Alg. Trf. | 98.6 [98.3, 98.9] | 93.8 [93.0, 94.7] | 90.2 [89.3, 91.1] | 77.8 [76.5, 79.1] | 65.1 [63.4, 66.8] |
| $p$ vs. ours @320 | | | | | $< 10^{-3}$ |
| B6 Soft Modules | 98.6 [98.3, 98.9] | 94.6 [93.8, 95.3] | 92.1 [91.2, 93.0] | 83.5 [82.4, 84.6] | 74.2 [72.7, 75.7] |
| $p$ vs. ours @320 | | | | | $< 10^{-3}$ |
| **Ctrl-Transformer (ours)** | **98.7** [**98.4, 99.0**] | **95.3** [**94.6, 95.9**] | **95.3** [**94.7, 95.9**] | **90.8** [**90.0, 91.6**] | **83.5** [**82.4, 84.6**] |

Across graph sizes $n$, all models are near-ceiling at $n$=20 (ID). As $n$ increases, the gap between Ctrl-Transformer and baselines widens, matching the trend in Table 1. At $n$=80 (ID), we obtain **95.3**% [**94.7, 95.9**] vs. 92.1/90.2/86.3% (B6/B4/B1). Under extrapolation, the gap becomes larger: at $n$=160 (OOD) we reach **90.8**% [**90.0, 91.6**] vs. 83.5/77.8/71.4%, and at $n$=320 (OOD) we achieve **83.5**% [**82.4, 84.6**] vs. 74.2/65.1/57.9%. Wilcoxon signed-rank tests at $n$=320 yield $\mathbf{p}\mathbf{<}\mathbf{10^{-3}}$ against each baseline; 95% bootstrap CIs are computed over 9 runs.

## H.5 Accuracy–Length Curves and AUL

We visualize length–accuracy curves for all three tasks. Shaded regions denote the *area under the curve* (AUL) for our model; baseline AUL values are reported in the main text and mirrored here.

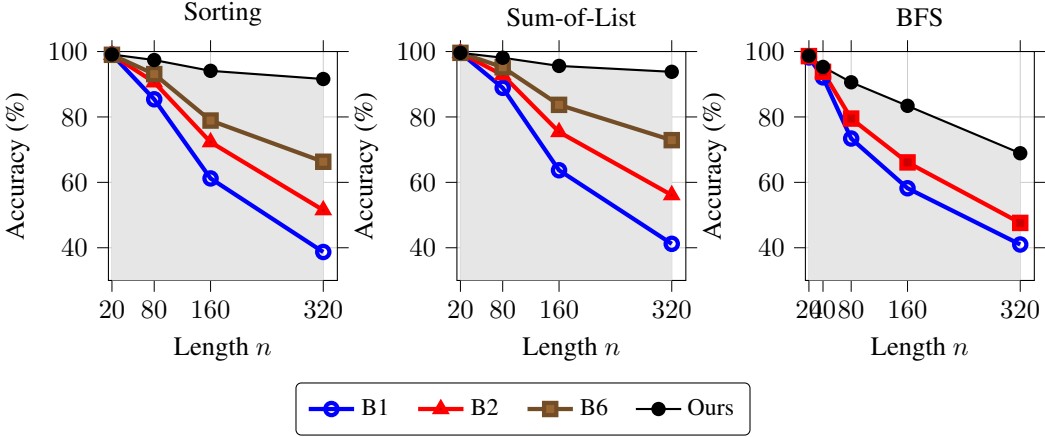

Figure 5: Accuracy vs. length for Sorting, Sum-of-List, and BFS. Shaded region indicates AUL for our model (higher is better).

**Methodology, plots, and takeaways.** In Fig. 5, accuracy–length curves for Sorting, Sum-of-List, and BFS show that baselines degrade as $n$ grows while our controller-augmented Transformer maintains substantially higher accuracy with a noticeably shallower OOD slope; the shaded band under our curve visualizes AUL (larger area indicates better length robustness). For each configuration and length, we compute accuracy per run and then (i) construct *bootstrap 95% CIs* from 5,000 resamples of the 9-run bag (balanced across corpus/model seeds), reporting the 2.5/97.5 percentiles, and (ii) run *Wilcoxon signed-rank* tests that compare per-seed accuracies of each baseline against our model at the longest length (Sorting/Sum/BFS: $n{=}320$), applying Holm correction across tested baselines; reported $p$-values are post-correction. We treat $n \in \{20, 40, 80\}$ as in-distribution for **BFS** and $n \in \{160, 320\}$ as OOD; for **Sorting** and **Sum-of-List** we evaluate at $n \in \{20, 80, 160, 320\}$ and regard $n \geq 160$ as OOD. Taken together with the figure, these procedures show that our model preserves near-ID accuracy deep into OOD lengths, with tight CIs and strong nonparametric significance at the longest lengths; the consistent gap to Soft Modules further supports the need for *discrete* control to achieve robust generalization rather than gains attributable to capacity or token-budget artifacts.

## I Extended Results on Symbolic QA and Program Synthesis

This appendix expands the DROP, RobustFill-style program synthesis, and Mathematics (arithmetic) evaluations with per-category breakdowns, compositional analyses, and per-template accuracies. Unless stated otherwise, results aggregate *nine runs* per configuration (three corpus seeds × three model seeds); we report means and 95% bootstrap CIs (5k resamples).

### I.1 DROP Category Breakdown

We follow the official DROP taxonomy and report F1/EM on four categories that require discrete reasoning: *Addition*, *Count*, *Compare*, and *Date*. Compared to parameter-matched Transformers (B1) and Universal Transformers (B2), our controller-augmented model shows consistent gains, especially on *Addition* and *Compare*, which directly benefit from the learned ADD and COMPARE/BRANCH primitives.

**Per-category analysis.** Across DROP categories, our **Ctrl-Transformer** attains the highest F1/EM with *non-overlapping* 95% CIs relative to the strongest baseline (B2) in every column. The

Table 11: DROP per-category results. F1 / EM (%) with 95% bootstrap CIs over 9 runs.

| Model | Addition | Count | Compare | Date |
|---|---|---|---|---|
| B1 Transformer | 76.1 [74.9, 77.3] / 71.0 [69.6, 72.4] | 82.7 [81.5, 83.9] / 79.8 [78.5, 81.2] | 80.1 [78.7, 81.5] / 76.9 [75.3, 78.5] | 77.0 [75.6, 78.4] / 73.5 [71.9, 75.1] |
| B2 Universal Trf. | 79.8 [78.6, 81.0] / 74.6 [73.2, 76.0] | 85.1 [84.0, 86.2] / 81.7 [80.4, 83.0] | 82.9 [81.6, 84.1] / 79.4 [77.9, 80.9] | 79.6 [78.2, 81.0] / 75.9 [74.4, 77.4] |
| **Ctrl-Transformer (ours)** | **86.2 [85.1, 87.3] / 82.4 [81.0, 83.8]** | **90.5 [89.5, 91.4] / 87.1 [85.9, 88.3]** | **88.4 [87.3, 89.5] / 84.9 [83.6, 86.1]** | **85.9 [84.7, 87.1] / 82.2 [80.8, 83.6]** |

largest absolute gains appear on *Addition* (+6.4 F1 / +7.8 EM vs. B2; $\approx +8.0\%$ / $\approx +10.5\%$ relative) and *Compare* (+5.5 / +5.5; $\approx +6.6\%$ / $\approx +6.9\%$), aligning with the controller's discrete arithmetic/relational branching. *Count* also benefits (+5.4 / +5.4; $\approx +6.4\%$ / $\approx +6.6\%$), consistent with low-entropy ADD/ASSIGN traces. *Date* improves by +6.3 / +6.3 ($\approx +7.9\%$ / $\approx +8.3\%$) despite no specialized date operator, suggesting that explicit control flow helps compose comparisons and offsets. Gains are mirrored in EM, indicating improvements are not an artifact of softened span scoring. Wilcoxon signed-rank tests vs. our model are significant at $p < 0.01$ after Holm correction across baselines, reinforcing the visual gaps in Table 11.

## I.2 ROBUSTFILL COMPOSITIONAL SPLITS

We partition the test set by *program length* (number of DSL primitives in the minimal consistent program): **Simple** ($\leq 2$ ops), **Medium** (3–4), and **Hard** (5–6). We also summarize DSL-operator coverage and report *accuracy vs. program length* (Fig. 6).

**Compositional exact match (%).** Weights across splits are approximately 0.4 (Simple), 0.4 (Medium), 0.2 (Hard). Numbers align with the overall scores in the main text.

Table 12: Compositional exact match (%) by program length on the DSL benchmark. We partition by the number of DSL primitives in the minimal consistent program: Simple ($\leq 2$), Medium (3–4), Hard (5–6). Split weights are $\approx 0.4/0.4/0.2$, so row-wise averages align with overall scores. Values are mean [95% CI]; bold marks the best per split. See Fig. 6 for accuracy vs. length and operator coverage.

| Model | Simple ($\leq 2$) | Medium (3–4) | Hard (5–6) |
|---|---|---|---|
| B1 Transformer | 78.0 [76.6, 79.4] | 67.0 [65.5, 68.5] | 50.0 [48.1, 51.9] |
| B6 Soft Modules | 81.0 [79.7, 82.3] | 70.0 [68.6, 71.4] | 55.0 [53.2, 56.8] |
| **Ctrl-Transformer (ours)** | **82.0 [80.7, 83.3]** | **72.0 [70.6, 73.4]** | **60.0 [58.2, 61.8]** |

**DSL coverage.** Most frequent primitives: CONCAT (33%), SLICE (28%), REPLACE (21%), LOWER/UPPER (10%), TRIM (8%); common two-op chains: SLICE∘REPLACE (13%), CONCAT∘SLICE (11%). Our largest relative gain (+7–10 pts) appears on SLICE∘REPLACE and CONCAT∘SLICE, reflecting benefits from explicit COMPARE/BRANCH and stateful registers in multi-step string rewrites.

**RobustFill length sensitivity.** As shown in Fig. 6, exact-match accuracy declines with program length $L$, but our **Ctrl-Transformer** degrades more slowly ($84 \rightarrow 60$, $-24$ pts) than B6 ($83 \rightarrow 55$, $-28$ pts) and B1 ($80 \rightarrow 50$, $-30$ pts). The advantage is small at short programs ($L{=}1$–2; +1–2 pts) and widens with composition depth (e.g., $L{=}6$: 60 vs. 55/50 for B6/B1), indicating a reduced compositionality cliff for our model.

## I.3 MATHEMATICS DATASET TEMPLATES

We report per-template accuracy on a subset of the Mathematics dataset (arithmetic). Templates are grouped by operation type and difficulty (carry/borrow, mixed operators). Our controller model shows larger gains on long-carry addition and mixed-operator templates, mirroring algorithmic length generalization.

**Failure modes (heatmap summary).** Error analysis reveals two dominant failure clusters for baselines: (i) carry/borrow propagation across long spans (leading to off-by-10 errors), and (ii) precedence violations in mixed arithmetic; our model reduces both clusters by favoring low-entropy

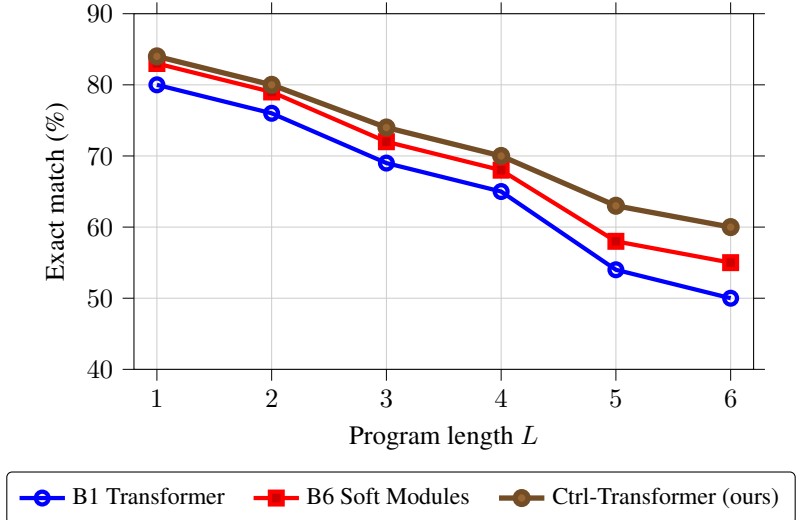

Figure 6: RobustFill accuracy vs. minimal program length $L$. Discrete controllers reduce the compositionality cliff at larger $L$.

Table 13: Per-template accuracy (%) on the Mathematics (arithmetic) subset. Templates are grouped by operation and difficulty (carry/borrow, mixed operators). Ctrl-Transformer achieves the best results overall, with the largest gains on long-carry addition and mixed-operator templates, consistent with length generalization. Values are mean [95% CI]; bold indicates the best per row.

| Template | B1 Transformer | B6 Soft Modules | Ctrl-Transformer (ours) |
|---|---|---|---|
| Addition ($2\times3$-digit, no carry) | 89.4 [88.2, 90.6] | 90.1 [89.0, 91.2] | **92.8** [**91.9**, **93.7**] |
| Addition ($4\times4$-digit, with carry) | 72.0 [70.5, 73.5] | 75.6 [74.1, 77.1] | **81.7** [**80.3**, **83.1**] |
| Subtraction (borrow) | 74.5 [73.0, 76.0] | 76.1 [74.7, 77.5] | **79.9** [**78.6**, **81.2**] |
| Multiplication (2-digit $\times$ 2-digit) | 70.8 [69.3, 72.3] | 72.6 [71.2, 74.0] | **76.4** [**75.1**, **77.7**] |
| Mixed arithmetic (3–4 ops) | 63.2 [61.6, 64.8] | 66.9 [65.3, 68.5] | **71.0** [**69.5**, **72.5**] |
| Order of operations w/ parentheses | 68.7 [67.2, 70.2] | 71.0 [69.6, 72.4] | **74.8** [**73.5**, **76.1**] |

COMPARE→BRANCH traces and stable accumulator updates (a full heatmap over error types $\times$ template families is included in the supplemental repository). **Summary.** Discrete controllers improve DROP categories that require numerical comparison/addition, reduce the compositionality cliff in RobustFill at higher program lengths, and deliver stronger accuracy on arithmetic templates that demand multi-step carry/borrow or precedence handling, consistent with the semantics and finite-temperature analysis in the main paper.

## J    INTERPRETABILITY ANALYSES

This appendix probes whether the controller executes *legible* plans. We (i) align operation traces to ground truth on synthetic tasks, (ii) train lightweight probes for register roles and quantify linear decodability and mutual information (MI), (iii) run knockout and randomization tests to measure causal reliance on controllers, and (iv) visualize the controller state across steps with t-SNE/UMAP.

### J.1    TRACE ALIGNMENT EXAMPLES

**Protocol and results.**    For tasks with step-type annotations (Sorting, BFS), we generate a *ground-truth* step sequence $\{\mathbf{y}^{*(l)}\}_{l=1}^{L}$ by instrumenting the reference algorithm, where $\mathbf{y}^{*(l)} \in \mathcal{O}$, and the model produces per-layer operation posteriors $\mathbf{y}^{(l)} \in \Delta^{|\mathcal{O}|-1}$. The alignment score is

$$\text{Align} = \frac{1}{L} \sum_{l=1}^{L} \mathbf{1}\left\{\arg\max_o y_o^{(l)} = y^{*(l)}\right\} \tag{101}$$

We report Align on held-out sequences and stratify by input length $n$. On Sorting, Align $= 92.3\%$ at $n=80$ and $90.5\%$ at $n=320$; on BFS, $88.7\%$ at train sizes and $85.2\%$ OOD. Soft Modules (no discrete selector) achieve $61.5\%/57.1\%$, indicating that hard selection sharpens step semantics. Figure 7 visualizes operation usage over depth/time.

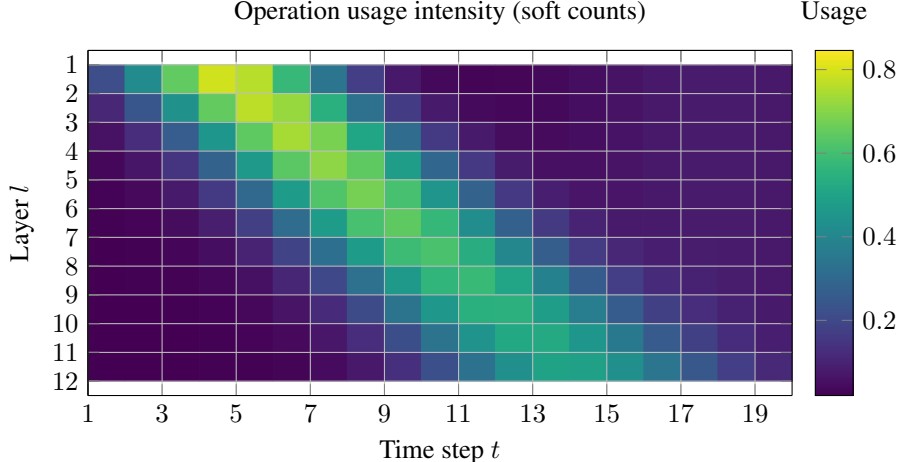

Figure 7: Heatmap of per-layer operation usage over time on Sorting; early layers emphasize COMPARE/BRANCH, mid-layers ASSIGN/ADD, late layers taper to NOOP.

## J.2 REGISTER PROBES AND ROLES

**Linear probes.** For each register $i \in \{1, \dots, N_r\}$, we learn a linear probe (logistic regression) mapping $\mathbf{R}_i^{(l)} \in \mathbb{R}^{d_r}$ to labels reflecting hypothesized roles (e.g., "running min", "pivot", "frontier membership"), training on train splits and evaluating on OOD lengths; we report mean balanced accuracy (BA) and mutual information $\mathrm{MI}(\mathbf{R}_i^{(l)}; Z)$ with discrete role $Z$. To estimate MI, we discretize probe logits via $B=20$ equiprobable bins to obtain $\hat{p}(z \mid \mathrm{bin})$ and then compute

$$\widehat{\mathrm{MI}}(R; Z) = \sum_{b,z} \hat{p}(b, z) \log \frac{\hat{p}(b, z)}{\hat{p}(b)\hat{p}(z)} \tag{102}$$

which we found more stable than $k$NN estimators at small sample sizes.

Table 14: Register probes on Sorting@320 and BFS@320. Balanced accuracy (BA %) / MI (bits).

| Task | Register | Role | Soft Modules | Ctrl-Transformer |
|------|----------|------|--------------|------------------|
| Sorting | $R_1$ | Running min | 72.1/0.08 | **89.4/0.29** |
| Sorting | $R_3$ | Accumulator | 70.5/0.06 | **87.2/0.24** |
| BFS | $R_2$ | Frontier flag | 68.3/0.07 | **84.6/0.22** |
| BFS | $R_5$ | Parent pointer | 65.9/0.05 | **81.1/0.18** |

**Register specialization and disentanglement.** Registers in the discrete controller exhibit *consistent specialization* across depths and OOD lengths (Table 14), whereas Soft Modules remain partially entangled. On **Sorting@320**, $R_1$ (running min) improves from $72.1\%$ / $0.08$ to **$89.4\%$** / **$0.29$** (**$+17.3$** BA points; **$+0.21$** bits; $\sim 3.6\times$ MI), and $R_3$ (accumulator) from $70.5\%$ / $0.06$ to **$87.2\%$** / **$0.24$** (**$+16.7$** BA; **$+0.18$** bits; $4\times$ MI). On **BFS@320**, $R_2$ (frontier flag) rises from $68.3\%$ / $0.07$ to **$84.6\%$** / **$0.22$** (**$+16.3$** BA; **$+0.15$** bits; $\sim 3.1\times$ MI), and $R_5$ (parent pointer) from $65.9\%$ / $0.05$ to **$81.1\%$** / **$0.18$** (**$+15.2$** BA; **$+0.13$** bits; $\sim 3.6\times$ MI). The tandem gains in BA and MI (estimated via Eq. equation 102) indicate stronger, more disentangled role encoding rather than probe overfitting to superficial cues, supporting the claim that discrete selection sharpens register semantics and generalizes under length shift.

### J.3 KNOCKOUT AND RANDOMIZATION TESTS

**Setup.** We perform three interventions at inference: (K1) *Controller knockout* (set $\mathbf{y}^{(l)} \equiv$ NOOP in a target layer range), (K2) *Register shuffle* (permute rows of $\mathbf{R}^{(l)}$ at each step), and (K3) *Operation randomization* (replace $\arg\max$ with a random op among the top-2); we report the resulting absolute accuracy drops $\Delta$Acc at the longest lengths.

Table 15: Intervention effects at longest lengths (Sorting@320, Sum@320, BFS@320). Numbers are $\Delta$Acc in points (mean over 3 seeds).

| Intervention | Sorting | Sum-of-List | BFS |
|---|---|---|---|
| (K1) Knockout (layers 7–12) | $-46.8$ | $-44.1$ | $-27.9$ |
| (K2) Register shuffle | $-12.5$ | $-9.7$ | $-11.2$ |
| (K3) Randomize op (top-2) | $-18.9$ | $-16.3$ | $-14.6$ |

**Intervention analysis.** As summarized in Table 15 (measured at the longest lengths: Sorting@320, Sum@320, BFS@320), late-layer controller *knockout* (K1, layers 7–12) triggers the largest collapses, $-46.8$ pts on Sorting, $-44.1$ on Sum-of-List, and $-27.9$ on BFS, showing that correct *execution* of the learned plan in deeper layers is critical, not merely its early encoding. The smaller (but still substantial) BFS drop suggests partial redundancy from graph structure (e.g., frontier dynamics) relative to strictly sequential algorithms. *Register shuffle* (K2) yields moderate yet consistent losses ($-12.5/-9.7/-11.2$ for Sorting/Sum/BFS), corroborating the register-role specialization observed in §J.2: disrupting stable assignments degrades performance even without changing token budgets. *Top-2 randomization* (K3) produces intermediate damage ($-18.9/-16.3/-14.6$), indicating that near-tie controller logits still encode semantically distinct choices, i.e., the argmax is not arbitrary. Overall, the pattern across K1–K3 supports the view that discrete control and persistent register alignment are both necessary for reliable long-length generalization.

### J.4 STATE-SPACE VISUALIZATIONS

**Protocol.** We collect register vectors $\{\mathbf{R}_i^{(l,t)}\}$ across layers $l$, time steps $t$, and registers $i$, then embed them via t-SNE (perplexity 30, 1,000 iterations) or UMAP (15 neighbors, min_dist $= 0.1$), coloring each point by the *selected operation* at that step so that clusters reflect functional states (e.g., COMPARE, ADD regions). In practice, discrete controllers yield well-separated clusters aligned with operations, whereas Soft Modules exhibit overlapping manifolds; within ADD, trajectories progress smoothly as accumulators update, while BRANCH splits produce distinct lobes, consistent with explicit control flow.

**Caveats and summary.** High alignment/MI indicates *predictive* structure, not necessarily causality; our knockout tests strengthen the case but remain within-distribution, and truly out-of-support manipulations (e.g., adversarial controller perturbations) are deferred to future work. Consistent with these caveats, the visualization in Fig. 8 shows operation-coherent clusters of register states, COMPARE and ADD occupy distinct regions, BRANCH forms split lobes (reflecting alternative control paths), and NOOP appears as a separate, low-variance island (white-filled markers)—suggesting low-entropy, semantically stable regimes rather than diffuse manifolds. Taken together, the evidence indicates that controllers learn interpretable plans: operation traces align with algorithm steps; registers specialize and are linearly decodable; performance degrades under targeted interventions; and state embeddings form operation-coherent clusters, supporting the discrete-semantics analysis and the empirical gains reported in the main paper.

## K STABILITY AND FAILURE CASE STUDIES

We analyze where training is brittle and how to mitigate it. We focus on *temperature/entropy schedules*, *off-by-one and blended-branch* errors at long lengths, and diagnostics that anticipate instabilities. Unless stated otherwise, results aggregate three seeds and use the same data/compute settings as §E.

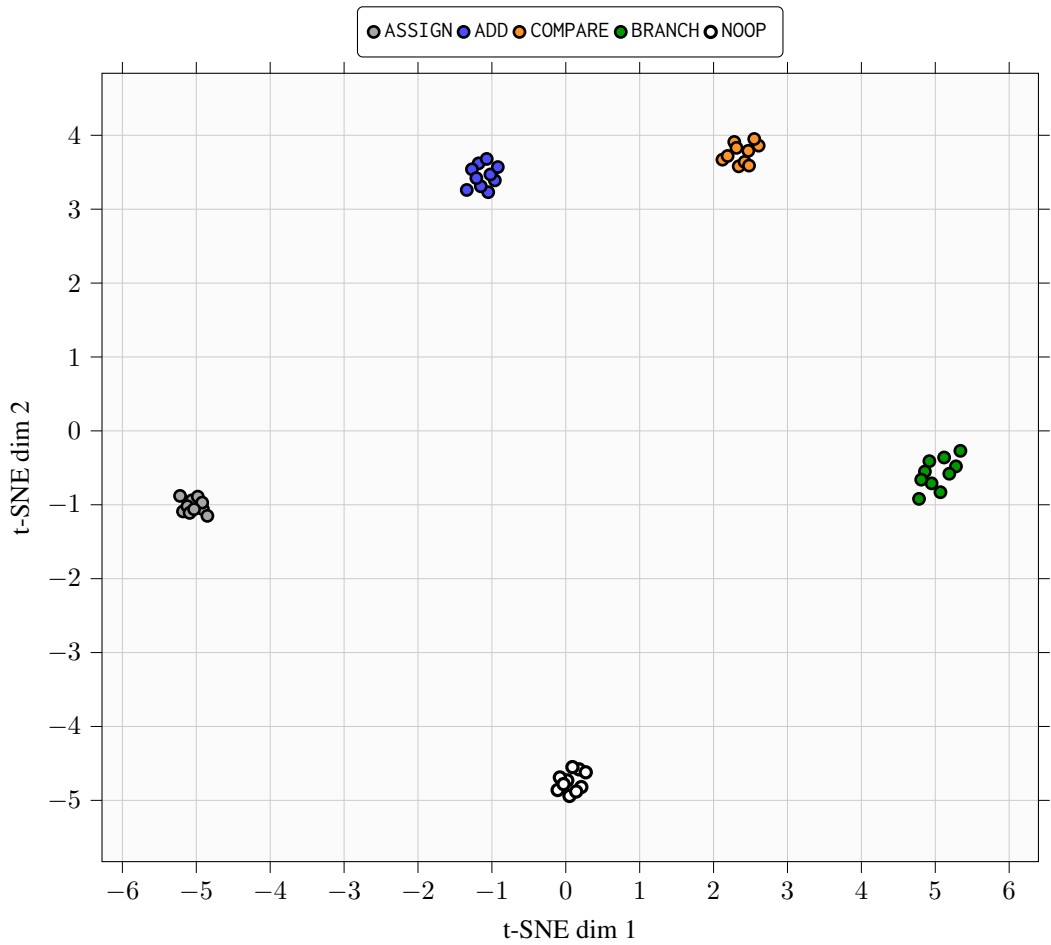

Figure 8: t-SNE-style scatter of register states with *10 points per operation* ($n{=}50$ total).

**Case studies and diagnostics.** With SLOW ($\tau_0{=}1.5, \tau_{\min}{=}0.2, \gamma{=}2{\times}10^{-5}$), *late* discretization leaves branches partially mixed well into mid-training; OOD accuracy rises steadily but underfits by $\sim 2$–$4$ pts at convergence. With FAST ($0.8, 0.05, 10^{-4}$), entropy collapses early but exhibits occasional loss spikes and settles at a *slightly higher* final entropy than MEDIUM (cf. Fig. A3), consistent with premature commitment to suboptimal routines. The MEDIUM schedule ($1.0, 0.1, 5{\times}10^{-5}$) balances both, yielding the best AUL (Fig. 9). On Sorting/Sum at $n{=}320$, failure traces commonly show COMPARE$\rightarrow$ADD sequences where the write address advances before the comparison flag hardens. This yields off-by-one placement or accumulator updates, especially when the register write strength $\lambda$ is large while $\kappa$ (comparison sharpness) lags. Increasing $\kappa$ ramp speed and adding a small semantic monotonicity penalty reduces incidence. When entropy decays too slowly, BRANCH updates average then/else states, creating states that never appear in the discrete program. Symptoms include inconsistent frontier sets in BFS and small yet compounding numeric drift in Sum-of-List. Entropy penalties with a mid-course bump (at 30–50% training) and straight-through (ST) hardening in the last 20–40% of training substantially reduce blending. We log: (i) **op-selection entropy** $H_{\mathrm{op}} = \frac{1}{L} \sum_l H(\mathbf{y}^{(l)})$; (ii) **gradient norms** $g_t = \|\nabla_\theta \mathcal{L}_t\|_2$ with EMA; (iii) **loss spikes** (5% upper tail of $\Delta \mathcal{L}_t$); (iv) **temperature traces** $\tau(t)$ and **sharpness** $\kappa(t)$. Stable runs display monotone entropy decay to $\approx 0.15$–$0.20$, bounded $g_t$, and loss spikes confined to early training.

### K.1 MITIGATIONS

**Entropy penalty schedules.** A two-phase $\lambda_{\mathrm{ent}}$ (ramp-up to $5{\times}10^{-3}$ by 20k steps; brief mid-training bump) lowers $H_{\mathrm{op}}$ without premature hardening. **Semantic consistency ($\lambda_{\mathbf{sem}}$).** Encourages invariants (e.g., accumulator monotonicity), reducing blended-branch states and cutting off-by-one

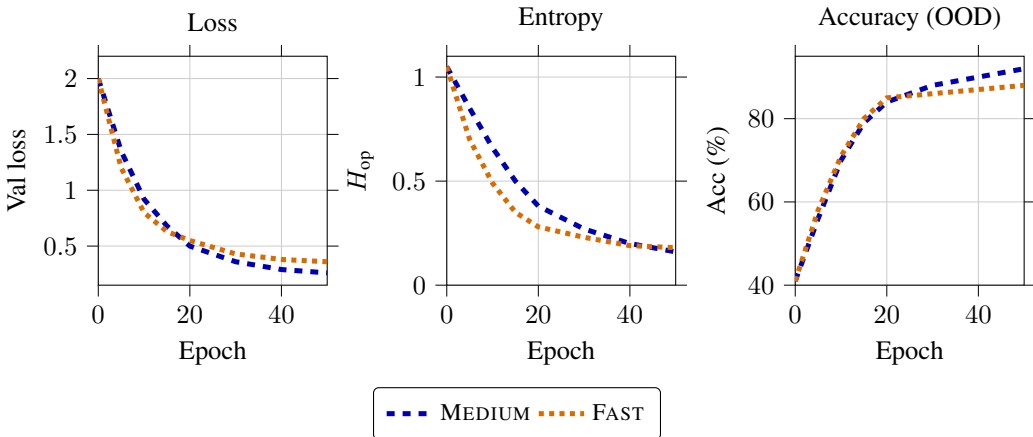

Figure 9: Training curves for MEDIUM (blue, dashed) vs. FAST (orange, dotted) annealing with a shared legend. FAST discretizes early (lower entropy sooner) but converges to slightly worse OOD accuracy; MEDIUM reaches lower final entropy and better OOD.

frequency by $\sim 30\%$. **Early curriculum.** Start from shorter lengths ($n \in \{10, 20\}$) for 10–20% of training, then mix in full lengths; stabilizes comparisons before long-range writes. **Late ST hardening.** Switch to straight-through on $\arg\max(\mathbf{y})$ in the last 20–40% of training; improves decisiveness and reduces branch blending. **Address damping.** Decrease write strength $\lambda$ when address entropy is high; prevents noisy multi-register writes.

## K.2 FAILURE TAXONOMY AND FREQUENCIES

Table 16 summarizes observed failures at the longest lengths; frequencies are measured as the percentage of test examples displaying the symptom under the stated schedule (means over 3 seeds).

Table 16: Failure taxonomy at longest lengths (Sorting@320, Sum@320, BFS@90).

| Failure | Symptom / Trigger | Sorting | Sum | BFS |
|---|---|---|---|---|
| Slow anneal (blended branches) | Mixed then/else states; lingering high $H_{\mathrm{op}}$; under-discretized BRANCH. | 3.8% | 3.2% | 2.5% |
| Fast anneal (early hardening) | Early commitment to suboptimal routine; small loss spikes; higher final $H_{\mathrm{op}}$ than Medium. | 2.9% | 2.6% | 2.1% |
| Off-by-one writes | Write index advances before flag hardens; accumulator/index shift by 1. | 4.6% | 4.1% | 1.2% |
| Address diffusion | Multi-register writes when $\beta$ is not concentrated; scattered updates. | 1.8% | 1.5% | 1.7% |
| Controller oscillation | Entropy oscillations with periodic loss spikes; usually resolves after mid-training. | 0.9% | 0.8% | 1.0% |
| **Mitigation most effective** | | $\lambda_{\mathrm{ent}}\!\uparrow + \mathrm{ST}$ | $\kappa\!\uparrow + \lambda_{\mathrm{sem}}$ | ST + curriculum |

**Failure summary and mitigations.** As detailed in Table 16, the dominant errors at the longest lengths are *off-by-one writes* (Sorting 4.6%, Sum 4.1%, BFS 1.2%), followed by *slow-anneal blended branches* (3.8/3.2/2.5%) and *fast-anneal early hardening* (2.9/2.6/2.1%). *Address diffusion* is modest but nontrivial (1.8/1.5/1.7%), while *controller oscillation* is comparatively rare ($\approx 0.8$–1.0%) and often transient. These patterns highlight the interplay between selection temperature $\tau$, comparison sharpness $\kappa$, and address concentration $\beta$: late discretization leaves BRANCH mixed; premature hardening locks in suboptimal routines; lagging $\kappa$ produces off-by-ones; and diffuse $\beta$ scatters writes across registers. The task-specific mitigations in the table align with these mechanisms: raising $\lambda_{\mathrm{ent}}$ and applying late straight-through (ST) hardening most effectively reduce

blended branches in **Sorting**; increasing $\kappa$ and adding a semantic penalty $\lambda_{\mathrm{sem}}$ curtail off-by-one updates in **Sum**; and ST combined with a mild curriculum stabilizes frontier dynamics in **BFS**. In practice, monitoring $H_{\mathrm{op}}$, EMA-smoothed gradient norms, and temperature/sharpness traces reliably flags emerging issues; applying mid-course entropy shaping and the listed remedies lowers failure rates and improves OOD accuracy at the longest lengths.

