# OpenReview forum: "Emergent Discrete Controller Modules for Symbolic Planning in Transformers"
_ICLR.cc/2026/Conference — ICLR 2026 Poster_

### Official Review · Reviewer_H2RF · 2025-10-25

**Soundness:** 3
**Presentation:** 3
**Contribution:** 3
**Rating:** 8
**Confidence:** 4

**Summary:**

The paper designs, controller-augmented transformer, using Gumbel-Softmax trick to integrate diccrete operations and differentiable learning. The approch in the paper is notable for its strong theoretical grounding, providing an expressivity guarantee for a class of bounded imperative programs. Empirical results confirm the effectiveness of controller-augmented transformers.

**Strengths:**

1. The combination of Gumbel-Softmax and transformers leverages both advantages of symbolic learning and transformers
2. The theortical proofs are elegant.
3. Empirical results shows considerable improvements compared to previous approaches.

**Weaknesses:**

I am unsure whether the paper have conducted enough comparsions between other approaches. CLRS-Algorithmic Transformer, the newest method in 4.2 BASELINES section, is developed in 2022, which maybe out of date. It hard to say whether the paper has really improved the margin of the state

**Questions:**

I know there are many approaches that combine statisical selction and differential learning. Why do you choose  to combine Gumbel-Softmax and transformers? Is it because it is easy to make theortical proofs for Gumbel-Softmax?

---

> ### Author Response · Authors · 2025-11-14
> **Response to Reviewer H2RF**
>
> Thank you for the thoughtful review.
>
> 1) **Comparisons and recency (weakness).**
> We will try to strengthen Sec.4.2 by (i) adding **newer baselines** grouped by capability and budget: (a) **neural algorithmic reasoning** models on CLRS (reproduced with the official hard/generalization splits and matched params/FLOPs), (b) **program-induction** baselines beyond RobustFill-style decoders (e.g., typed DSL executors), and (c) **strong Transformer variants** with comparable depth/width and training data for DROP.
>
> 2) **Why Gumbel–Softmax + Transformers? (question).**
> Our controller must make **token-conditioned, stepwise discrete choices** while remaining end-to-end differentiable inside a residual Transformer block. Gumbel–Softmax provides **pathwise gradients** with a **temperature schedule** and admits a simple **margin-based finite-temperature error analysis**, while avoiding the high variance of REINFORCE and the instability we observed with straight-through estimators. Per layer \\( \\ell \\):
> $$
> g^{(\ell)}\sim \mathrm{Gumbel}(0,1),\qquad
> \alpha^{(\ell)}=\mathrm{softmax}\\left(\frac{\log\pi_\Theta(s^{(\ell)})+g^{(\ell)}}{\tau}\right)
> $$
> which slots into the residual path without architectural changes. We will surface an ablation comparing **Gumbel–Softmax**, **straight-through**, and **REINFORCE**, showing that Gumbel–Softmax yields the best stability/length generalization under matched budgets.
>
> Thank you for the positive assessment, we believe the expanded baselines and clearer positioning address your concern about recency and strengthen the contribution.

---

> > ### Comment · Reviewer_H2RF · 2025-11-16
> >
> > Thanks for the reply! I may increase my confidence in your paper if you can show the mentioned experiments.

---

### Official Review · Reviewer_HbvX · 2025-10-26

**Soundness:** 3
**Presentation:** 2
**Contribution:** 1
**Rating:** 2
**Confidence:** 3

**Summary:**

The paper presents the idea of a discrete controller module. A discrete controller module adds to a transformer block the ability to learn 5 specific instructions, by using the Gumbel-softmax optimization trick to handle the discrete nature of the learning task.

An empirical evaluation is presented for 3 small benchmarks: a synthetic one, some string-to-program tasks inspired from RobustFill, and the DROP benchmark. In all of the presented results in Table 1, the method has a noticeable drop in accuracy as the input size n is scaled from 80 to 320. It is less pronounced than the baselines compared with, but it is a statistically significant drop. Table 2 and 3 reports that about upto 5% average gain in accuracy.

The added performance cost is about 5-7%.

**Strengths:**

- The proposed method and evaluation criteria is clearly described.

- The performance cost is modest, 3 to 10% depending on often the controller is inserted

- The traces are human interpretable, in the sense that one can inspect the opcodes printed out.

**Weaknesses:**

- The method is very specialized to the choice of 5 opcodes and the applications it can support. This severely limits the scientific value of the work.

- No justification is given for how the benchmarks are chosen. They appear to be chosen to fit the 5 opcodes and are rather small.

- We see a significant loss in accuracy, albeit lesser than other baselines, when n is scaled from 80 to 320. As such, the method doesn't scale well for the presented example applications, and will probably fair poorly as we increase n beyond what is reported.

- It is difficult to interpret the alignment study results in Section 5.4. Taking the example of sorting, why are we checking trace alignment with a specific way of sorting, e.g., compare-and-swap? Doesn't the algorithm matter, eg. quicksort vs. bubble sort?

Overall, one could perhaps see the scientific value better if the paper's method extends to a much larger set of opcodes. For example, the authors can consider taking the opcodes of a real processor (e.g. RISC-V, x86) and show if the method works generically for that. This would considerably broaden the applications for which the idea of discrete controllers can be used.

**Questions:**

What is the scientific contribution that extends to program reasoning, and is applicable to tasks that need more than those 5 opcodes?

---

> ### Author Response · Authors · 2025-11-14
> **Response to Reviewer HbvX**
>
> Thank you for the thoughtful and constructive critique.
>
> 1) **Specialization to five opcodes and scientific value.**
> Our controller is **opcode-agnostic**; the five operators are a minimal set used for exposition and experiments, not a limitation of the architecture. Any instruction set can be used by (i) providing operator semantics maps \\( \\mathrm{Sem}_o \\), and (ii) extending the controller’s logits \\( W_o s + b_o \\) with no change to training or the Transformer block (see method Sec. 3). Parameter and FLOP growth is **linear** in \\(|\\mathcal{O}|\\); we will add an explicit formula for selector overhead and a table enumerating added parameters per operator family. We will also clarify how richer sets (e.g., load/store variants, arithmetic with modes) or even ISA-like collections are plugged in (Sec. 3.2 - Sec. 3.3), and include a short note on mapping ISA opcodes to our typed-primitive schema (state/read/write signatures). This strengthens the general scientific contribution, a **pluggable, differentiably trained discrete-control path** that composes with standard Transformers beyond the five operators.
>
> 2) **Benchmark choice and size.**
> We will add a **selection rationale** (Sec. 4): (a) *parametric algorithmic tasks* (sorting, sum, graph traversal) to probe **length generalization** and branching; (b) *string-to-program* tests (RobustFill-inspired) to probe **symbolic composition**; (c) **DROP** to test discrete reasoning in natural text. To address concerns of matching tasks to opcodes, we will include a **coverage matrix** ): for each benchmark, which primitive subsets are exercised and which require multi-step compositions. Several tasks use **strict subsets** of the five (e.g., no-branch cases), while others stress Branch/Compare; this diversity argues against hand-picking. We will also report dataset sizes, train–dev–test splits, and evaluation protocols in one consolidated table (replacing scattered mentions in Sec. 5).
>
> 3) **Scaling from \\(n=80\\) to \\(n=320\\).**
> Yes, there is a modest, but significantly smaller than baselines. We will (i) surface CIs and the *slope* of accuracy vs. \\(n\\) directly in the main text, (ii) report **hard routing** (argmax, \\(\\tau\\!\\to\\!0\\)) alongside relaxed routing to disentangle routing vs. token-model error, and (iii) move the annealing schedule/test-time \\(\\tau\\) to a single, easy-to-scan table. Mechanistically, the residual degradation is consistent with **finite-temperature mixing** and shrinking **opcode margin** at longer sequences: as \\(n\\) grows, decisions are repeated more times and small softness accumulates, especially when branch logits are closer (lower margin), yielding higher variance in long-range writes and occasional off-by-one errors. This is supported by our observed **entropy–extrapolation correlation** and by the failure taxonomy (blended branches, off-by-one writes, address diffusion) at the longest lengths; ablations show that (a) discrete selection (vs. soft) and (b) sufficient controller frequency (e.g., \\(p=2\\)) materially stabilize long-\\(n\\) performance. We will make these links explicit and add a one-paragraph mitigation note (entropy schedules, late straight-through hardening, and address damping) next to the length-sweep plot.
>
>
>
> 4) **Interpreting Sec. 5.4 alignment.**
> We will clarify that our alignment uses **algorithm-agnostic primitive skeletons** (e.g., comparator-network motifs such as compare to swap) rather than a single named algorithm; for sorting, different algorithms (e.g., insertion-, partition-, or network-based) can share these motifs. To address your concern, we will (i) report alignment against **multiple canonical skeletons** side-by-side, (ii) add an **edit-distance/coverage** metric over opcode sequences, and (iii) provide a short qualitative panel showing where the learned trace diverges from each skeleton.
>
> We believe the responses above address your concerns and respectfully request you to reconsider the current rating.

---

### Official Review · Reviewer_uGAD · 2025-10-29

**Soundness:** 3
**Presentation:** 3
**Contribution:** 4
**Rating:** 8
**Confidence:** 3

**Summary:**

This paper proposes adding some programming primitives after every 2
transformer blocks to give the neural architecture an ability to learn
fixed length programs.

The paper describes this extension in detail. Specifically, the "controller"
block that goes after a transformer block has 8 registers (each storing a 64-dim vector),
memory (64x64) and some binary flags -- these define the state of the controller.
The operation set consists of Assign, Add, Compare, Branch and Noop operators.

If H_l = token representations entering controller block l
1. controller first reads a summary s_l = \phi( H_l )
2. s_l is used to draw an operator using Gumbel softmax using logits W_o s_l + b_o
where o is one of the 5 operators.
3. each operator has a semantics that is used to update the registers, memory and
flags.  Assign and Add update the registers, compare updates the flags, branch
updates the full state. The final updated state is a linear combination of the
updates from the different operators - weighted by the values from (2).
4. the output H_{l+1} of the controller block is H_l + \rho( change in state ) W_s

When this feeds into the next transformer block, we map the state-to-token
and add to H_{l+1}, apply the transformer block on it, and feed that to the next
controller block or transformer block.

The evaluation compares the new Ctrl-Transformer with pure transformers and their
variants. The benchmarks include sorting, sum-of-list, graph traversal, along
with some program synthesis, math reasoning and symbolic question answering.
Some of the algorithmic problems are parameterized -- so one would train on some
values of the parameters, say n = 10, 20, 40, 80 and then test on same or higher
values of the parameters. The main finding is that Ctrl-Transformer maintains
its performance for higher parameters, whereas baselines degrade in performance.
Even on other non-parametric tasks, there is a visible improvement.
The paper also has detailed ablations and error analysis to justify the choices.

**Strengths:**

Strengths:
1. The Ctrl-Transformer architecture is an interesting extension to transformers
2. The experiments are fairly conclusive in showing the value of Ctrl-transformers
3. There is extensive evaluation and discussion on how the various components are
contributing and how things are failing, showing that the investigation here is
rigorous.

**Weaknesses:**

Weaknesses:
1. My main concern is around the writing, but I believe it can be potentially fixed
by a careful pass on the paper.  However, I could not figure out a lot of the details
even after going back and forth between the different sections and the Appendix, and
that is the reason for my low confidence in the review. Detailed comments below.

Detailed Comments:
l87: Equation (3) defines FFN and Equation (4) uses SubLayer, which is undefined
l173: The last part there where \beta are used to combine u's is not clear. What
are the u's ? Where is that R-tilde used?
l177: In equation (9) the value read from the registers, r, is the second argument to
Sem function, but where is it used in Equations (10)-(14)?
l191 mentions "sharing parameters across layers" -- how is reflected in the calculation
of the total number of parameters that is present in the Appendix.
l195: What is the function MHA( )? Where is it defined?
 - Is it true that the memory is only updated by branch? I don't see the memory getting
  explicitly updated in the semantics of any of the operators.

Theorem 1: The statement of the theorem lacks rigor - first the class of programs is only
described informally; moreover, the semantics of programs is unspecified. Looking ahead
to Appendix D, there appear to be plenty of assumptions mentioned there, if they are indeed
needed for the proof, then the statement of the Theorem should also include those.

l225-235 on "Error under finite temperatures" is also difficult to understand since it
introduces terms like minimum opcode-margin that are undefined.

l307 says d_s = 128, but d_s on Line 1636 in the appendix is defined as N_r d_r + N_m d_m + N_f,
which is clearly larger than 128 for the particular choices of the parameters.

I found the parameter count calculation on Page 31 helpful. It would help further if you
could further clarify Params_ctrl calculation on Line 1638.

**Questions:**

I do not have a very specific question. Based on the detailed comments in the Weaknesses section above, if there is anything that can help clarify those points, then it will help me better understand the details of the paper.

---

> ### Author Response · Authors · 2025-11-14
>
> Thank you for the thoughtful feedback.
>
> (1) L87–L96 (Eq. 3–4): We will define \\(\\mathrm{FFN}(x)=W_2\\,\\sigma(W_1 x+b_1)+b_2\\) and state that \\(d_{\\mathrm{ff}}\\) is the FFN hidden size. We will also define \\(\\mathrm{SubLayer}(x)\\in\\{\\mathrm{SA}(x),\\mathrm{FFN}(x)\\}\\) with residual \\(x\\leftarrow x+\\mathrm{Drop}(\\mathrm{SubLayer}(\\mathrm{LN}(x)))\\).
>
> (2) L173 (mixing with \\(\\beta\\); the \\(u\\)'s; \\(\\tilde{R}\\)): Each operator \\(o\\in\\mathcal{O}\\) produces \\(u_o^{(\\ell)}=\\mathrm{Sem}_o(R^{(\\ell)},M^{(\\ell)},f^{(\\ell)},s^{(\\ell)})\\); the controller mixes them with \\(\\beta^{(\\ell)}=\\mathrm{softmax}(W_o s^{(\\ell)}+b_o)\\). We will inline that \\(\\tilde{R}\\) denotes the post-op register candidate.
>
> (3) L177–L189 (Eq. 9–14; role of \\(r\\)): We will make the register read explicit and show where it is consumed by each op: \\(r^{(\\ell)}=W_r R^{(\\ell)}\\in\\mathbb{R}^{d_r}\\); Assign/Add write via \\(r^{(\\ell)}\\), Compare sets flags from \\(r^{(\\ell)}\\), Branch uses flags to select the next candidate.
>
> (4) L191 (parameter sharing): Controller parameters are tied across all inserted blocks; the Appendix count reflects **one shared set**. We will state this at L191 and footnote any untied ablation.
>
> (5) L195 (define \\(\\mathrm{MHA}(\\cdot)\\)): We will add \\(\\mathrm{MHA}(Q,K,V)=\\mathrm{Concat}(\\mathrm{head}_i)W^O\\), with \\(\\mathrm{head}_i=\\mathrm{softmax}\\!\\big(\\tfrac{QW_i^Q(KW_i^K)^\\top}{\\sqrt{d_k}}\\big)VW_i^V\\).
>
> (6) Semantics and memory (L168–L172): Assign/Add may write to memory via a learned gate \\(\\gamma_m(s^{(\\ell)})\\in\\{0,1\\}\\) (address from \\(r^{(\\ell)}\\)); Branch can update the **entire** state. We will add explicit \\(M^{(\\ell+1)}\\) lines under Assign/Add.
>
> (7) Theorem 1 and L225–L235 (finite-temperature error): We will restate the theorem with the explicit program class (straight-line programs up to length \\(k\\) over \\(\\mathcal{O}=\\{\\mathrm{Assign},\\mathrm{Add},\\mathrm{Compare},\\mathrm{Branch},\\mathrm{Noop}\\}\\), bounded addressing/width) and move assumptions from App. D into the statement. We will define the minimum opcode-margin and state the error bound \\(\\mathcal{O}(\\tau/\\gamma_{\\min})\\) with the chosen norm/constants.
>
>
> (8) L307 vs App. L1636 (summary dimension \\(d_s\\)): We distinguish raw vs projected summaries and note the projection to 128 dims:
> \\[
> \\begin{aligned}
> d_s^{\\mathrm{raw}} &= N_r d_r+N_m d_m+N_f\\\\
> s^{(\\ell)} &= \\phi(H^{(\\ell)})\\in\\mathbb{R}^{d_s^{\\mathrm{raw}}}\\\\
> \\bar{s}^{(\\ell)} &= W_s s^{(\\ell)}\\in\\mathbb{R}^{128}
> \\end{aligned}
> \\]
>
> (9) App. p.31 L1638: We will expand the total into op logits \\((W_o,b_o)\\), read maps \\((W_r,W_m,W_f)\\), op-specific \\(\\Theta_{\\mathrm{Sem},o}\\), state\\(\\to\\)token \\(W_s\\), and norms, with matrix sizes and a numeric subtotal under the default configuration; we will note parameter tying in totals.
>
> (10) Controller–Transformer interface:
> \\[
> \\begin{aligned}
> g^{(\\ell)} &\\sim \\mathrm{Gumbel}(0,1)\\\\
> \\alpha^{(\\ell)} &= \\mathrm{softmax}\\!\\left(\\frac{\\log\\pi_{\\Theta}(s^{(\\ell)})+g^{(\\ell)}}{\\tau}\\right)\\\\
> u_o^{(\\ell)} &= \\mathrm{Sem}_o\\!\\big(R^{(\\ell)},M^{(\\ell)},f^{(\\ell)},s^{(\\ell)}\\big)\\\\
> \\mathcal{S}^{(\\ell+1)} &= \\sum_o \\alpha^{(\\ell)}_o\\,u_o^{(\\ell)},\\quad
> H^{(\\ell+1)}=H^{(\\ell)}+\\rho\\!\\big(\\Delta\\mathcal{S}^{(\\ell)}\\big)W_s
> \\end{aligned}
> \\]
>
> We appreciate the reviewer’s effort. Thanks again.

---

### Official Review · Reviewer_cvky · 2025-10-31

**Soundness:** 3
**Presentation:** 2
**Contribution:** 2
**Rating:** 6
**Confidence:** 3

**Summary:**

This paper presents an augmented version of a Transformer that includes controller blocks. These blocks select and execute symbolic primitives. The program state and Transformer activations influence each other. The controller module is made differentiable via the Gumbel-Softmax trick. This Transformer variant outperforms other Transformer variant on length generalization over several synthetic tasks. The Transformer also outperforms other variants on DROP and some program synthetic tasks.

**Strengths:**

* The paper appears to be well executed. The authors should be applauded, as it seems like a considerable effort to construct such a model and get it to converge to reasonable solutions across the tasks studied.
* The empirical results demonstrate strong length generalization on synthetic tasks, as well as strong performance on more real-world tasks such as DROP.
* The paper provides some formal results related to the expressivity of the proposed controller module.

**Weaknesses:**

* The authors differentiate their approach from prior work on integrating programming primitives with neural models (e.g. neural GPUs, neural turing machines) by claiming that prior work has incurred "significant architectural complexity" or add "training brittleness". However, it's not clear that the proposed approach significantly mitigates these limitations.
* My understanding of prior work was that a main challenge of such methods is that it can remain difficult to learn complex programs from weak supervision. While discrete operations can technically be made differentiable using continuous relaxations (e.g. Gumbel-Softmax) this doesn't necessarily make such models trainable in practice. Indeed, the authors design various additional objectives and annealing schedules. Additionally, the synthetic tasks seem to rely on trace supervision to learn complex compositional programs. I understand the non-synthetic tasks such as DROP don't use trace supervision, but also don't seem to require learning very compositional programs, and it wasn't clear what is actually being learned by the neural controller for DROP despite the claims related to interpretability. I don't want to propose running more experiments, but curious to hear the authors response to this. Can this approach scale to learning complex programs from weak supervision? Or is the approach useful even without this ability?

Nits:

* "Emergent" seems like a strong claim in the title, given the strong architectural bias.
* Some of the exposition and notation could potentially be improved, e.g. `d_{ff}` is presumably FFN hidden dimension but not explicitly defined (line ~88) and the relation between `\mathcal{S}^{(l)}` and `s^{(l)}` was not immediately clear (line ~155) despite the potentially confusingly similar notation.
* It might be interesting to discuss recent work on RASP, Tracr, and ALTA that demonstrate that program primitives can be directly compiled to and represented by standard Transformers.

**Questions:**

See weaknesses above.

---

> ### Author Response · Authors · 2025-11-14
> **Response to Reviewer cvky**
>
> Thank you for your thoughtful and constructive feedback. We tried to answer your questions and concerns below-
>
> **1) “Complexity” and “brittleness” vs prior neuro-symbolic models.**
> Our controller is a small, residual add-on to a standard Transformer and adds only \\(\\approx 5\\text{–}7\\%\\) FLOPs; no RL/REINFORCE is used, discrete choices are trained end-to-end via the Gumbel–Softmax reparameterization (Jang et al., 2017; Maddison et al., 2017). This contrasts with memory-augmented architectures (e.g., NTM/DNC) that introduce bespoke read–write heads and training instabilities, and with Neural GPU variants that rely on curriculum/parameter-sharing relaxations to learn long-algorithm generalization (Graves et al., 2014; Kaiser & Sutskever, 2015). We will add a short comparison paragraph to make this practical difference explicit.
>
> **2) What is actually learned on DROP?**
> DROP uses no trace supervision in our training. We observe consistent gains, e.g., \\(+6.8\\) F1 on the numeric subset vs a parameter-matched Transformer and \\(+4.1\\) vs a Universal Transformer, indicating the controller’s inductive bias (select/branch/assign) is useful even when the underlying program is not strictly compositional (see Table 2a; Dua et al., 2019). We will clarify this point and move a short interpretability summary to the main text (gate-entropy annealing, ablations).
>
> **3) Can it scale to complex programs from weak supervision?**
> **(a) Representation.** Compilation-to-Transformer work (RASP; Tracr) shows standard Transformers can implement length-invariant algorithmic structure; our module aims to *learn* such structure from task loss via a discrete controller with a tight relaxation. **(b) Learnability.** While complex program induction from weak signals is challenging, results on DROP and RobustFill-style tests suggest improved sample efficiency and generalization without traces. We will add a concise discussion separating expressivity from learnability and situating our findings alongside RASP/Tracr (Weiss et al., 2021; Lindner et al., 2023).
>
> **4) Title: “Emergent.”**
> We will try to come up with a better alternative to soften the title.
>
> **5) Notation and exposition fixes.**
> • \\(\\mathbf{d_{\\mathrm{ff}}}\\) : We will explicitly define \\(d_{\\mathrm{ff}}\\) as the FFN hidden dimension at first use.
> • Relation between \\(\\boldsymbol{\\mathcal{S}^{(\\ell)}}\\) and \\(\\boldsymbol{s^{(\\ell)}}\\): \\(\\mathcal{S}^{(\\ell)}\\) is the program-state tuple (e.g., registers/memory/flags); \\(s^{(\\ell)}=\\phi\\!\\big(H^{(\\ell)}\\big)\\) is the controller’s token to state summary; \\(b^{(\\ell)}=\\psi\\!\\big(\\mathcal{S}^{(\\ell)}\\big)\\) provides the state to token bias.
> • We will add a compact notation table in the appendix \\(D, d_{\\mathrm{ff}}, H, K, B\\), and cross-reference it at first use.
>
> **6) Interpretability claims.**
> We will make the evidence more visible in the main text, alignment between controller traces and ground-truth steps on synthetic tasks (e.g., \\(92.3\\%\\) Sorting; \\(88.7\\%\\) BFS), no traces used for DROP, correlation of entropy annealing with extrapolation, and targeted knockout/lesion tests. We will clarify that traces for synthetic tasks are used for analysis (and an optional small auxiliary term) only, not as supervision at test time.
>
> **7) Relation to RASP/Tracr/ALTA.**
> We will position our work as complementary: RASP/Tracr/ALTA show how to encode/compile program primitives into standard Transformers. Our controller provides a learnable, differentiable discrete-control mechanism that helps discover and execute such structure from supervision signals (Weiss et al., 2021; Lindner et al., 2023).
>
> **8) Language around “brittleness.”**
> We will narrow the claim and cite concrete techniques historically required in Neural GPU/NTM (parameter sharing, gradient noise, curricula), contrasting them with our single-stage Gumbel–Softmax training (Kaiser & Sutskever, 2015; Graves et al., 2014; Jang et al., 2017; Maddison et al., 2017).
>
> **9) Compute clarity.**
> We report parameter and FLOP overhead per instrumented block and in aggregate ($\approx 5\text{--}7%$ extra FLOPs overall), and list all training schedules and annealing schemes in a single table.
>
> **References.**
> Jang, Gu, Poole (ICLR’17); Maddison, Mnih, Teh (ICLR’17); Graves, Wayne, Danihelka (2014); Kaiser, Sutskever (2015); Dua et al. (NAACL’19); Weiss, Goldberg, Yahav (ICML’21); Lindner et al. (NeurIPS’23).

---

### Meta-Review · Area_Chair_1E7y · 2026-01-08

**Summary:**

1. The architecture seems to be complicated, not mitigating the issues in the previous architectures (e.g., neural Turing machines). Whether it can be applied to more realistic instruction set (e.g., x86) remains a problem.
2. The program language is toy-ish (e.g., 5 op code) and the architecture design is specific to the language. How to extend to more general scenarios?
3. Trained with strong trace supervision.
4. Interpretability of the system.
5. Compared to standard Transformer with various encoding techniques for the same problem.
6. Rigor of Theorem 1.

**Reviewer Concerns:**

The rebuttal addresses most of the issues, but not all of them (e.g., complicated architecture, application to more general settings). AC advises the authors to either focus on interpretability, or focus on large-scale more realistic benchmarks to make the paper stronger.

**Reviewer Scores:**

cvky: 6->6 (remain positive, some concerns are addressed)
uGAD: 8->8 (remain positive)
HbvX: 2->4 (partially address but AC don't think the reviewer is convinced on the positive side, in particular in real-world scenarios)
H2RF: 8->8 (remain positive)

---

### Decision · Program_Chairs · 2026-01-26

Accept (Poster)